# Sedimentary record of historical seismicity in a small, southern Oregon lake

Ann E. Morey[1,2], Mark D. Shapley[3], Daniel G. Gavin[4], Alan R. Nelson[5] and Chris Goldfinger[1]

[1]College of Earth, Ocean and Atmospheric Sciences, Oregon State University, Corvallis, OR 97331, USA
[2]Cascadia Paleo Consulting, Corvallis, OR 97330, USA
[3]Continental Scientific Drilling Facility, University of Minnesota School of Earth and Environmental Sciences, 116 Church St SE, Minneapolis, MN 55455, USA
[4]Department of Geography, 1251 University of Oregon, Eugene, OR 97403-1251, USA
[5]Geologic Hazards Science Center, U.S. Geological Survey, Golden, Colorado 80401, USA

*Correspondence to*: Ann E. Morey (ann@cascadiapaleo.org)

**Abstract.** We compare event deposits from the historical portion of the sedimentary record from Lower Acorn Woman Lake, Oregon, to historical records of regional events to determine if the lake records Cascadia earthquakes. We use the sedimentological characteristics and X-ray fluorescence (XRF) provenance of disturbance deposits (labelled A-J) from the historical portion (~1650 CE and younger) of the record to discriminate between deposit types. We show that earthquake-triggered deposits can be differentiated from flood deposits, and Cascadia earthquakes deposits can be differentiated from other types of earthquake deposits. Event deposit J dates close to 1700 CE (1680-1780 CE) through multiple approaches, suggesting it was the result of shaking from the M8.8-9.2 1700 CE Cascadia megathrust earthquake. Event deposits H and I are interpreted to be a complex sequence deposited in response to the ~M7.0 1873 CE Brookings earthquake which we argue is a crustal earthquake rupture sequence, possibly with a subduction component. These results demonstrate the usefulness of lake sediments to infer earthquake hazards in Cascadia.

## 1 Introduction

### 1.1 Approach

Lake sediments can provide high resolution, continuous records of earthquake-triggered disturbances (Goldfinger et al., 2017; Howarth et al., 2014; Moernaut et al. 2014; Monecke et al., 2004; Praet et al., 2022; Strasser et al., 2013). They have even been used to distinguish the imprints of megathrust and intraplate earthquakes (Van Daele et al. (2019). Lakes are also good recorders of other types of disturbances, such as floods (Gilli et al., 2013; Wilhelm et al., 2013, 2018; Praet et al., 2020), postfire erosion (Colombaroli et al., 2018), flood-induced erosion (Howarth et al., 2012), and wildfires (Bradbury, 1996; Long et al., 1998; Colombaroli et al., 2010; Hennebelle et al., 2020). Lake sediments in Cascadia are increasingly exploited for their paleoseismic potential (Leithold et al., 2018, 2019; Goldfinger et al., 2017; Morey et al., 2013); however, most other studies that have examined the differences between earthquake-triggered deposits and those from other types of disturbances are from

large lakes ($>100$ km$^2$). Understanding the influence of Cascadia earthquakes on lakes is crucial because researchers have long suspected an influence from megathrust earthquakes in Cascadia lakes in the Oregon Coast Range (Long et al., 1998), Lake Washington (Karlin & Abella, 1992, 1996; Karlin et al., 2004), the Olympic Peninsula (Leithold et al., 2018), Seattle area (Goldfinger et al., 2017), and in the Klamath Mountains and Coast Range of Oregon, (Morey et al., 2013). Understanding the

impact of megathrust earthquakes in Cascadia lakes has the potential to provide insight into Cascadia earthquake behaviour, provides opportunities to obtain information about intraslab and crustal earthquakes, and improve the interpretation of environmental proxy data.

Differentiating between flood and earthquake-triggered deposits can be challenging. Some research suggests that flood-triggered turbidites are more evenly distributed throughout the lake and earthquakes are thickest in the lake's depocenter

(Vandekerkhove et al., 2020, and references therein). Other studies show that flood deposits exhibit reverse, then normal, grading in contrast to the normal grading of turbidites (Mulder et al., 2003; St-Onge et al., 2004; Beck, 2009; Wirth et al., 2011) or contain larger clastic particles compared to background sediment (Toonen et al., 2015; Schillereff et al., 2014; Campbell, 1998). Many factors contribute to the characteristics of the resulting deposits (particle size, clastic supply, slope angle, basin shape and slope stability, etc.), therefore each lake must be evaluated independently.

To determine if small ($<10$ km$^2$) lakes record evidence of Cascadia earthquakes and ascertain how the resulting deposits differ from other types of event deposits (such as flood deposits), we investigated the sedimentary record from Lower Acorn Woman Lake, Oregon, located ~180 km inland of the Cascadia subduction zone trench. We compare the historical portion of the sedimentary record to the record of extreme events known to influence the region. The lake is an ideal study site because it has experienced extreme events, including earthquakes and floods, and the bedrock that immediately surrounds the lake is

locally distinctive from the bedrock of the steep watershed that contributes sediment to the north end of the lake. This heterogeneity of bedrock is important because it improves the ability to determine sediment provenance and suggests possible mechanisms controlling deposition. Lower Acorn Woman Lake is also located adjacent to Upper Acorn Woman Lake, which has an existing record of watershed disturbances, some of which are already suspected to be the result of Cascadia earthquakes (Morey et al., 2013; Colombaroli et al., 2018).

**1.2 Background**

**1.2.1 Setting**

Upper and Lower Acorn Woman Lake (42°01′55″ N, 123°00′56″ W) are located in the Klamath Mountains, ~180 km inland of the trench (the surface expression of the Cascadia Fault; Figure 1, top left), at an elevation of ~920 m. The lakes formed when a landslide dammed Acorn Woman and Slickear Creeks near their confluence, creating two basins draining watersheds

of different sizes and bedrock types. The lakes are located near the southern extent of the Cascadia subduction zone, just inland of the boundary between the obliquely subducting (~27-45 mm/yr) Juan de Fuca Plate and the Gorda deformation zone ~200 km north of the northward migrating Mendocino triple junction. The lakes are ~35 km above the inferred location of the

transition from seismic to aseismic slip on the Juan de Fuca Plate (yellow star in Figure 1, left; McCrory et al., 2014; Yeats, 2004) near the zone of maximum episodic tremor density.

The tectonic setting at Acorn Woman Lakes is complex. The lakes are located within the Klamath Mountains Province (KMP), which acts as a relatively rigid block which is subjected to forces related to the northward migration of the Mendocino triple junction (MTJ; and associated Mendocino Crustal Conveyor (MCC)), the NNW translation of the Sierra Nevada – Great Valley block (SNGV), the influence of oblique subduction of the Gorda and Juan de Fuca Plates from the north and west, and Basin and Range Province (BRP) extension from the east (McKenzie and Furlong, 2021). As a result, the KMP is currently moving

NNW at ~8-12 mm/yr, primarily as a result of pushing from the south by the SNGV block (McKenzie et al., 2022). The Franciscan complex, which surrounds the western boundary of the KMP, is accommodating high NNW and NE directed shortening strain related to the MCC and subduction coupling. McKenzie et al. (2022) state that these forces, including the inferred injection of Franciscan crust into the southwest KMP which they suggest contributes to KMP uplift, may explain the faulting regime within and west of the KMP over the past ~5 my.

The KMP is bounded to the east by the Walker Lane – Eastern California Shear Zone (WLECSZ), which extends northward from the Gulf of California to Crater Lake (Waldien et al., 2019). The WLECSZ is a shear zone, indicating it is an incipient transform fault, which is propagating northward independent of the MTJ (Waldien et al., 2019). Although the majority (~75%) of the Pacific-North American plate motion is currently along the San Andreas transform fault, ~25% of the motion is along the WLECSZ (Thatcher et al., 1999; Oldow et al., 2001; Bennett et al., 2003). Coincident with the WLECSZ at the approximate

latitude of Acorn Woman Lakes is the northwest expanding extensional BRP, which terminates at the Western Klamath Fault Zone (WKFZ; Waldien et al., 2019).

The most likely sources of regional seismicity with the potential to disturb Lower Acorn Woman Lake sediments are earthquakes within the subducting plate, megathrust earthquakes on the plate interface, and earthquakes on crustal faults in the overriding North American plate. The USGS Quaternary Fault and Fold database for the United States (with additional

information from the California Geological Survey; accessed May 2019, https://earthquake.usgs.gov/hazards/qfaults/) identifies few active regional faults, however the simplified Cascadia forearc fault model of Wells et al. (2017) identifies a series of trench perpendicular faults along the coast of northern California and Oregon. An additional likely active fault, the NNE trending Cave Junction fault (also called the Eight Dollar Mountain fault), has been recently identified in the region ~50 km northwest of Acorn Woman Lakes (von Dassow and Kirby, 2017; Kirby et al., 2021; shown on Figure 1).

The largest historical earthquake in Oregon since 1700 CE was a ~M7 earthquake that occurred on November 23, 1873 (Wong, 2005). This earthquake was strong enough to topple chimneys in Jacksonville, Oregon, (Wong and Bott, 1995) 15–20 km east of Acorn Woman Lakes and has been interpreted as an intraplate earthquake primarily because of the lack of reported aftershocks (Wong, 2005). Numerous investigations of felt reports published in regional newspapers suggest the intensity centre was located ~10 km inland from the coast, from just south of Cape Blanco, OR, to Crescent City, CA (Bakun, 2000;

Toppozada et al., 1981).

Both Upper and Lower Acorn Woman Lakes are situated within the Condrey Mountain Schist ("lake bedrock," dark grey unit in Figure 2), a heavily foliated quartz-muscovite schist (Hotz, 1979), that has been described as failure-prone (Coleman et al., 1983). The northern portion of Lower Acorn Woman Lake is fed primarily by Slickear Creek, which is almost entirely located in a unit mapped as metavolcanics sediment and flows (andesite) and quartz-diorite ("Slickear Creek watershed bedrock," orange unit shown in Figure 2). The boundary between these two units is located at the Slickear Creek delta. The Slickear Creek watershed rocks have a different composition and are more resistant to erosion than the schist that surrounds the lakes and most of the Acorn Woman Creek watershed.

### 1.2.2 Climate

The Klamath Mountains ecoregion experiences a Mediterranean climate characterized by hot dry summers and wet winters (Sleeter & Calzia, 2012). The wet winters are the result of equatorward shifts in midlatitude storm tracks during the winter months (Swain et al., 2018). The latitude at which this shift occurs is variable through time and can result in extreme shifts between flooding and drought (Horton, et al. 2015). Atmospheric rivers are narrow pathways of tropical moisture that are regionally important because they provide a large amount of rainfall and snow at high elevations to the region (Goldenson et al., 2018). Sustained atmospheric river events can produce extreme flooding (Safeeq et al., 2015), such as occurred during the 40-day event that occurred in 1861–1862 (Engstrom, 1996).

### 1.2.3 Sediment transport

For the post-logging era (1930–present) the dominant influence on sediment accumulation rates identified from the sedimentary record from Upper Acorn Woman Lake is rainfall (Colombaroli & Gavin, 2010; Colombaroli et al., 2018). Prior to that, the largest accumulation rates are related to postfire erosion and possibly earthquakes, as part of complex feedback processes (Colombaroli & Gavin, 2010; Colombaroli et al., 2018). Slope failures and slumps around Lower Acorn Woman Lake are common on the steep hillslopes and were observed as changes in the landscape and vegetation, suggesting possible instability during shaking. Rainfall, the dominant influence on slope wash from hillslopes to streams in upland regions (Lamoureux, 2002; Zolitschka, 1998), occurs primarily during the wet season from November to April (Sleeter & Calzia, 2012). Stream bank outcrops suggest occasional extreme, erosive flow. Snowmelt floods, which occur when rain-on-snow events melt snow in the upper reaches of the watershed, may also introduce pulses of sediment into the lake. Flash floods in Slickear Creek have been observed to transport and deposit sand to boulder-sized on the delta near the lake margin where vegetation is dense (Bert Harr, personal communication, September 2015; landowner).

### 1.2.4 Lake and watershed characteristics

Lower Acorn Woman Lake (previously called Lower Squaw Lake) is a long, narrow (area = 0.226 km$^2$), deep (~40 m) lake at 915 m elevation. The Slickear Creek watershed to the north is smaller (7.7 km$^2$) than the Acorn Woman Creek watershed to the east (40.2 km$^2$). The level of Lower Acorn Woman Lake was raised above its natural level by ~5 m in 1877 when a dam

was built to increase water pressure for hydraulic gold mining (Jacksonville Times, September 25, 1878). The ~0.5 km long Slickear Creek delta is composed of coarse sand, cobbles, and a few boulders near the shore of the lake where vegetation is dense. The delta has been built rapidly by floods that have occurred every ~10–20 years, occasionally depositing a thick layer of coarse sediment over the entire delta (Bert Harr, personal communication, September 2015; see Table 1). Most of the water flows into the lake from the north as subsurface flow; however overland flow occasionally occurs along the narrow (a few meters wide) but incised stream channel on one side of the delta.

Upper Acorn Woman Lake (previously called Upper Squaw Lake) is a small (0.073 km$^2$), shallow (14.2 m) lake at ~930 m elevation with a capacity of ~564,000 m$^3$. Upper Acorn Woman Lake drains a large watershed (40 km$^2$) of steep terrain (~1,020 m relief), and Acorn Woman Creek flows throughout the year into the southern portion of Lower Acorn Woman Lake near the dam. Although the terrain is steep throughout much of the watershed, the proximal ~2.0 km near Lower Acorn Woman Lake becomes gently sloping, and the creek meanders and branches as it nears the lake, then enters the lake over a delta front composed of angular, well-sorted, medium to coarse sand. Groundwater likely flows through the delta, as water-tolerant trees and shrubs are present.

The Atlas of Oregon Lakes (Johnson, 1985) describes Lower Acorn Woman Lake as an unusually deep lake for its size, with a high concentration of ions, especially of calcium and magnesium. Alkalinity and conductivity are also high, with pH value of 8+. It has been classified as a mesotrophic to oligotrophic lake based on a secchi disk depth of 6.2 m. The lake shows evidence of oxygen depletion at depth. A phytoplankton sample taken on 7/13/1982 identified the dominant species as *Ceratium hirundinella*. Also present were *Dinobryon sertularia*, *Melosira granulata* and *Asterionella fomosa*. Water column measurements of temperature, oxygen and specific conductance were acquired in 2014 (this study) which are presented in Figure 3 (right) along with the water column data collected by Larson et al., 1975.

Upper Acorn Woman Lake continuously overflows into Lower Acorn Woman Lake via Acorn Woman Creek. Water accumulates in Lower Acorn Woman Lake from upper and lower lake watersheds (Figure 3, left). Local people and Forest Service employees observed flood waters in 1997 filling Lower Acorn Woman Lake to capacity, forcing water, sediment, and downed trees to the south, blocking the outflow and raising the lake level above the dam (Peter Jones, personal communication; January 2020). Accounts of this event describe Lower Acorn Woman during this event as a wide, fast flowing stream that undercut the lake shore resulting in soils and colluvium to slump into the lake.

## 2 Methods

### 2.1 Sediment cores

We collected sediment cores from Lower Acorn Woman Lake during field seasons in 2013, 2014, and 2015. We used a modified Livingstone corer (Wright, 1967) deployed from a custom platform fitted with a stainless-steel pipe attached to two inflatable rafts (2013) or canoes (2014), to collect Cores 1, 2, and 4,5 (overlapping drives at a single location). We used a Kullenberg piston corer (Kelts et al., 1986) to collect cores in 2015 and collected surface samples from the same locations with

a gravity corer, both deployed from an aluminium platform supported by two 7-m skiffs (Continental Scientific Drilling

Facility, CSD; 2015). We acquired single-beam bathymetric data in May 2015 by canoe fitted with a Garmin GPS-enabled "fish finder" and receiver. In addition, a 10 m composite sediment core from Upper Acorn Woman Lake, taken from near the lake centre (water depth of 14.1 m) that contains a record of watershed-sourced deposits over the past ~2,000 years (Colombaroli & Gavin, 2010; Colombaroli et al., 2018).

## 2.2 Sediment properties

We described the sedimentology and deposit characteristics of core sediment using the following data types: Munsell colour, sediment texture, composition (microscopic analysis of smear slides), and grading and contact characteristics (sharp, gradational, discontinuous, etc.). We acquired particle-size data at 0.5 cm intervals through event deposits, and less frequently between them. Particle-size data (volumetric % by size) were determined by laser diffraction analysis using a Horiba Grain Size Analyzer (LA-920; CSD, University of Minnesota) or Beckman Coulter Grain Size Analyzer (LS 13-320; Oregon State

University). We measured volume magnetic susceptibility ($k$) using a Bartington MS2E point sensor at 0.5 cm resolution. We acquired combustion data at 0.5–1.0 cm intervals through event deposits and less frequently elsewhere, resulting in data for percentage of inorganic content (clastic particles other than $CaCO_3$), percentage of organic matter, and percentage of $CaCO_3$ (calculated from dry weights). We acquired CT imagery and data using the Toshiba Aquillon 64 slice CT unit at the Oregon State University Veterinarian Hospital (at 0.5 mm resolution). Inferred minerology was spot-checked using the CSD desktop

scanning electron microscope (Hitachi TM-1000).

## 2.3 Identification of event deposits and lithostratigraphic correlation

We identified event deposits in cores as abrupt increases in petrophysical property data (magnetic susceptibility and CT imagery and data – also called radiodensity) in contrast to the typical organic-rich background sediment, then used lithostratigraphic methods to correlate units between cores. Petrophysical properties typically reflect the vertical grain size

distribution of the particles in marine cores (Kneller and McCaffrey, 2003; Goldfinger et al., 2012; Patton et al., 2015) and lake cores (Karlin and Seitz, 2007), however they have also been shown to also reflect the inorganic content in lake cores dominated by organic sedimentation (Morey et al., 2013).

Lithostratigraphic correlation takes advantage of the characteristics of both the sequence pattern of event deposits as well as the characteristics of the petrophysical properties through the event deposits themselves. The petrophysical properties of the

event deposits can be considered fingerprints of the time history of deposition of the event deposit (Goldfinger et al., 2008, 2013; Patton et al., 2015), and individual events from independent records have been shown to correlate over long distances even though they are from different depositional settings (Goldfinger et al., 2012; Morey et al., 2013). Radiodensity has been shown to be the most sensitive property to changes in fine-grained inorganic disturbances (Inouchi et al., 1996), and does not display as much of an edge effect as magnetic susceptibility, therefore the high-resolution radiodensity was heavily relied upon

for correlation.

## 2.4 Sediment provenance data

We used X-ray powder diffraction spectra (XRD) to determine the mineralogy of the two endmember bedrock types. XRD allows qualitative and semiquantitative analysis of the mineralogy of sediments and rocks by measuring the diffraction properties of their mineral components. We interpreted the results using the automated pattern-matching routine in Jade Software (http://ksanalytical.com/jade-9/), which compares the relative peak heights and areas from unknowns to those from samples of known mineralogy contained in the software database.

We acquired X-ray fluorescence data (XRF) with an ITrax core scanner (Oregon State University) from downcore sediment at 0.4 mm intervals and from discrete samples of lake-margin beach sand and Slickear Creek streambed sand (also using the ITrax core scanner; source locations are shown as blue triangles in Figure 3, left). The XRF downcore data were used to determine the upper and lower boundaries of each deposit in addition to identifying sediment provenance.

## 2.5 Development of event-free stratigraphy and age-depth model

Event-free stratigraphy. Rapidly deposited sediment was removed from the stratigraphic sequence prior to creating the age-depth model to avoid misinterpreting it as being deposited at the same rate as background sediment. This event-free stratigraphy was created by identifying the event deposit boundaries using XRF and estimating missing sediment at erosional contacts using correlations between shallow and depocenter cores. XRD data from endmember rock samples were used to initially determine the best choice of elemental variables to use as XRF provenance indicators. XRF variability through the event deposits was then used to determine where deposit boundaries exist. Sharp increases in sediment density (higher HUs; lighter values), compared to lower density background sediment (lower HUs; darker values), indicate rapid deposition or reworking as described in Morey et al. (2013).

Radiocarbon samples and data. We sampled Lower Acorn Woman Lake sediment cores for radiocarbon dating after splitting cores longitudinally. We removed macroscopic samples of fragile plant material (such as fir needles and buds) from the targeted horizons of undisturbed sediment, cleaned and dried them, then had them analysed by AMS (accelerator mass spectrometer) for radiocarbon. We selected the target horizons for sampling based on a suspected temporal tie point between the Lower Acorn Woman Lake record and the dated sequence from Upper Acorn Woman Lake. We used the strong similarity in sequences between the upper and lower lakes to infer that the first of these clastic event deposits was deposited close to 1964–1965 (as shown in the Supplementary Data).

Age-depth model. An age-depth model was developed using a Bayesian approach in OxCal (v 3.4.2; Bronk Ramsey, 2017).

# 3 Results

## 3.1 The historical record of extreme events

Historical events with the potential to influence Lower Acorn Woman Lake sedimentation are compiled from personal accounts (from landowners and U.S. Forest Service Rangers), published hydrologic data, regional historical newspapers, and Forest Service documents (Table 1). We did not include large land-use events (logging efforts and road building) or wildfire in Table 1 because these events require water to transport the resulting increase in available sediment into the lake, however extreme runoff from these types of events can cause debris flows (Wall et al., 2020). Homesteading began in the region when gold was

found by settlers between 1850-1852 CE (Lalande, 1995).

## 3.2 Sediment core locations and recovery

    This study investigated historical records from the northern cores (Figure 4; orange circles) near the Slickear Creek delta, which is saturated near the surface. Sediment core locations, lengths and water depths are shown in Table 2. Several of the upper sections of the first Kullenberg cores were distorted (especially the historical portions) during coring and stuck in the

casing, therefore the focus is on the shallower water cores (with comparison to depocenter cores where possible). Small adjustments to composite sediment depths accounted for core section breaks and minor distortion.

## 3.3 Sediment facies

Background facies. Background sediment is a very dark brown to black (Munsell colour: 2.5YR 2.5/1) organic-rich sediment containing planktonic diatoms (~30%), particulate organic matter, and angular, poorly sorted medium to coarse silt (50–60%).

Split sections change colour quickly (over a period of hours to days) from very dark brown (or black if the core was taken in deep water) to a lighter brown, or slightly orange colour and become concreted if exposed to air. A shift in sediment radiodensity occurs just below deposit I in all cores (see Figure 5). This horizon, referenced in Figure 5 and Table 3 as "inflection," refers to the change in sediment radiodensity that is suspected to reflect changes in sedimentation from around the time settlers cleared land (mid 1800's) and first dammed the lower lake (1877 CE). Loss on ignition and physical property

changes associated with this shift are shown in Figure 5 (data are from core SQB2; Table 3).

    Event deposit facies. Ten event deposits from the sediment cores were identified as abrupt increases in sediment radiodensity compared background levels (based on the downcore CT radiodensity data). The events identified in core SQB5 using this method are labelled A–J in Figure 5. Disturbance event facies are inorganic layers (> 80% clastics of total by weight) of two primary types: a lighter grey (Munsell colour: 2.5Y 4/1) medium-grained silt without visible mica flakes, and a darker grey

(Munsell colour: Gley2 4/5PB) coarse micaceous silt. We observed two thick (5–25 cm, depending on core location and water depth), visually similar, massive to normally-graded silt units in the upper ~100 cm of each core. A third type of event deposit facies is slightly denser than background sediment with little change in magnetic susceptibility, and although visible by eye,

not identifiable as different by Munsell colour. This facies type is more common in the cores but is generally thinner compared to the other types of event deposit facies.

## 3.4 Characteristics of event deposits A-J

### 3.4.1 Deposits A and B

Deposits A and B are thick units (5–20 cm, depending on the location of the core in the lake; Figure 6) with sharp basal contacts. These disturbances are found throughout the lake. They vary with distance from Acorn Woman Creek in the south: deposits are thicker and more complex to the south and show evidence of erosion in all cores except those recovered from near the lake's depocenter (cores SQB9 and SQB10). Interevent sediment sections are also thicker and more complex in the south. Basal sediment contains rootlets and other particulate and degraded organic matter (Figure 7). Grading proceeds from poorly sorted coarse silt and fine sand upward to well-sorted fine-medium silt, followed by a thin, poorly sorted multimodal fine silt and thin (< 1 cm) silty-clay tail. The deposits are grey (2.5Y 3/2), however, the lower half of deposits A and B are browner compared to the upper half of the deposit. Smear-slide inspection suggests this brown colour is from degraded organic matter entrained in the sediment. Magnetic susceptibility is highest near the top of the sequence, just below the silty-clay cap. Although deposits A and B have similar characteristics, the base of deposit A has a thin layer of lighter-coloured coarse silt-fine sand without visible mica, which was not observed at the base of deposit B. The presence of rootlets and broken diatoms in the lower portion of the deposit and sharp contact indicate erosion and reworking of lake-margin sediment.
Core log data for deposits C-J are shown in Figure 8.

### 3.4.2 Deposit C

Deposit C is a fine-grained (medium silt), light-coloured (5Y 2.5/2) unit that is wavy and discontinuous. It is thin (<1 cm) and becomes thinner with distance from Acorn Woman Creek but is thicker (~3 cm) and slightly brown (but still 5Y 2.5/2) in colour in core SQB15. It is present in all cores except for core SQB5 (where it has been eroded away).

### 3.4.3. Deposit D

Deposit D is a sequence of two thin (<0.5 cm) wavy medium silt layers in background sediment that is slightly stiffer than the surrounding sediment. The basal layer is distinctive because it appears orange (5Y 4/1).

### 3.4.4. Deposit E

Deposit E is a thin (~1 cm) dark grey (GLEY2 4/5PB) medium-grained, poorly sorted silt deposit containing visibly large mica flakes. This deposit is present only in core SQB5. The deposit has a sharp base and is normally graded. We observed rootlets and other organic matter, particulate and degraded, near the deposit base.

### 3.4.5. Deposit F

Deposit F is a slightly lighter grey (compared to background, 5Y 2.5/2), normally graded, medium silt unit with a sharp basal contact. Both magnetic susceptibility and radiodensity are higher than background sediment, suggesting a high concentration of inorganic particles, supported by the loss on ignition data indicating a 5% increase in inorganic content compared to background (data from SQB2A). Loading of this silt layer into the organic sediment below suggests rapid deposition of denser sediment (most obvious in the radiodensity image of SQB2A at about 40 cm depth in Figure 8).

### 3.4.6. Deposit G

Deposit G is visually indistinct with a gradual increase and decrease in radiodensity (see radiodensity trace in Figure 9). There is little change in magnetic susceptibility through the deposit. It is ~3 cm thick (based on radiodensity) and found in all the northern cores. Smear-slide data do not show differences in composition through the deposit; however, a 2-3 percent increase was observed in the inorganic content (with a peak near the centre of the deposit). The base and top are indistinct; peak radiodensity occurs at the midpoint of the deposit.

### 3.4.7. Deposit H

Deposit H (Figure 9) is slightly lighter grey (2.5Y 4/1 at the very base) and appears slightly stiffer than background sediment. It is a thick unit (basal silt is ~1 cm, and radiodensity suggests the entire deposit may be up to ~4 cm thick in Core SQB2) with normal grading and a sharp basal contact. Grading proceeds from poorly sorted medium silt upward to a more well-sorted medium silt, and loss on ignition data indicate an upward increase in the ratio of organics to inorganics with grading. This deposit tail appears mottled in the radiodensity imagery. Deciduous leaves were observed at the basal contact in some of the cores.

### 3.4.8. Deposit I

Deposit I (Figure 9) is a dark grey (GLEY2 4/5PB) coarse silt dominated by large, visible mica flakes (~90% inorganic). The deposit has a sharp basal contact and is initially reverse (upward coarsening), then normally graded. The reverse-graded portion of the deposit has a higher percentage of organics (including rootlets) compared to the normally graded portion of the deposit. The base of the deposit is very sharp with evidence of erosion (truncated beds). Correlation to other cores indicates missing sediment below it, more so in SQB1, SQB2 and SQB14 compared to SQB5. Deposit characteristics (mica-rich graded deposit) are similar to deposit E; however, unlike deposit E, deposit I is found in all cores.

### 3.4.9. Deposit J

Deposit J (Figure 10) is a lighter grey (2.5Y 4/1) silt unit. It is thick (~7-15 cm), dense (~1,000 HU at the base), weakly graded, medium to fine-grained silt unit with a long tail. The silt is fine-grained and well-sorted (~90% inorganics) at the base lacking

other components such as reworked diatoms and organic matter. The visible layer of silt is 3–4 cm thick and becomes less well sorted (but not finer grained) upward. Percentage of organic matter increases upward with fining. There is evidence of loading of the silt into the less-dense sediment below (Figure 8, right), suggesting rapid deposition, of the silt into the less-dense sediment below. The silt identified as deposit J is preceded by a thin silt unit with an organic tail (Figure 10).

### 3.5 Radiocarbon results

Radiocarbon determinations are shown in Table 4. Samples 6, 7 and 8 are included here even though they are more than 1000 $^{14}$C yrs BP because they provide a key temporal tie point between the cores. Sample 0 was not included in the age model because it is much older than the others in the sequence (suggesting it does not represent the time of deposition). Samples 1-4 were used to create the age-depth model for the historical portion of the core, and samples 5-7 are included here to show the temporal relationship between cores at the lower end of the section used in the age-depth model. Note the close similarity in
ages between samples 6 and 7.

### 3.6 Composite core section and correlation points among cores

We used the radiocarbon ages from detrital plant fragments (Table 4), core imagery and descriptions and physical property data to (a) create the composite core, SQB1/2/ss, for the historical portion of the Lower Acorn Woman Lake record, and (b) identify stratigraphic tie points for the northern cores (Figure 11). Radiodensity was heavily relied upon for correlations
because of it is very high resolution (sub-mm scale; see Figure 11 inset). Little sediment is missing between each 1 m section in core SQB2, based on a comparison to SQB1, therefore only section, SQB1A, was needed to complete the splice. We used the surface sample (ss) to reconstruct the two upper event deposits because they are missing from the top of SQB2 and SQB1. Note that only the northern cores were used for the splice because core length differences from north to south suggested large changes in sedimentation rate.

The radiocarbon data were used to tentatively link cores for the splice. The numbered stars in Figure 11 identify the locations of radiocarbon samples 1–3 (Table 4). We used additional radiocarbon data (samples 4–8) to create the tie points and splices but did not include them in the age model for the historical record because they predate it. The radiocarbon age for Sample 0 (grey text) is too old (reversed) for the sequence and we did not use it. We identified the historical portion of the record as just younger than Sample 3 because this radiocarbon age likely represents a horizon that is older than the 1700 CE earthquake. We
used this horizon to define the lower boundary of the historical section of the record. A splice table (Table 5) shows the depth equivalencies of sections used to create the composite core. Note that the length of the SQB-ss image was adjusted to match the stratigraphy of SQB2-A. The age model was created using the upper 2.5 m of the composite section. Sections showing significant erosion are the surface samples; these cores are shown in false colour (far left) to highlight the stratigraphy and to improve the identification of erosional contacts and missing sediment.

Lake-wide tie points were created using the distinctive sequence of event deposits in the record using core-log correlation and radiocarbon data where these data were available (Figure 12). The distinctive sequence includes two, thick upper disturbances,

followed by one disturbance with two at the very end of the sequenced used for the upper portions of the cores. The relationship between the Lower Acorn Woman Lake and Upper Acorn Woman Lake sequences is shown (Figure 12, left).

A compilation of all core data for key cores from Lower Acorn Woman Lake are shown in Figure 13.

## 3.7 Sediment provenance data

### 3.7.1 XRD

We identified provenance endmembers by XRD analysis of watershed and lake bedrock samples (Table 6; data available in the Supplementary Data). The bedrock that surrounds the lake is composed of a quartz muscovite schist with chlorite minerals, similar to what has been mapped, whereas the Slickear Creek bedrock is composed of Ca- and Fe- amphibolites, chlorite minerals and albite. We also analysed silt samples from the upper portion of core SQB2, and a gravel sample from the base of the core. Results confirmed a schist bedrock source for the dark grey silt layers containing visibly large mica flakes, and a Slickear Creek watershed source for the lighter grey silt and basal gravel samples. Inspection by Energy-dispersive x-ray spectroscopy (Bruker Quantax 50 EDS; CSD Facility) showed that some mica flakes from the Condrey Mountain Schist surrounding the lake contain a large amount of carbon (as much as 77%) and scanning-electron-microscopic analysis shows the presence of pyrite. This finding suggests that some of the schist is likely graphitic, which (along with pyrite and reduced manganese) may contribute to the black colour of some of the sediment from deeper water cores.

### 3.7.2 XRF

The downcore XRF (core SQB5) data and results from the analysis of individual sand samples (from locations identified by blue triangles in Figure 3) are shown in Figure 14. Figure 14 (left) shows that, as expected, the raw counts for elemental variables covary downcore with radiodensity and magnetic susceptibility, especially iron, silicon, and potassium. Some deviations from this relationship emerge, however. For example, whereas most elements covary with sediment density, deposit E and deposit I do not have coincident increases in calcium and manganese with radiodensity. Virtually no overlap exists between the samples of sand surrounding the lake and the Slickear Creek bed sand, regardless of whether it was normalized by titanium, strontium, or left as raw counts. This lack of overlap between Ca versus K in the scatterplots of the XRF data from lake margin beach sand and watershed streamed sand (Figure 14, right) is consistent whether data are represented as peak areas (raw counts, top) or normalized by titanium (middle) or strontium (bottom).

Each of the event deposits identified downcore are shown as separate scatterplots in Figure 15. These scatterplot patterns are different in terms of direction in which the deposit evolves (clockwise or counterclockwise), in terms of direction with respect to the axes, and the changes in one variable with respect to the other. Most of the disturbances (deposits B, D, E, F and I) show counterclockwise rotation whereas deposits J and H rotate clockwise. Those with clockwise rotation are also those enriched to some degree in Ca compared to the counterclockwise deposits which are enriched relatively in K. Deposit J is the only deposit that shows complex variability not in the direction of Ca or K, but rather loops between the two.

**3.8 Age-depth model**

We created the age-depth model using the radiocarbon ages (samples 1-4) shown in Table 4, the event-free stratigraphy for SQBss/1/2, and erosion estimates using a P_sequence in OxCal (v 3.4.2; Bronk Ramsey, 2017) as described in the methods section. We use a *k* value of 1 (typical for cm scale sedimentation rate variability to allow changes every 1 cm) and then define a prior for $\log_{10}(k/k_0)$ that allows variation by two orders of magnitude. The goal of this is to allow some flexibility in the age model to account for a variable sedimentation rate. The upper end of the record is constrained based on the sediment inflection representing approximately the time of land use changes in the mid 1800's (Lalande, 1995; including the raising of the lake

level in 1877; see inflection in sediment radiodensity in Figure 5) and using the assumption that deposit B was deposited in 1964. The 1964 horizon was determined based on a comparison of the upper and lower lake records (as shown in the Supplementary Data). The resulting modelled and unmodelled age ranges, and agreements between them, are shown in Table 7a. For comparison, the same information is shown for the same model, but instead of sample 2 a calendar date of 1700 CE is used (Table 7b). The resulting model and estimated age ranges for event deposits A-J are plotted as shown in Figure 16 and

listed in Table 8.

**4 Discussion**

This study seeks to determine if sediments from Lower Acorn Woman Lake, Oregon, contain evidence of Cascadia earthquakes. Each description of event deposits below includes a reference to the age-depth model (Figure 16) and resulting modelled calendar ages for each of the deposits in the sequence. To determine if Cascadia earthquakes disturb sediments in

the lake, we evaluate the timing and characteristics of the disturbances in the sedimentary record, starting with deposit J, which is suspected to have been deposited in response to the 1700 CE Cascadia earthquake. First, however, we summarize types of disturbances and how previous studies have differentiated between them.

**4.1 Possible sources of event beds**

The event deposits in this record could be a result of floods, post-disturbance (wildfire, land-clearing, earthquake) erosion, or

earthquakes (including megathrust, intraplate and crustal earthquakes). Because water is required to carry sediment, any watershed event deposits are included with flood deposits. Flood deposits can be of two basic types based on relative flood water (plus sediment) density: 1) inter- or surface-flow deposits or 2) density current (turbidity or hyperpycnal flow) deposits. Given that turbidites are not a unique identifier of earthquake deposits, we must differentiate between earthquake-triggered turbidites and other types of turbidites.

Turbidites can be of many forms and can result from different types of extreme events, including earthquakes (e.g., Goldfinger et al., 2012; Morey et al., 2013; Howarth et al., 2014; Moernaut et al., 2014; Monecke et al., 2018; and Vandekerkhove et al., 2020) and floods (e.g., Wilhelm et al., 2012; Gilli et al., 2013; Wirth et al., 2013; Vandekerkhove et al., 2020). Earthquake-triggered deposits are typically mass-transport deposits (resulting from subaquatic and subaerial landslides and debris flows)

which form thick turbidites (Moernaut et al., 2014; Simmoneau et al., 2013), although smaller turbidites can result as well (Wilhelm et al., 2016; Moernaut et al., 2017; and Monecke et al. 2018). Flood deposits are of two general types. Whereas interflow deposits result in simple normally graded deposits, floods that last days to weeks frequently produce deposits that may have pulses and reflect the waxing and waning of the flow which form inverse, then normally, graded deposits (Alexander and Mulder, 2002; St-Onge et al., 2004). In contrast, earthquake-triggered turbidites are typically normally graded deposits. Only one study (Van Daele et al., 2019) has successfully discriminated between intraplate and megathrust earthquakes using lake sediments. The authors demonstrate that megathrust earthquakes are more likely to produce subaquatic slope failures as compared to intraslab earthquakes which produced subaerial rock wall failures. These intraslab earthquake deposits are followed by post-seismic removal of sediment from the watershed. The explanation is that intraslab earthquake spectra peak at a higher frequency (>5 Hz) compared to megathrust earthquakes (<5 Hz) and therefore have a greater potential to cause subaerial rockslides. The soft lake sediments, however, are more likely to amplify the low-frequency accelerations from subduction earthquakes resulting in subaquatic slope failures.

### 4.2 Does the Lower Acorn Woman Lake record contain an event deposited in 1700 CE?

Finding an event deposit dated to 1700 CE in the sedimentary record from Lower Acorn Woman Lake would be strong evidence that the lake records Cascadia earthquakes, however, this is difficult to determine because of the challenges presented by the radiocarbon production curve (IntCal20; Reimer et al., 2020). Over the past 300–400 years variations in the radiocarbon production curve result in multiple intersections during radiocarbon calibration for a sample deposited in 1700 CE. The result is that there are multiple calendar ages that cannot be evaluated for their likelihood without additional information. As previously mentioned, the radiocarbon samples are detrital and therefore may be older than the time of deposition (typically by decades to centuries; see, for example, Streig et al., 2020) because they resided in the watershed for an unknown amount of time prior to emplacement, and therefore must be considered maximum limiting ages.

Sample 2, taken from just below deposit J, produced a radiocarbon determination of 110 +/-25, similar to what would be expected for a sample that died around 1700 CE, however calibration of this sample results in three probability peaks. If we assume that the age of the samples represents the stratigraphic order in which they were deposited, there are two options for radiocarbon samples 1-4 with respect to the radiocarbon production curve that are younger than 1950. A third option (youngest calendar age distribution) is discounted because sample J was deposited prior to the inflection in sediment density that reflects a change in land use (logging and road building). Option 1 is the modelled distribution determined by the age-depth model (shown in Figure 17, left), which places sample 2 at 1680-1780 and option 2 is an alternative which places sample 2 between 1800-1940 (Figure 17, right). Constraining the upper portion of the record can be approached by using the sedimentation rates from nearby lakes. Given there are 71 cm of event-free sediment above deposit J and a sedimentation rate of ~ 1-3 cm/10 years (based on nearby Bolan Lake sedimentation rate; Briles et al., 2005, and the Upper Acorn Woman Lake sedimentation rate (known to have a higher sedimentation rate than Lower Acorn Woman Lake, but is used here to as a maximum possible rate); Colombaroli et al., 2018), the range of possible time represented between the time of collection and deposition at the location

of sample 2 is between ~240-710 years. Using the average sedimentation rate from the age-depth model in Figure 16 (~2 cm/10 yrs), the time represented by 71 cm is ~350 years. Because the radiocarbon age for this sample is 1680-1940 (unmodeled), the maximum age for deposit J is 1680 CE, which is ~330 years prior to the time of collection. This places the likely age of the sample between ~240-330 years prior to the date of collection, supporting an interpretation that the older of the calibration peak options for sample 2 (Figure 17, left) is more likely.

Formation of deposit J at around the time of the 1700 CE earthquake is supported by additional information from Upper Acorn Woman Lake. Colombaroli et al. (2018) used CT scans to estimate the proportion of fluvial silt at mm-resolution, then modelled the age-depth time series using the fluvial silt-free sediment depths. Their method identified seven anomalously thick rapidly deposited layers (compared to a frequency-magnitude distribution) that were suspected to have been formed in response to a different process than erosion, possibly earthquakes. One of these thick events correlates to deposit J (see Figure 12). Their age-depth model resulted in an age range of 1718-1758 CE for this deposit, close to the maximum limiting radiocarbon age for deposit J, suggesting that the older part of the radiocarbon distribution is consistent with the sedimentation above the event, and the likelihood of stratigraphic order.

Replacing radiocarbon sample 2 with a calendar age of 1700 CE (blue lines indicating envelope boundaries shown in Figure 16) produces model agreement statistics that support the assumption that deposit J was deposited close to 1700 CE. For the original model with sample 2 results in moderate agreement between the data and model (overall agreement is 64.7%) whereas the model using the 1700 CE date in place of sample 2 the agreement between the data and model are higher suggesting a better fit than obtained by the model with the radiocarbon sample 2 date (Acomb = 0.8 and 0.65 respectively).

## 4.3 Insight into depositional processes and attributions

### 4.3.1 XRF data

Distinctive patterns exist in the raw XRF data through the event deposits (Figures 14 and 15). To investigate these patterns, we represent the relationship between endmember indicators Ca and K (expressed as raw counts, after smoothing with a 9-point gaussian filter) for core SQB5 as a scatterplot (Figure 18). Ca and K are useful elements for provenance tracking because Ca amphiboles (sourced primarily from the Slickear Creek drainage to Lower Acorn Woman Lake from the north) are present in Slickear Creek watershed rocks while K is more prevalent in muscovite (sourced from the Condrey Mountain Schist which provides material from Acorn Woman Creek to Lower Acorn Woman Lake; Table 7). No calcium rich rocks were detected in the Condrey Mountain Schist.

The scatterplot of the raw, smoothed data (Figure 18a) reveals patterns that were not obvious from the downcore representation of the elemental data (compare Figure 18 to Figures 14 and 15). A cartoon (Figure 18b) illustrates that the original interpretation of the patterns in the raw data appear to reflect the relative amounts of each component with deposition (key to colour coding for each disturbance A–J shown in Figure 18c) through deposits. The XRF variables Ca and K expressed as peak area (raw counts), however, are not true endmember concentrations because the data contain artifacts. These artifacts are primarily

related to bulk density (porosity) which affects the amount of material present in the X-ray beam and are different for each sample as a result of the "closed sum effect" (Rollinson, 1993; van der Weijden, 2002). A variety of factors, such as X-ray tube age, surface roughness, dilution by organic matter, and X-ray attenuation differences from variations in water content also contribute to these artifacts (Boyle et al., 2015; Löwemark et al., 2011; Thomson et al., 2006). To account for this, another variable is frequently used to normalize XRF elemental data (i.e., the log ratio method of Weltje et al., 2015) to better quantify the composition of different components, at the expense of potentially adding analytical noise.

We normalized the raw data to silicon (Figure 19a) then plotted potassium versus calcium for comparison to the raw data. The results clearly shows that deposits J and H are enriched in calcium compared to other event deposits. When scaled by radiodensity (Figure 19b), the patterns in the scatterplot look very similar to the scatterplots of the raw data (Figure 18a). Because the raw data are highly reproduceable, and to first order the raw data represent the composition and density of the sediment, the raw data were chosen for this investigation, and are interpreted to approximate endmember composition changes with the implicit third variable, radiodensity. Although using raw XRF data in this way does not remove artifacts, it does allow for the characterization of sediment layers (as was done in Lu et al., 2021). The resulting loops in the scatterplots, then, are interpreted to represent approximate elemental changes with grading through deposits, where the XRF loops begin to deviate from the initial background value at the base of the deposit (identified by the encircled letters with subscript $i$ in Figure 18a). The identification of boundaries between disturbed and undisturbed sediment using this method was essential for creating the event-free stratigraphic sequence for the development of the age-depth model.

Here we demonstrate the usefulness of the XRF scatterplots, along with other information, to infer mechanisms controlling deposition for deposits A-J, then use all data (including age data) to make inferences as to attribution.

Deposit J. Deposit J, described previously as a medium-silt deposit displaying unusual grading characteristics, is well-sorted at the base (becoming less-so upward) and lacks diatoms and particulate and degraded organics in the basal silt. The base of the deposit is sharp, and there appears to be evidence of loading into the organic-rich sediment below. The age-depth model suggests that deposit J may have been formed in response to the 1700 CE Cascadia earthquake.

To gain insight into the processes influencing deposition, we look to the raw XRF geochemical data (Figure 20) as scatterplots of potassium to calcium for deposit J. Note that the visible base and top of the deposit do not start and end at the background ratio (represented by the initial positions of deposits G, H and I). This "gap" suggests that there is more at the base and the top of the deposit than is visible by eye. In other words, the basal silt is preceded by, and followed by, sediment that is part of the event deposit (or from closely space events without interevent sedimentation between them). Sediment below the primary silt unit (blue line, centre top) suggests the preferential reduction in K prior to the more obvious base of the silt which has a preferential initial increase in Ca (representing Slickear Creek watershed-sourced sediment). Microscopic inspection identifies a thin (sub-mm scale) micaceous silt layer followed by organic-rich sediment a few centimetres below the primary silt. Whereas this silt layer is indistinct in the northern cores, it is more obvious in the deeper water cores, especially SQB9. This suggests that the precursor may be a bypass turbidite (fine-grained turbidites that are formed when the coarser sediment bypasses the

location; see Bouma, 2000) that is visibly present in deeper water cores but indistinct in the northern shallower water cores. Alternatively, this may be the result of a separate event that occurred immediately prior to deposit J.

Sediment above the primary silt unit (green line, centre top) represents a long tail that returns to the initial position with respect to the background ratio. This tail is also apparent in the radiodensity (Figure 20, lower panel). The tail is followed by a very low radiodensity layer (a few cm thick) that visually appears to be part of the background sediment; however, the diatoms and other water column organisms are of different species (see smear slides in Figure 20, bottom right). The presence of an organic tail is supported by the loss on ignition data that shows a 30% decrease in inorganic content along the length of the tail.

The XRF data suggests that grading through the dominant silt layer is complex; as grading progresses upward, the XRF pattern changes in way that appears to reflect the partitioning of entrained sediment into components slightly enriched in each elemental endmember: as the deposit grades upward, first in the direction of Ca, looping around (suggesting the influence of another variable), then returning to background slightly depleted in Ca with respect to background sediment. This complexity is unlikely to reflect multiple events through time because values don't ever go back to background. Possible explanations for the characteristics of deposit J are that suspended material remained in the water column while being partitioned as shaking continued then slowed (possibly influenced by an internal wave), or it could be the result of simple differential settling as shaking ceased. Differentiating between these interpretations of the XRF grading pattern require experimentation and further investigation to prove.

The lack of organic matter and diatoms and associated enrichment of calcium in the base of the silt has two possible explanations. The first is that the silt is not the result of a turbidity flow, but instead the result of silt settling from intermediate water depths after shaking partitioned sediment in the water column. This would mean that the silt would have been introduced into the water column from this depth, likely from a delta source. This could happen if liquefaction of the delta matrix forced sediment and water into the lake horizontally. The alternative explanation is that the silt is the result of differential settling of the suspended fraction of a turbidity flow, and that schist-sourced particles remained in the water column (because the platy mica would remain suspended for longer because of its larger surface area) allowing the Slickear Creek sourced silt to settle first. In both cases, the long tail is likely the result of flocculation and rapid settling of fine particles. Although liquefaction of the delta is plausible, it is considered unlikely to be an exclusive explanation for the pure Slickear Creek watershed-sourced silt at the very base because it would suggest that the 1700 CE earthquake would not have produced a turbidite, other than a small bypass turbidite, even though shaking was strong enough to have caused liquefaction of the delta.

Deposits H and I. Deposits H and I form a complex sequence that formed between 1819-1875 CE (based on the age-depth model). We describe them together in order of deposition because they appear to have formed in response to the same event. Deposit I is a turbidite composed of disaggregated schist with visible mica fragments. The XRF scatterplot for deposit I demonstrates it varies almost entirely in the direction of K, and therefore is not sourced from lake margin slope sediment (which would be a mixture with some enrichment in Ca). It displays reversed, then normal grading from a coarse- to medium-grained silt upward to form a short organic tail followed by a thin layer of deciduous leaves (forming the boundary between the schist turbidite and the silt from deposit H above). Deposit I is very similar to deposit E, a local subaerial lake-margin

slope-failure deposit, in that it is a turbidite formed of dark grey schist with large mica flakes. It contrasts with deposit E in that it is found in all cores, suggesting that deposit I was formed because of a disturbance great enough to result in terrestrial landslides around the lake. As a result, this event deposit is interpreted to be a subaerial lake-margin slope failure from an earthquake, likely the 1873 CE Brookings earthquake because of timing and because shaking was strong enough in the region
to topple chimneys.

In contrast, deposit H is composed of Slickear Creek watershed-sourced sediment in core SQB5 (more so than any other deposit based on XRF; Figure 15). The event deposit in SQB2 appears to have a tail which is hummocky with respect to radiodensity instead of smoothly grading upward (Figure 21). SQB9 (from the lake's depocenter) contains a temporal correlative to deposit H, but it is composed of multiple turbidites (Figure 21). The multiple turbidites forming deposit H in
deep-water core SQB9 have several possible explanations: they could be (a) synchronously triggered "amalgamated turbidites" (using the terminology of Van Daele et al., 2017, p. 77-78) from a single earthquake producing multiple individual slope failures that travelled different distances (and therefore different travel times) to reach the lake's depocenter (SQB9) depositing one over the other, (b) the result of reflection waves or a seiche from a single earthquake producing multiple stacked deposits, c) a mainshock and aftershock sequence for a single earthquake; (d) an earthquake with a complex source function, (e) a post-
earthquake retrogressive failure sequence, or (f) "turbidite stacks" (again, using the terminology Van Daele et al., 2017, p. 77-78) suggesting multiple earthquakes occurred in sequence.

The simplest explanation is that the multiple turbidites forming deposit H in core SQB9 are the result of mechanism (a) and as a result are amalgamated turbidites. Although slightly deformed as a result of coring, deposit J appears to have two pulses in core SQB9, suggesting an amalgamated turbidite fed by two channel systems (Figure 21). Deposit H in core SQB9, however,
has more than two turbidite pulses, suggesting that the presence of two channels is not the only explanation for the observed sequence. This, then, could be a form of amalgamated turbidite produced when multiple ruptures trigger turbidity currents very closely in time that travelled along one, or in this case two, channel systems. This suggests the turbidite sequence represented by deposit H is similar to an earthquake doublet (e.g., Lay and Kanamori, 1980), the result of closely spaced (within a few hours' time) ruptures on crustal fault segments. This hypothesis could be tested using the methods described in Wils et al.
(2021). The interpretation of the 1873 CE Brookings earthquake as a rupture sequence suggests the earthquake was a crustal earthquake and not an intraplate earthquake, which would present a significant, and previously unknown, earthquake hazard in this area.

There is little evidence supporting the following alternative hypotheses. Mechanism (c), explaining the deposit characteristics as the result of a mainshock and aftershock sequence seems unlikely because the pulses do not get smaller upward as would
be expected if a combination of main and aftershocks and there is no background sediment between pulses. This explanation is also unlikely because the deposits are less likely to have been deposited over time (weeks or longer) because the XRF scatterplot through deposit H in core SQB5 is a continuous loop (which does not go back to background until the end; Figure 22). A single event with aftershocks can sometimes occur immediately after the mainshock and continue for hours (Toda and Stein, 2018); however, there were few aftershocks felt (Wong, 2005) in response to the 1873 CE Brookings earthquake

(although the area was sparsely populated by settlers at the time). A seiche could cause multiple delta failures, however, it seems likely that the wave energy would result in suspended sediment composed of a mixture of sediment sources instead of the Slickear Creek watershed-sourced sediment observed in deposit H at site SQB5. This suggests that mechanism (b) is unlikely. There is no evidence to support mechanism (f), explaining the deposit as the result of multiple earthquakes at this location (based on newspaper accounts, T. Brocher, personal communication, April 2024). Mechanism (e), explaining the

multiple turbidites in SQB9 as post-earthquake retrogressive subaerial failures, is also unlikely because the turbidites are not composed of schist-derived sediment (like deposit I) in both deep and shallow cores.

There is evidence for mechanism (d), which explains the observed sequence as a result of a complex source function. Crustal fault earthquakes, like intraplate earthquakes, produce higher frequency ground motions as compared to subduction earthquakes and therefore are more likely to cause subaerial failures. In contrast, the longer duration and lower frequency

content of a subduction earthquake are more likely to cause failures of the soft sediments of the Slickear Creek delta which are dominated by Ca-rich sediment from the Slickear Creek watershed. While crustal and intraplate earthquakes would also cause delta failures, they would be less voluminous as compared to delta failures from a subduction earthquake. In this case, the 1700 CE Cascadia earthquake, represented by deposit J, caused a smaller delta failure as compared to the earthquake that caused deposit H. This suggests that there may have been a subduction component to the 1873 CE Brookings earthquake.

Other explanations for this observation are that there may have been conditions in 1700 CE compared to 1873 CE that limited the amount of sediment available to be disturbed (such as drought, which is suggested by the sedimentation rate at ~1700 CE), the influence of rupture geometry, or the influence of specific site or seismic source parameters that modify the generalized relationship between earthquake type and frequency content.

A complex source is supported by evidence from newspaper reports of the 1873 CE earthquake. The pattern of ground motions

in the region influenced by the earthquake is highly variable and does not simply dissipate with distance from the inferred epicenter (see Supplementary Data). There was also a report of a small local tsunami wave just north of Cape Blanco, Oregon (Crescent City Courier, 29 November 1873): *"about ten minutes after the shake he heard a noise off to the westward loud as a report of a hundred cannon, and that he noticed indications on the beach of very high water mosses and sand being thrown up to the highest water marks."* Although far from conclusive, this supports the possibility of a subduction component to the

earthquake.

The preferred interpretation is that the 1873 CE Brookings earthquake was a complex crustal fault rupture sequence, possibly with a subduction component. This suggests the 1873 CE earthquake may have been similar to the 2016 Kaikoura earthquake, which was a synchronous rupture on the megathrust and a set of crustal faults above the rupture in the overlying plate (Furlong and Herman, 2017). Although the Kaikoura earthquake is considered unusual in its complexity, several other fault systems,

including southern Cascadia, have similar conditions required for this type of simultaneous rupture (Herman et al., 2023).

Deposit G. Deposit G, based on the age-depth model, settled between 1827-1892 CE. It is visibly indistinct in SQB2 but slightly denser than background sediment, with maximum radiodensity at the deposit centre. The age-depth model suggests that deposit G could be the result of the large flood in 1890 (#2 of 3 based on stream gage data), a flood and dam failure in

1881 CE, the 1873 CE Brookings earthquake, or the flood of 1861-62 CE. The 1873 CE Brookings earthquake caused strong shaking in the region (Wong, 2005) and therefore is more likely to have caused large event deposits H and I (see discussion above) which occurred prior to deposit G. Because the size of the 1890 CE flood was larger than the rain-on-snow event that caused the dam to fail in 1881, this deposit is most likely the result of the 1890 CE flood. This interpretation of deposit G is supported by a comparison of shallow and deep-water sites. Figure 21 shows that deposit G in SQB9 is a set of stacked flood deposits that accumulates through time because XRF indicates the deposit remains at background levels through the deposit (with a slight enrichment in K in the centre of the deposit; Figure 15). The proximity of the event deposit to the deposits H and I, and the likelihood of subaerial landslides from the 1873 CE Brookings earthquake, suggests it is plausible that this deposit was created when flooding removed post-seismic sediment from both watersheds (because the sediment is of mixed composition) during the flood of 1890 CE.

Deposit F. Deposit F is a simple graded deposit which the age-depth model suggests settled between 1835-1908 CE. This deposit shows evidence of loading onto the organic sediment below in SQB2. The XRD mineralogy is unknown, however the XRF data suggests a mixed composition enriched in K that does not vary in composition with changes in radiodensity at the midpoint of the deposit. Lower Acorn Woman Lake was influenced by a sequence of closely spaced events that began with the 1861-62 event. These include the 1861-62 CE atmospheric river flood event, the 1873 CE Brookings earthquake, the installation of the dam in 1877 CE, the winter rain-on-snow event in 1881 CE which caused the dam to fail, a large flood in 1890 (#2 of 3 based on stream gage data), a smaller flood in 1892 CE, and the 1906 CE San Andreas earthquake. Given that this appears to be an interflow deposit (because of its sharp base and upward fining) it is likely not the result of the large flood of 1890 CE, therefore it is suspected to be the result of the flood of 1892 CE. The source of this event deposit, however, remains uncertain.

Deposit E. This deposit is only found in core SQB5, which is located on a steep (~45°) slope. There is no age data for this deposit because it cannot definitively be identified in the chronology core SQBss/1/2 composite, however we know that the time of deposition is between deposits D and F. Deposit E is a subaerially-sourced turbidite predominantly composed of Condrey Mountain Schist (based on XRD and the dark grey colour of the deposit with visible mica flakes). The XRF data shows that changes through the deposit goes primarily in the direction of K, implying a relative increase in schist-derived sediment. Slope failures are common at the location of SQB5, indicated by the large amount of sediment missing between deposits B and J in the short cores identified as narrow and wide diameter short cores (see false colour image of core density at the top left of Figure 11). This deposit was likely the result of a local terrestrial lake-margin wall failure because it is found in only one core, and the schist-derived composition suggests a lake-margin bedrock source for the sediment (not a mixed source as would be expected from the disturbance of subaquatic surficial sediment). An aseismic local wall-failure deposit could have resulted from heavy winter rains (like the deep-seated slope failure that occurred in response to heavy rains during the winter of 2016 (which post-dates field work). Alternatively, it is possible that deposit E is a local wall failure that resulted from an earthquake, however it is considered unlikely because an earthquake is more likely to disturb sediment at more than just one location.

Deposit D. The age-depth model suggests that deposit D settled between 1870-1940 CE. Deposit D is unusual in several ways. It is a sequence of two silt units (the lower-most is thickest and is visually obvious) within a stiff layer of organic-rich sediment. This deposit displays a small counterclockwise loop in XRF (Figure 15) and the lower silt unit is a simple graded unit. This basal silt is orange in colour and is fine grained and well sorted. The XRF and XRD data suggest that although the majority of the deposit is of mixed composition, there is some enrichment of Slickear Creek watershed-sourced sediment at the deposit base.

The age-depth model suggests that deposit D may be the result of a flood event or the 1906 CE San Andreas earthquake. Given that there are 58 cm (event-free) over the past 126 years (based on the location of the inflection in sediment density), and there are 14 cm between this inflection and deposit D, this makes the age of deposit D: the date of inflection (~1850 to ~1880) + 30 years = 1880 to 1910 CE. This is very close to the time of the 1906 CE San Andreas earthquake or the 1890 flood. The presence of an event deposit from the ~M7.9 1906 CE San Andreas earthquake is considered plausible because felt reports from the region suggest minimum MMI values of IV in this region (Dengler, 2008; close to the limit of V-VI for delta failures suggested by Van Daele et al., 2019). Directivity of the 1906 CE San Andreas earthquake was from south to north (toward Acorn Woman Lakes), and seismic waves travelling in the direction of the rupture can have higher frequencies and amplitudes compared to those traveling perpendicular to the rupture direction (Daxer et al., 2024). Attribution to the 1906 San Andreas earthquake is further supported because the earthquake occurred along two main slip patches (Song et al., 2008) and deposit D contains two pulses of silt in sediments from Lower Acorn Woman Lake.

Could deposit D be the result of an interflow flood deposit containing Slickear Creek watershed-sourced sediment? Any flood would transport a mixture of sediment sources into the lake because there are two creek systems that feed the lake. To produce a flood deposit that is enriched in Slickear Creek watershed-sourced sediment would require a very localized rainfall event only in the Slickear Creek watershed. Although the orange colour of the Slickear Creek watershed-sourced portion of the deposit indicates oxide formation on the grains (suggesting that the sediment resided in a subaqueous, but oxygenated environment), this could either happen in a stream system or the delta. Because a small, localized flood (affecting the Slickear Creek system only) is unlikely, the preferred interpretation is that deposit D was deposited in response to the 1906 CE San Andreas earthquake.

Deposit C. The age-depth model suggests deposit C settled between 1880-1950 CE. Deposit C is a normally graded unit with an unknown composition that becomes thinner with distance from Acorn Woman Creek (see below deposits A and B in Figure 5) near the dam, suggesting it may be the result of a flood. XRF data does not exist because deposit C is not in core SQB5 (it was eroded out of this core by the event producing deposit B). The most likely events to produce this deposit (based on the size of the event) are the third largest flood (which occurred in 1955) or the flood in 1927 associated with a debris dam failure. Given that the average sedimentation rate is ~2.5 yr/cm and the interevent sediment thickness between Deposits B and C is 18 cm (after accounting for erosion; top left Figure 8), it is suggested that deposit C is the result of the 1927 CE flood with debris dam failure. This interpretation is supported by comparison to previous work that suggests that dam failures result in coarser and thicker deposits towards the dam (e.g., the 1929 CE dam collapse in Eklutna Lake; Boes et al., 2018), however this may

be confounded by other influences that change with distance from the dam (such as water depth). No sediment provenance data exists for this deposit; however, in core SQB2, this deposit is brown in colour, similar to the lower portions of deposits A and B. Because deposit C was deposited well after the inflection point in radiodensity assumed to be the result of land use changes at ~1850, the possibility that it is the result of the 1861-62 CE flood event is considered very unlikely. The simplest explanation is that deposit C was deposited as a result of the 1927 CE flood and dam failure.

Deposits A and B. Deposits A (deposited between 1980-2013 CE) and B (attributed to the 1964 CE flood based on comparison to Upper Acorn Woman Lake) are 5-20 cm thick, depending on location in the lake, with lake-wide extent and similar characteristics (Figure 6). They have sharp bases with sediment likely missing below in all cores other than the deepest water cores (SQB9 and SQB10), contain basal sediment with rootlets and degraded organic matter, and are coarse-grained, normally graded deposits. These characteristics suggest they are the result of erosive turbidity currents. Although the deposits are similar to one another, the base of deposit A (which is incomplete in core SQB5) has a thin layer of calcium-rich coarse silt at the base, whereas the base of deposit B is composed of potassium-rich coarse micaceous silt.

Deposits A and B were most likely deposited in response to large flood events because the most recent historical events are the two largest flood events that occurred in 1997 CE and 1964 CE. Multiple first-hand reports describe the nature of the extreme flood of 1997 CE in the vicinity of Lower Acorn Woman Lake: A landowner described the flood as having transported watershed-sourced beach sand from one end of the lake to the other (B. Harr; June 2015), and U. S. Forest Service employees (personal communication, P. Jones; December 2019; J. McKelligott, December 2020) described debris caught at the dam that caused the lake level to rise a few feet above the maximum water level. Water was seen shooting 10 feet out of the spillway and caused damage to the gate. At Applegate Reservoir, a few kilometres downstream from Acorn Woman Lakes, water was flowing over the earthen dam and observed to undercut surficial slope sediment, causing slumping into the reservoir (P. Jones, personal communication, December 2019). The extreme nature of this flood, relative timing compared to the 1964 flood and observations of beach sand suggest that the 1997 flood produced deposit A, the uppermost event deposit in the record. There were no other disturbance events around this time. Because there are no other event deposits with similar characteristics downcore, flood events are either more extreme than in the past, or the supply of readily mobilized sediment has increased (which is likely given logging contributions to sediment), or both. It is also possible that the built dam is more likely to trap debris and elevate the lake level in response to extreme flooding than the natural landslide dam. We conclude that deposit A resulted from the 1997 CE flood and deposit B resulted from the 1964 flood.

A summary of the deposit characteristics described in the preceding sections and their attributions to closest temporal historical events based on the age-depth model are shown in Table 9.

## 4.4 Two other events of note

There are two event deposits that were not initially identified in the Lower Acorn Woman Lake record that allow us to check our use of XRF to identify disturbances and differentiate between deposit types. The first is located between deposits I and J at 40 cm in depth in core SQB5 (Figure 5), and the second is below that, located in what appears to be the tail of deposit J (not

visible by eye in the physical property data, but located at ~43 cm core SQB5, Figure 5). To determine the origins of these deposits, we turn to the XRF data.

     The deposit located between deposits I and J rises then falls in radiodensity and is followed by denser sediment that could be a tail. The XRF data, however, show that there is no sequential grading pattern to the deposit (black trace, Figure 23). The interpretation is that this must be the result of flood pulses. One of the largest atmospheric river events in recorded history in

the west occurred in 1861-62 CE and lasted for 40 days (Engstrom, 1996). The stratigraphic position of this deposit supports this interpretation as having occurred just prior to the 1873 CE Brookings earthquake. If correct in that this deposit is the result of the 1861-62 CE atmospheric river event, this would explain the denser sediment between this deposit and deposit I as a reflection of changes in land use around 1850 CE and does not represent a deposit tail.

     In contrast, when evaluating the tail of event deposit J using XRF, a disturbance appears (shown by the red trace, Figure 23).

This sequence is very similar, but at a much smaller scale, to that of deposits H and I and is located at ~43 cm in core SQB5 (Figure 5). This suggests that another earthquake occurred soon after the 1700 CE earthquake (before the tail completely settled which takes days to weeks, e.g., Van Daele et al., 2017; Wils et al., 2021). The southern Cascadia marine paleoseismic record also contains a Cascadia triggered turbidite with a likely northern San Andreas turbidite in the tail (Goldfinger, 2021), suggesting a San Andreas earthquake origin for this deposit as well. This seems plausible given that there is evidence that the

1906 CE San Andreas earthquake is present in Lower Acorn Woman Lake (represented by deposit D).

     These examples demonstrate the usefulness of the XRF scatterplots to identify deposit origins.

## 4.5 Summary of interpretations

     Flood deposits are composed of mixed composition with some enrichment in K as compared to background, reflecting the multiple watershed sources of these deposits. Enrichment in K appears to be associated with high water and slumping of

sediment into the lake when undercut by fast flowing water. Three types of flood deposits were described. The first type is represented by deposits A and B. They are thick turbidites with high magnetic susceptibility and radiodensity, but organic matter entrained in the base results in a brown colour and lower magnetic susceptibility. The deposit bases show evidence of erosive flow. These deposits are interpreted to be the result of fast-flowing, high water from the large floods of 1997 CE and 1964 CE. These floods may be influenced by land use changes (increasing runoff) because similar deposits are not found

further downcore (see companion paper Morey et al., 2024, this volume). The second type of flood deposit is interpreted to be an interflow deposit which is a simple graded silt unit exemplified by deposit F). The third type of flood deposit is represented by deposit G and the unlabelled deposit between deposit I and J. These deposits display reverse then normal grading; this type of deposit is interpreted to be the result of pulsed floods lasting days to months and may reflect atmospheric river rain events. Because deposit G is attributed to a flood that occurred after an earthquake that caused subaerial landslides, it likely removed

post-seismic terrestrial landslide sediment into the lake.

     Earthquake deposits are enriched to various degrees in Ca, a result of liquefaction of the delta, differential settling of the suspended fraction, or both. Three types of earthquake deposits are suggested by the data. The first type is represented by

deposit J. Deposit J is a turbidite slightly enriched in Slickear Creek watershed-sourced sediment, primarily in the lower portion of the deposit. The base of the silt appears to be preceded by a bypass turbidite. The deposit displays loading or coseismic insertion of silt into organic sediment at the base of the primary silt unit. This silt is followed by an organic-rich tail that shows evidence of extensive sediment partitioning of the suspended sediment during settling. The tail is followed by a change in the organisms in the water column. The second type of earthquake deposit is the sequence represented by deposits H and I, which is a schist-derived subaerially sourced turbidite (deposit H) followed by silt enriched in Slickear Creek watershed-sourced sediment from multiple delta slope failures (deposit I). Deposits H and I are interpreted to have formed in response to the 1873 CE Brookings earthquake, which is suggested to be the result of a crustal earthquake rupture sequence, possibly with a subduction component. A third possible type of earthquake deposit is represented by deposit D. It is a multipulsed graded deposit slightly enriched in Ca followed by silt of a mixed composition. Although a seismic source is uncertain, the interpretation is that deposit D is the result of the 1906 San Andreas earthquake.

## 5 Conclusions

The setting at Lower Acorn Woman Lake, Oregon, (~180 km inland of the deformation front in Cascadia) provided a unique opportunity to determine if it is possible to differentiate between event deposits resulting from floods and different types of earthquakes. The different rock types (K-rich schist and Ca-rich amphiboles) in the two watersheds (Acorn Woman Creek and Slickear Creek, respectively) provided the opportunity to evaluate the source of sediments and how deposits evolve during deposition using XRF, which provided clues to the mechanisms forming them. Whereas flood deposits are more likely to be composed of a mixture of sediment sources (as a result of being sourced from both watersheds), earthquake deposits are more likely to be enriched in Ca, indicating a Slickear Creek watershed source (suggesting processes that partition suspended sediment (shaking), introduce Slickear Creek sediment from the delta (liquefaction), or both). The following disturbance types were identified:

Flood deposits. Flood deposits have a mixed source composition with some enrichment in schist-derived sediment, no enrichment in Slickear Creek watershed-sourced sediment (except for a thin layer of coarse silt at the base of deposit A, attributed to the observed rapid transport of Slickear Creek delta sediment from one end of the lake to the other during the 1997 CE flood), and contain terrestrial organic matter. The deposits do not exhibit complex grading. They are highly variable and of three basic types:

- Thick turbidites like deposits A and B, with erosive bases containing organic matter and a thin silty-clay cap, are interpreted to be the result of fast-flowing, high water that cuts into the lake margin sediment causing it to fail. Deposit A is interpreted to be the result of the 1997 CE flood, and deposit B is interpreted to be the result of the 1964 CE flood.

- Simple graded deposits, such as deposit F, is of mixed composition. It shows increasing thickness toward Acorn Woman Creek. It is interpreted to be the result of an interflow flood, possibly as a result of a flood in 1892 CE.

-    Deposit G and an unlabelled deposit below deposit I are very similar in that they have a mixed composition, showing an increase, then decrease in radiodensity through them and are correlated to pulsed deposits in the deep water cores. The composition remains mixed throughout the deposits. Both deposits are interpreted to be the result of sustained flooding. The flood causing deposit G likely removed postseismic sediment from the watersheds into the lake. The unlabelled deposit is interpreted to be the result of the 1861-62 CE atmospheric river event and deposit G is interpreted
to be the result of a flood in 1890 CE.

Earthquake deposits. Earthquake deposits identified in this study are enriched in Slickear Creek watershed-sourced sediment and show complexity in deposit composition, structure, and grading. The earthquake deposits are as follows:

     -    Deposit J, interpreted to be the result of the 1700 CE Cascadia earthquake. Deposit J is a turbidite showing unusual grading characteristics. The base of the silt is slightly enriched in Slickear Creek delta-sourced sediment and is very
well-sorted and pure (lacking organics) at the base. The deposit then grades upward into a long, organic-rich tail which changes in composition (based on XRF) in distinctive ways compared to all other deposits in the sediment core, suggesting extensive partitioning during shaking. This suggests sustained ground motion. Deposit J has two pulses in cores from the lake's depocenter and one elsewhere, suggesting the lake has two channel systems.

     -    Deposits H and I, interpreted to be the result of an earthquake rupture sequence, possibly with a subduction
component. Deposit I is a lake-wide turbidite composed of almost entirely of terrestrial schist-derived sediment and as such, is enriched in K and not Ca. This is followed by deposit H which is composed of Slickear Creek watershed sediment and is a multipulsed turbidite. This suggests an earthquake rupture sequence because it has more than two pulses in the depocenter cores than channel systems (suggesting earthquakes closely-spaced – minutes to hours - in time). An alternative explanation for these multiple pulses is that they are the result of a seiche, however it would be
expected that wave action would result in a mixture of sediment instead of the Slickear Creek watershed-sourced sediment observed. The interpretation that the 1873 CE earthquake was a rupture sequence suggests it was a crustal fault earthquake, not an intraplate earthquake. The concentration of calcium-rich sediment in deposit H is higher than observed for deposit J attributed to the 1700 CE Cascadia earthquake suggesting a possible subduction component. This deposit sequence is followed by deposit G, which is suggested to represent a flood that removed post-seismic
sediment from the watersheds.

     -    Deposit D is a two-pulsed deposit enriched in Slickear watershed-sourced sediment at the base. The attribution is uncertain, however, because the orange colour of the sediment suggests that the sediment grains are coated in iron oxide, which would happen if sediments were exposed to water and air (such as in a stream). This requires further investigation, but this deposit could be the result of a very localized flood (which would be unusual), or timing (1870-
1940 CE), structure (two-pulses), and composition (basal sediment is slightly enriched in Slickear Creek watershed-sourced sediment) suggests it could be the result of the 1906 CE San Andreas earthquake (more likely).

We conclude that Cascadia earthquakes disturb sediments at Lower Acorn Woman Lake, and that it is possible to distinguish between plate boundary and other earthquakes, and identify flood deposits, using the sedimentological characteristics and

provenance data in sediments from Lower Acorn Woman Lake. This study relied on cores from a proximal setting (near the Slickear Creek delta) because of distorted upper sections in the deeper water cores. Although depocenter cores were not the primary source of data used in this study, we find that comparing these proximal cores to depocenter cores to be essential for the interpretation of deposit characteristics and inferring depositional processes influencing lake sediments. We find that XRF is a particularly useful tool because it is a sensitive, high-resolution indicator of provenance and deposit evolution that has allowed for the identification of cryptic features, deposit boundaries, and complex grading in the sediment cores, and aids in interpretation of mechanisms and processes controlling deposit formation. Although the XRF scatterplot approach to differentiating between deposit types for this study has been very useful, the successful application may be site-specific, and methods used in this study may not be applicable in all settings. The results support the use of turbidites enriched in light-coloured Slickear Creek watershed-sourced sediment as a reliable indicator of earthquakes when interpreting the downcore record.

These results suggest small lakes in Cascadia hold promise to improve our understanding of subduction and crustal earthquakes in the Cascadia forearc where there is a greater potential to impact people and infrastructure.

**Author Contributions**

Ann E. Morey conceptualized, designed, and administered the project, supervised field operations during all 3 field seasons, acquired funding, managed field operations, acquired the data, developed the methodologies, analysed and interpreted the data (including the use of all software for interpretation and modelling), and wrote the manuscript at all stages. Mark D. Shapley helped with core acquisition (2015 field season) and curation, data acquisition, and participated in discussions of the data. Daniel G. Gavin provided field gear and coring assistance for the 2013 and 2014 field seasons, and participated in discussions of the data, especially the comparisons to his previous work at Upper Acorn Woman Lake. ARN participated in the 2013 and 2014 field seasons, provided expert guidance on radiocarbon sampling, funds for radiocarbon determinations, and helped acquire bathymetric data. CG participated in early discussions of project development and contributed funds for a radiocarbon age. MDS, DGG, ARN, and CG also provided input on the manuscript draft. The authors declare they have no conflicts of interest. This manuscript was significantly improved based on insightful comments by Maarten Van Daele and Shmuel Marco.

**Acknowledgements**

This research was partially supported by the National Earthquake Hazards Reduction Program of the U.S. Geological Survey (through a grant to Andrew Meigs and Simon Engelhart, USGS Grant G17AP00028). The USGS Earthquake Hazards Program through Alan R. Nelson (USGS, Golden) partially supported the collection of some Livingstone cores and CT scans of Livingstone and Kullenberg cores. Geological Society of America awards provided additional funding: a graduate student

research grant and the Kerry Kelts Limnogeology Award. Coring in 2015 would not have been possible without contributions from Joseph Stoner, Roy Haggerty, and a donation from Ruth Morey.

The US Forest Service granted permission for this study (special thanks go to Starr Ranger Station employees and the regional office in Grants Pass, OR). We are extremely grateful for the assistance and knowledge provided by Ranger John McKelligott, and for field assistance by Mark Anthony (USFS employee who participated in coring in 2015). Thomas Brocher provided newspaper reports that reported shaking impacts from the 1873 Brookings earthquake. Bert Harr generously allowed access to his Slickear Creek property during this investigation and contributed significantly to this project through his vast knowledge

of local extreme events and local and regional history since 1900. Peter Jones provided personal historical accounts of the more recent historical floods.

This project could not have happened without the generous assistance from numerous volunteers. Katie Alexander (Western Washington University) spent a few days canoeing in the cold to acquire bathymetric data. Maureen Walczak (OSU) generously analysed my first radiocarbon samples and provided guidance on how to use the radiocarbon production curve to

select samples. Jamie Howarth (then at GNS, NZ) provided useful coring information and guidance, including sharing his approach to dating an earthquake event in lake sediments from about the same time as the 1700 CE Cascadia earthquake. Other volunteer field assistants included Randy Keller, Brendan Reilly, Katie Alexander, and many others. Christy Briles (Colorado University, Denver) helped train me in the fine art of lake coring during a fateful summer week in 2010.

CSD and the University of Minnesota, provided Kullenberg coring equipment and expertise. Thanks also go to the OSU core

repository (especially Maziet Cheseby) for housing cores and providing the tools to process them. Carol Chin aided core processing of the Kullenberg cores, for which we are extremely grateful. We would like to thank Anders Carlson for allowing me to be one of the first users of the Itrax. Finally, AEM would like to thank her Ph.D. committee and other key advisors, including Joseph Stoner, Eric Kirby, Alan C. Mix, Jason R. Patton, Maarten Van Daele, Lonnie Leithold, Karl Wegmann, Joseph D. Ortiz, and Nicklas G. Pisias for discussions and guidance.

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

Table 1. Historical events with the potential to disturb the sediments of Lower Acorn Woman Lake.

| Code | Event | Description | Date (CE) |
|------|-------|-------------|-----------|
| E1 | Flood | #5 of 5 largest historical floods | 2006 |
| E2 | Flood[1,2,4] | #2 of 5 largest historical floods | 1997 |
| E3 | Local summer storm[2] | | 1980s |
| E4 | Flood[2] | | Late 1970s |
| E5 | Flood | #4 of 5 largest historical floods | 1974 |
| E6 | Lake drained to pre-dam level | All 17' | 1972 |
| E7 | Flood[2] | #1 of 5 largest historical floods | 1965 |
| E8 | Flood | #3 of 5 largest historical floods | 1955-6 |
| E9 | Flood; *debris dam failure* | #3 of 3; stream gage | 1927 |
| E10 | San Andreas EQ | M7.9 | 1906 |
| E11 | Flood | | 1892 |
| E12 | Flood | #2 of 3; stream gage | 1890 |
| E13 | Dam failure[3] | Flood (winter rain-on-snow) | 1881 |
| E14 | Dam installed[3] | Raised the lake ~5 m | 1877 |
| E15 | Brookings/Crescent City EQ | ~M7.0 Intraplate EQ | 1873 |
| E16 | Flood | ArKstorm; #1 of 3; stream gage | 1861-2 |
| E17 | Cascadia EQ | ~M9.0 subduction EQ | 1700 |

[1]An observer described water shooting out 10 feet past the dam, [2]A local landowner described a thick layer of coarse sediment deposited over the entire Slickear Creek delta as looking like a "moonscape," [3]reported in the *Jacksonville Times* newspaper, and [4]U.S. Forest Service personnel observed and removed logs that blocked the overflow at the dam, elevating the lake level by 3-4 feet. This flood caused the Applegate dam, located downstream from Lower Acorn Woman Lake, to overflow and begin to erode the sediment on the sides of the dam (P. Jones; pers. communication, December 2019).

Table 2. Sediment core locations, water depths, and lengths.

| Core name | Type | Length (*m*) | Water depth (*m*) | Latitude (°) | Longitude (°) |
|---|---|---|---|---|---|
| **SQB-ss** | **Surface core** | **0.80** | **16.9** | **42.04405** | **-123.01853** |
| **SQB1** | **Livingstone** | **6.74** | **16.9** | **42.04405** | **-123.01853** |
| **SQB2** | **Livingstone** | **7.37** | **16.5** | **42.04405** | **-123.01853** |
| **SQB5** | **Livingstone** | **3.98** | **23.5** | **42.04264** | **-123.01909** |
| SQB6 | Livingstone | 5.51 | 10.5 | 42. 04336 | -123.01732 |
| SQB8 | Kullenberg/Gravity | 8.01 | 30.0 | 42.04227 | -123.01908 |
| SQB9 | Kullenberg/Gravity | 8.29 | 37.0 | 42.03982 | -123.02050 |
| SQB10 | Kullenberg/Gravity | 10.08 | 35.0 | 42.03857 | -123.02108 |
| SQB11 | Kullenberg/Gravity | 7.55 | 29.2 | 42.03778 | -123.02175 |
| SQB12 | Kullenberg/Gravity | 5.24 | ~20.0 | 42.04191 | -123.01864 |
| SQB13 | Kullenberg/Gravity | 6.24 | 25.0 | 42.02056 | -123.02056 |
| **SQB14** | **Kullenberg/Gravity** | **8.28** | **30.0** | **42.04356** | **-123.01836** |
| SQB15 | Kullenberg/Gravity | 4.55 | 28.5 | 42.04197 | -123.01945 |

Cores highlighted in bold text are those identified by orange circles in Figure 5. *Cores SQB4 and SQB7 are not included in this list because they are less complete due to partial recovery compared to cores SQB5 and SQB6 (from the same locations). Note: SQB-ss is a surface sample (push core). Kullenberg cores are mildly to moderately disturbed at the top because the coring tubes collapsed some during coring. Sediments in the deeper water cores contained methane; when cutting coring tubes into sections, some sediment was extruded and captured in small subsections.

Table 3. Background sediment characteristics.

| | | Wet density $g/cm^3$ | Dry density $g/cm^3$ | %Water | % Organic | % CaCO$_3$ | % Inorganic | Mag susc (SI) | CT (HU) |
|---|---|---|---|---|---|---|---|---|---|
| After inflection | Ave | 1.13 | 0.34 | 70.01 | 14.48 | 10.74 | 74.78 | $-2 \times 10^{-5}$ | ~400 |
| ($n = 13$) | SD | 0.02 | 0.04 | 3.34 | 1.29 | 1.96 | 1.89 | | |
| Before inflection | Ave | 1.05 | 0.22 | 79.42 | 22.33 | 17.20 | 60.47 | $-10 \times 10^{-5}$ | ~200 |
| ($n = 9$) | SD | 0.02 | 0.03 | 0.83 | 3.23 | 2.55 | 3.80 | | |

Mag susc = magnetic susceptibility; CT = computed tomography density. Diatom tests were not removed from samples prior to combustion; therefore inorganic:organic data include a small influence from the remaining silica from diatoms in the percentage-inorganic data (estimated to be less than 6%). CT density is expressed in Hounsfield Units (HU). Note: Organic, inorganic and CaCO$_3$ percentages were calculated from dry weights. Percent inorganic data does not include percent CaCO$_3$.

Table 4. Age control data. The sample in gray text (Sample 0) was not included in the age model because it is much older than the others in the sequence. Samples 1–4 were used to create the age-depth model, and the other ages were used to align the sections shown in Figure 6. *S-ANU = Australian National University AMS lab, UCIAMS = University of Southern California – Irvine AMS lab, NOSAMS = National Ocean Sciences Accelerator Mass Spectrometry Facility.*

| Sample #, section ID and depth in section | Depth, cm (composite) | Depth, cm (event free) | Description | Laboratory and sample no. | $^{14}$C yrs BP |
|---|---|---|---|---|---|
| 0 SQB1A; 14.0-14.5 cm | 85-85.5 | 64 | Fir needle | S-ANU 42418 | 865+/-35 |
| 1 SQB1A; 15.5-16.0 cm | 86-86.5 | 65 | Fir cone frag | S-ANU 42419 | 255+/-25 |
| 2 SQB1A; 25.5-26.0 cm | 96.5-97 | 71 | Fir needle | S-ANU 42618 | 110+/-25 |
| 3 SQB1A; 35.5-36.0 cm | 106.5-107 | 81 | Fir needle | S-ANU 42617 | 190+/-25 |
| 4 SQB1A; 84.0-85.0 cm | 155-156 | 101 | Fir needle | S-ANU 42616 | 260+/-40 |
| 5 SQB1A; 95.0-96.0 cm | 166-167 | 115 | decid. plant frags | S-ANU 42417 | 630+/-25 |
| 6 SQB1B; 67.0-68.0 cm | 254-255 | 185 | plant frags | UCIAMS 140214 | 1155+/-20 |
| 7 SQB5C; 27-28 cm | 263-264 | 194 | Cone bract | NOSAMS | 1270+/-20 |

Table 5. Splice data for SQB1/2/ss composite as shown in Figure 6. Section SQB1-A is the only section in this core that was used to create the composite core. SQB2A 101 cm is at the same stratigraphic location as SQB1-A 56 cm, as shown in Figure 6 (which is a graphical representation of the relationships between sections presented in the SQB1/2/ss splice represented by this table). Note that SQB-ss is expanded relative to cores SQB1 and 2 and was compressed to match stratigraphy as shown in Figure 12a.

| [1]Composite depth (cm) | [2]Composite depth (cm) | Core section | Depth in section (*cm*) | Core section | Depth in section (*cm*) |
|---|---|---|---|---|---|
| 0 | 0 | | | SQB-ss | 0 |
| 33 | 26 | SQB2-A | 2 | SQBss | 33 |
| 103 | 97 | SQB2-A | 72 | SQB1-A | 24 |
| 135 | 129 | SQB2-B | 4 | SQB1-A | 56 |

[1]Composite depth (cm) without adjusting the length of section SQB-ss; [2]Composite depth (cm) after compressing section SQB2-ss to match the stratigraphy of SQB2-A.

Table 6. XRD mineralogy.

| | Classification | Formula |
|---|---|---|
| **Lake bedrock:** | | |
| Clinochlore – 1MIIb, ferroan | Chlorite grp | $Mg_5Al(AlSi_3O_{10})(OH)_8$ |
| Quartz, syn | Silicate | $SiO_2$ |
| Chlorite-serpentine | (greenschist) | $(Mg,Fe)_3(Si,Al)_4O_{10}(OH)_2 \cdot (Mg,Fe)_3(OH)_6$ |
| Muscovite-2M1, 3T | phyllosilicate | $KAl_2(AlSi_3O_{10})(F,OH)_2,$ $(KF)_2(Al_2O_3)_3(SiO_2)_6(H_2O)$ |
| **Watershed bedrock:** | | |
| Clinochlore – 1Mllb, ferroan | Chlorite group | $Mg_5Al(AlSi_3O_{10})(OH)_8$ |
| Quartz, syn | Silicate | $SiO_2$ |
| Ferro-actinolite | Fe-rich amphibole | $Ca_2(Mg_{2.5-0.0}Fe_{2+2.5-5.0})Si_8O_{22}(OH)_2$ |
| Albite, calcian, ordered | Plagioclase feldspar | $NaAlSi_3O_8$ |
| Pottassicpargasite | Ca amphibole | $KCa_2(Mg_4Al)(Si_6Al_2)O_{22}(OH)_2$ |

XRD mineralogy for single samples of lake and watershed bedrock, and samples of sediment taken from Core SQB2 (light gray, dark gray and basal gravel units). The lake bedrock is a quartz muscovite schist with chlorite minerals, and the watershed bedrock is composed of chlorite minerals, plagioclase, and Fe- and Ca-amphiboles.

Table 7. Unmodeled and modeled calendar age distributions for radiocarbon (RC) samples 1-4. Agreement (Amodel) indices represent the % overlap between the modeled and unmodeled distributions.

a. OxCal P_sequence results.

| RC # | Unmodeled (AD) | Modeled (AD) | Amodel |
|---|---|---|---|
| RC sample 1 | 1523 - | 1770 - 1800 | 54 |
| RC sample 2 | 1682 - 1935 | 1680 - 1780 | 52.9 |
| RC sample 3 | 1654 - | 1650 - 1770 | 88.1 |
| RC sample 4 | 1492 - | 1490-1670 | 113.1 |

b.  OxCal P_sequence results with 1700 CE used as a calendar date (C_date) in the model in place of radiocarbon sample 2.

| RC # | Unmodeled (AD) | Modeled (AD) | Amodel |
|---|---|---|---|
| RC sample 1 | 1523 – 1800 | 1770 – 1800 | 60 |
| RC sample 2 | C_date = 1700 CE | | |
| RC sample 3 | 1652 - | 1660 – 1690 | 101.8 |
| RC sample 4 | 1495 - | 1520 – 1670 | 101.9 |

Amodel is the model agreement index used to see if the model as a whole is not likely given the data and should usually be over 60%.

Table 8. Event ages for deposits A-J based on the age model shown in Figure 16.

| Event ID | Mean | Median | Min | Max |
| --- | --- | --- | --- | --- |
| A | | | 1980 | 2013 |
| Between A/B | 1970 | 1970 | 1954 | 1985 |
| B | | | | |
| C | 1920 | 1920 | 1870 | 1970 |
| D | 1900 | 1900 | 1860 | 1960 |
| E | N/A | | | |
| F | 1870 | 1870 | 1830 | 1930 |
| G | 1920 | 1860 | 1830 | 1910 |
| H | 1850 | 1850 | 1820 | 1890 |
| I | 1850 | 1850 | 1820 | 1890 |
| J | 1740 | 1720 (1780)* | 1680 | 1780 |

*Multiple peaks

Table 9. Table of deposit characteristics and attributions.

| Attribution | Grading | Basal contact | XRD | Color[2] | Radiodensity (*HU*) | Magnetic susc.[3] scaled to radiodensity | Particle size and sorting |
|---|---|---|---|---|---|---|---|
| A 1997 CE Flood | normal | Sharp; erosive | Mixed | Brown; darker in the lower half (organics) | High (~8–900) throughout | High throughout, but rises slowly from the base upward (compared to radiodensity) | Dominated by coarse silt, capped by a thin layer of fine-grained silty clay. Thin layer of coarse silt and fine sand at base. |
| B 1964 CE Flood | normal | Sharp; erosive | Mixed | Brown; darker in the lower half (organics) | High (~8–900) throughout | High throughout, but rises slowly from the base upward (compared to radiodensity) | Dominated by coarse silt, capped by a thin layer of fine-grained silty clay. |
| C 1927 CE Flood and dam failure | normal? | Wavy; discontinuous | N/A | Slightly grayer than background | Moderately high | Similar to radiodensity, but lower amplitude | Too thin to sample for accurate particle size; missing in core SQB5 (eroded from section). |
| D 1906 CE San Andreas EQ | normal? | Wavy; irregular | Watershed | Orange | Moderately high | Slight increase | Two discontinuous wavy layers: the lower layer is orange in color and composed of fine-medium grained silt. |
| E Local subaerial wall failure deposit | normal | Sharp; erosive | Schist | Dark Gray | Moderately high | Slight increase | Visible mica flakes in this unit. This unit is observed in only one core (SQB5), which is near a steep slope. CT density increases but magnetic susceptibility change is subtle. |
| F Uncertain; flood and dam failure, 1892? | normal | Sharp | N/A | Medium Gray | High | Slight increase | Thin and difficult to characterize, of different thickness in each core. Radiodensity increases, but magnetic susceptibility change is subtle. Load structures at base. |
| G 1890 flood | reverse, then normal | Indistinct | N/A | Faint | Moderately high; rounded profile | Indistinct, but similar to radiodensity | Particle size data N/A. Slightly stiffer higher density sediment; peak radiodensity midunit. |

| Attribution | Grading | Basal contact | XRD | Color[2] | Radiodensity (*HU*) | Magnetic susc.[3] scaled to radiodensity | Particle size and sorting |
|---|---|---|---|---|---|---|---|
| H 1873 CE intraplate earthquake | normal | Sharp | Watershed | Light Gray | High | Similar to radiodensity, but lower amplitude | Medium silt; fining upward. |
| I 1873 CE intraplate earthquake | reverse, then normal | Sharp; erosive | Schist | Dark Gray | High | Similar to radiodensity, but lower amplitude | Coarse silt, poorly sorted. In some cores reverse, then normal grading. |
| J 1700 CE Cascadia Earthquake | normal | Sharp | Watershed | Light Gray | High (~8–900) | High; correlated with radiodensity | Basal silt is very fine-grained and well-sorted, becoming less-well sorted upward. Load structures at base. |

[1]See Figure 12. [2]Brown = 2.5Y 3/2, Light Gray = 2.5Y 4/1, Dark Gray = Gley 2 4/5PB, Orange = 5Y 4/1. Variations in colors through deposits were visibly obvious but frequently difficult to differentiate from one another using Munsell color charts. [3]Magnetic susceptibility variability was compared to the variability in radiodensity. Note that magnetic susceptibility data is influenced by surrounding sediment (exponential decrease with distance), and therefore the magnitude can be influenced by the thickness of the unit if thin (~1 cm or less; see Figure 13). Note: "Watershed" refers to the Slickear Creek watershed.

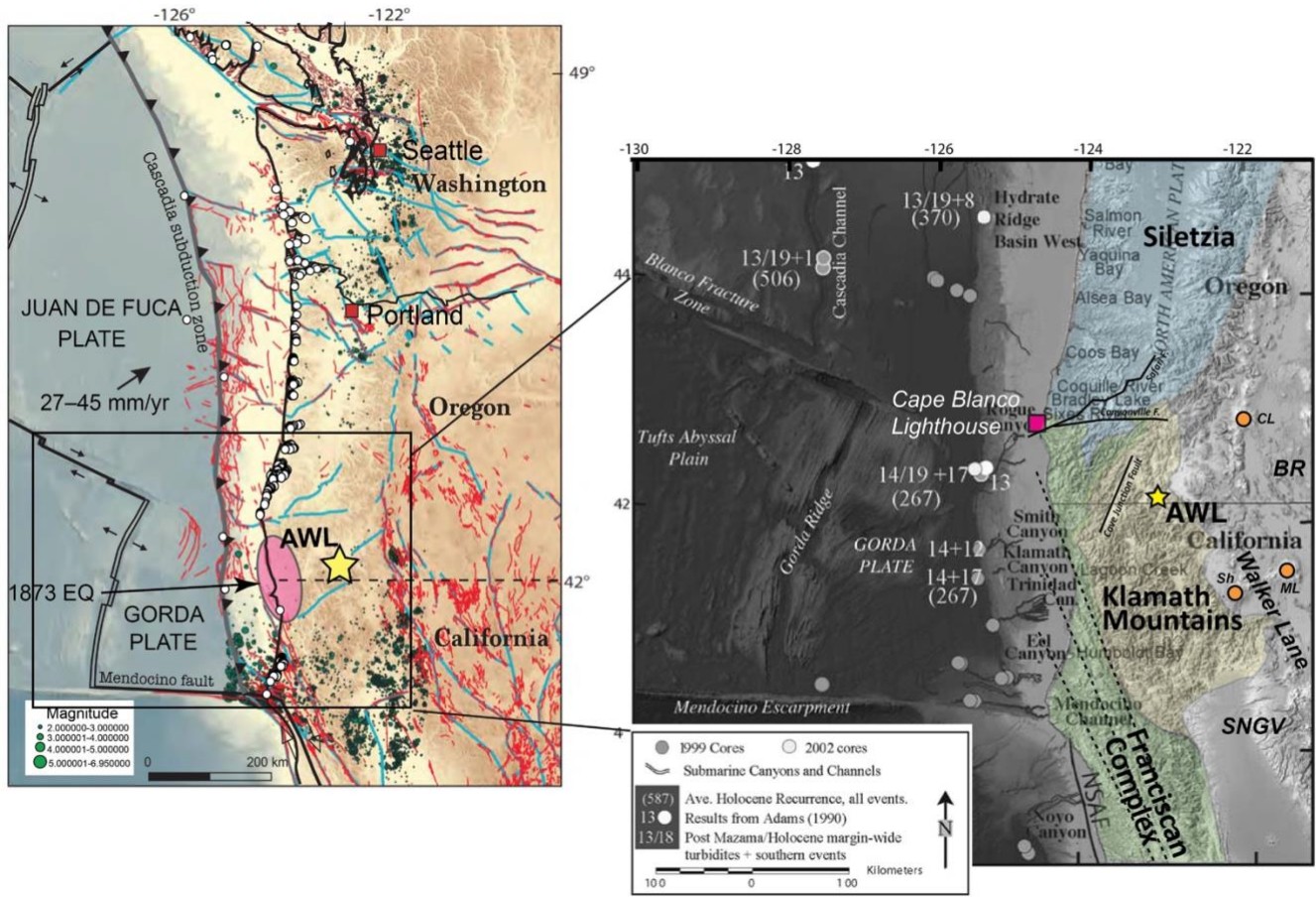

Figure 1. *Location Map and Tectonic Setting*. Left: Map showing the location of Acorn Woman Lakes with respect to the Cascadia subduction zone. Upper and Lower Acorn Woman Lake are located in southern Cascadia approximately 180 km inland east of the deformation front. The open circles indicate the locations of coastal paleoseismic sites (base map modified from Leonard et al., 2010). The pink oval represents the approximate location of the epicenter for the 1873 CE Brookings earthquake. Faults from a simplified Cascadia forearc fault model (Canyonville and Safari faults; Wells et al., 2017). Acorn Woman Lakes are located at ~35 km above the transition from the seismically to aseismically slipping reaches of the plate interface. Figure adapted from Data Repository Figure 3c by Wells et al., 2017; Mb > 2.0 from McCrory et al. (2012); USGS Quaternary Fault database (red lines). Right: Map expanded to show details of the geologic terranes and highlight the locations of local crustal faults near Acorn Woman Lakes; open circles are from Goldfinger et al., 2012. AWL = Acorn Woman Lakes, CL = Crater Lake, ML = Medicine Lake volcano, Sh = Mount Shasta, NSAF = northern San Andreas fault. Dashed lines are a simplification of the zone of faults that include the Lost Man fault, Bald Mountain fault zone, Grogan fault, Eaton Roughs fault zone, and Lake Mountain fault zone.

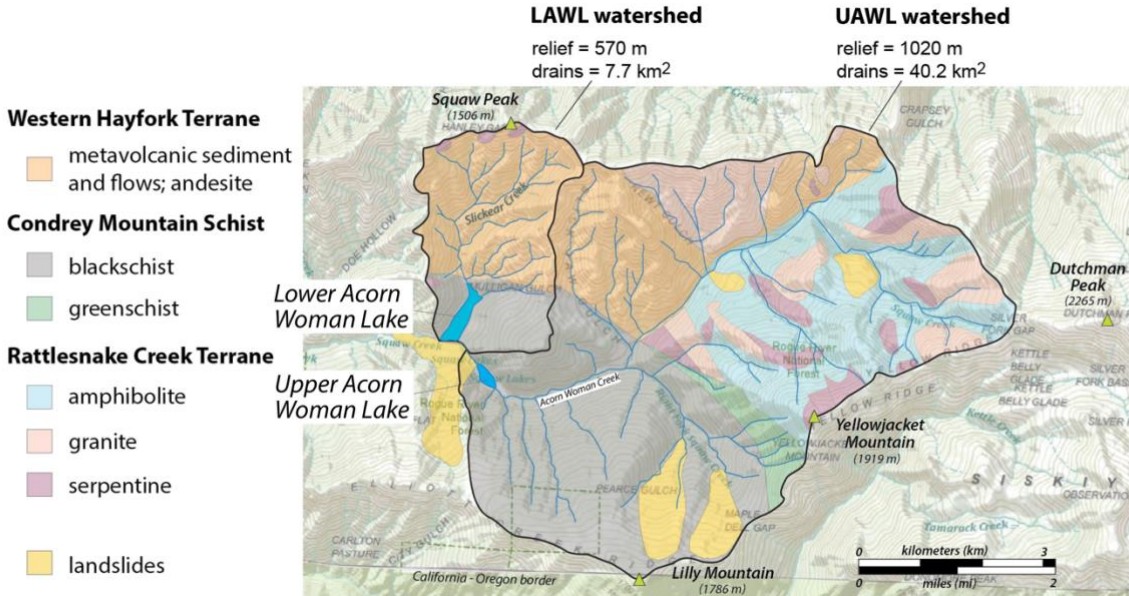

Figure 2. *Geologic and Geomorphic Setting.* Upper and Lower Acorn Woman Lakes are situated in the Condrey Mountain Schist; however, the bedrock of the lake catchments consists of distinctive metamorphic and plutonic lithologies. Note the large landslide responsible for lake creation. Geology from Donato, 1993.

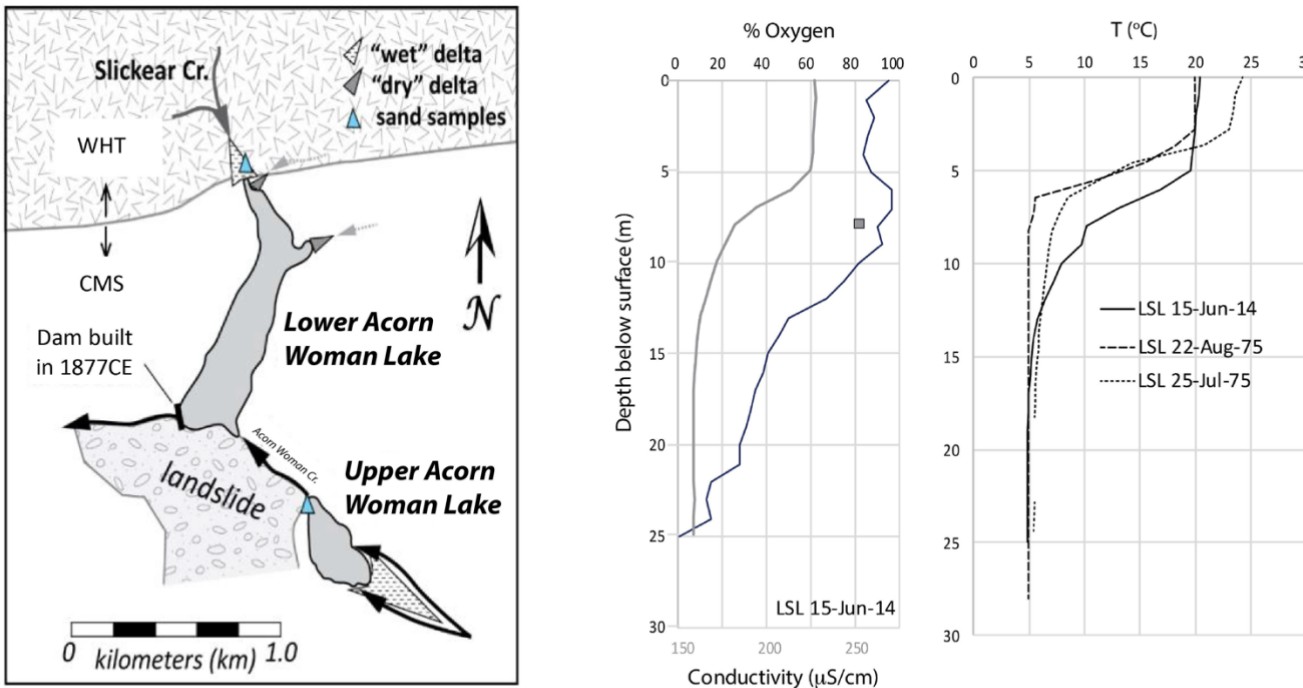

Figure 3

Figure 3. Left. *Upper and Lower Acorn Woman Lakes can be thought of as a single system, with the landslide between them separating the two basins.* Upper Acorn Woman Lake is fed by Acorn Woman Creek, which forms a large sandy delta ~0.5 km long as it enters the lake from the southeast. Acorn Woman Creek continuously flows between the two lakes, transporting water and sediment from Upper Acorn Woman Lake to the southern end of Lower Acorn Woman Lake. Blue triangles indicate the locations of sand samples taken to determine sediment provenance. CMS: Condrey Mountain Schist, and WHT: Western Hayfork Terrane. Wet deltas are saturated and dry deltas are not (inferred from vegetation types). Right: Temperature, conductivity, and % oxygen for Lower Acorn Woman Lake. Data collected in 2014 were supplemented by data from Larson (1975). The lake is stratified with a thermocline between 5-7m water depth. At this same depth an unstable spike in conductivity was measured (left) indicating the presence of groundwater flowing into the lake at that depth.

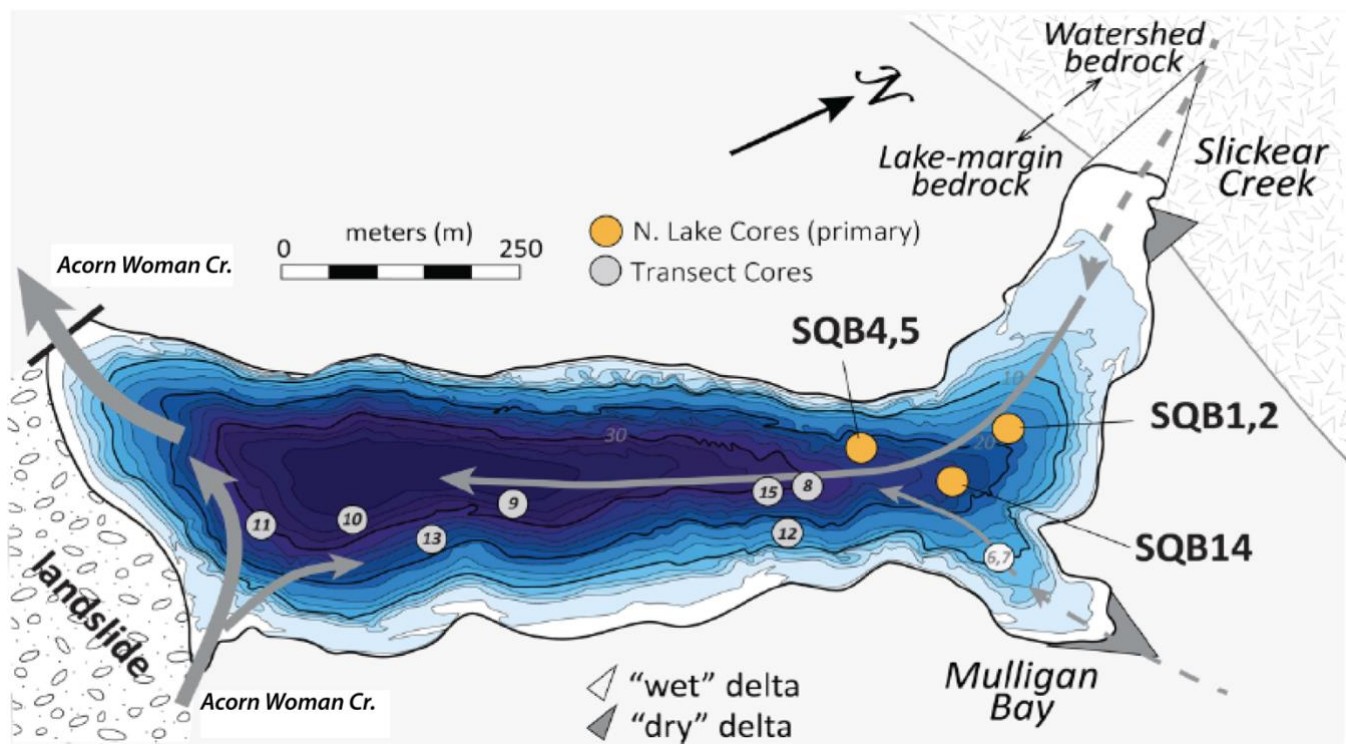

Figure 4. *Core locations*. Cores were collected along a north–south transect. The primary core sites (northern cores) are identified by the orange circles, and the other cores collected are identified by the numbered grey circles. The northern cores were used because the deeper water cores were disturbed as a result of coring in the historic portions of the records.

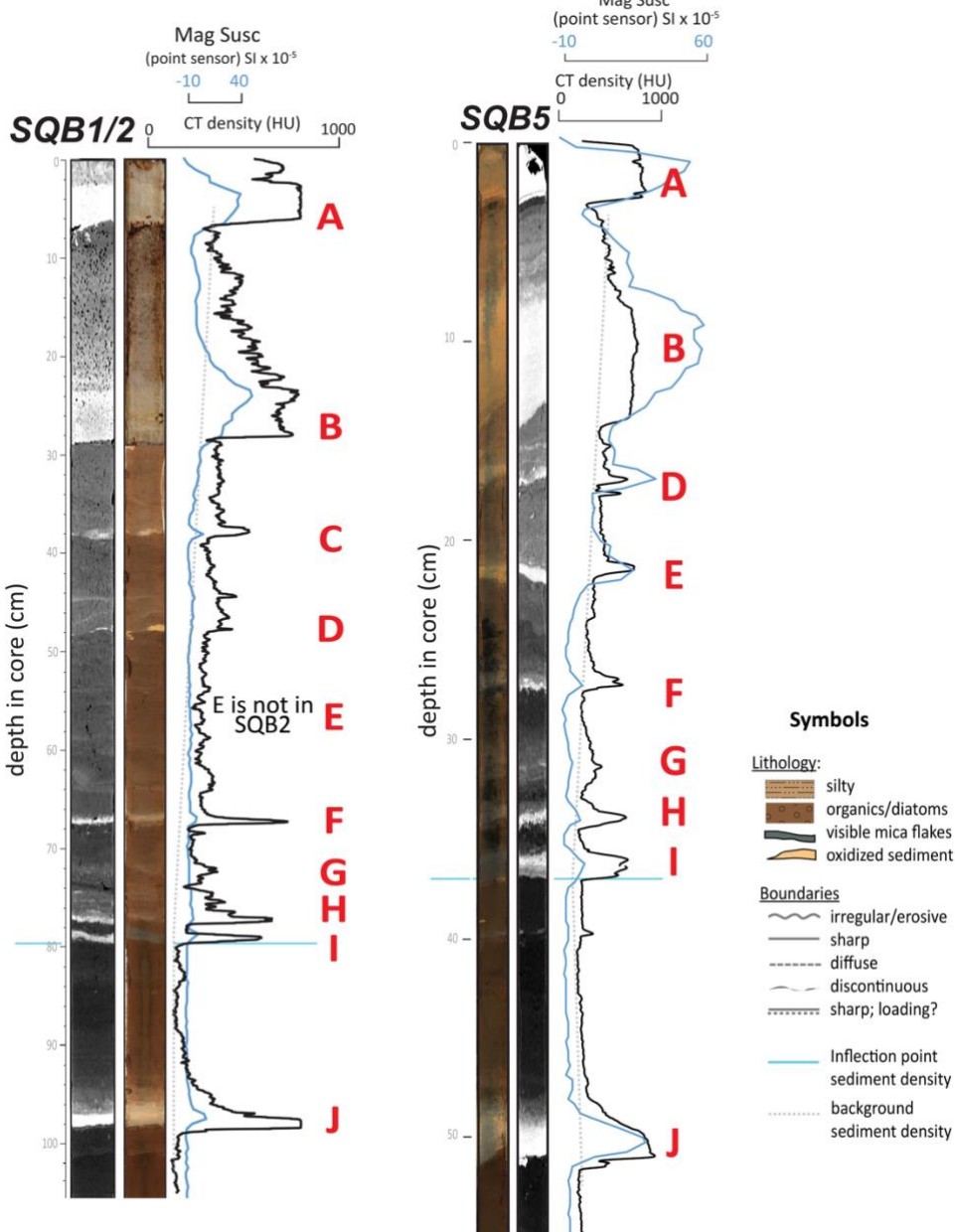

Figure 5. *Disturbance deposits A-J were identified as excursions in radiodensity and magnetic susceptibility in cores SQB1/2 and SQB5*. Shown are grayscale images of radiodensity and RGB images of core color. Blue traces are magnetic susceptibility (SI units) and black traces are radiodensity (Hounsfield units). Note that deposit C, present in core SQB2, is missing from core SQB5, and deposit E is missing from core SQB1/2 (and all other cores). The dashed line reflects background sediment density, showing an inflection in the trend near deposits H and I.

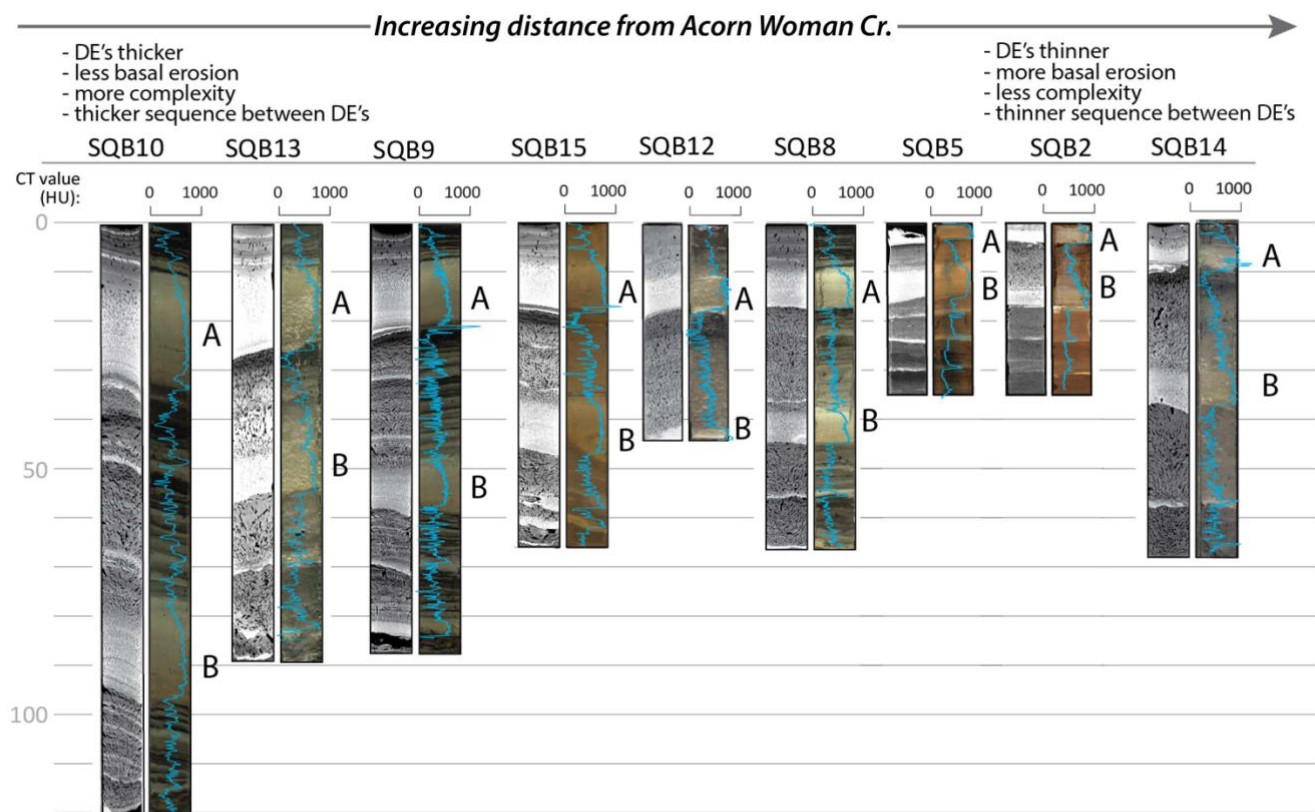

Thickness, in cm:

|  | SQB10 | SQB13 | SQB9 | SQB15 | SQB12 | SQB8 | SQB5 | SQB2 | SQB14 |
|---|---|---|---|---|---|---|---|---|---|
| Deposit A | 22 | 16 | 13 | 10 | 11 | 9 | partial | 3 | 6 |
| Between | 42 | 13 | 27 | 18 | 10 | 20 | 5 | 6 | 13 |
| Deposit B | 20 | 17 | 12 | 10 | partial | 8 | 7 | 4 | 15 |
| water depth | 35 | 25 | 36 | 29 | 20 | 30 | 24 | 17 | 30 |

Figure 6. *Two distinctive lake-wide inorganic disturbance event deposits were observed in the upper portions of all cores.* Event deposits and interevent sediment thicknesses are all greater closer to creek inflows and in the deeper water cores.

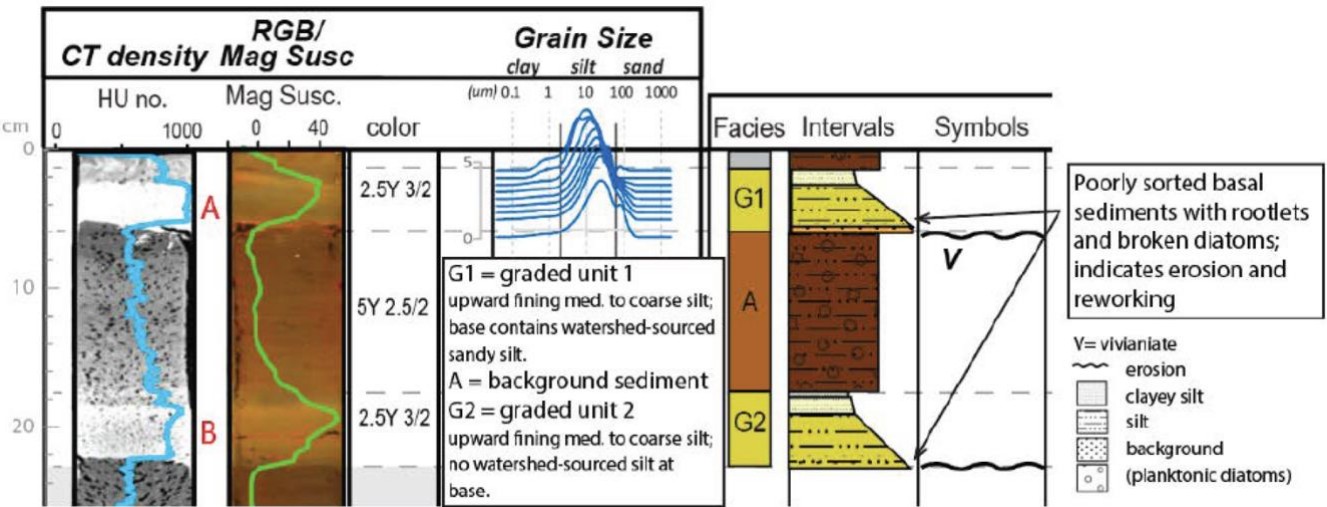

Figure 7. *Deposit types A and B in the surface core SQBss (at the location of cores SQB1 and SQB2).* Facies types G1 and G2 are very similar, other than the presence of a thin layer of sandy silt at the base of deposit A. The bases of each deposit are poorly sorted and coarse-grained, containing rootlets and broken diatoms, indicative of erosion and reworking. Deposits are upward fining, with a clayey silt cap.

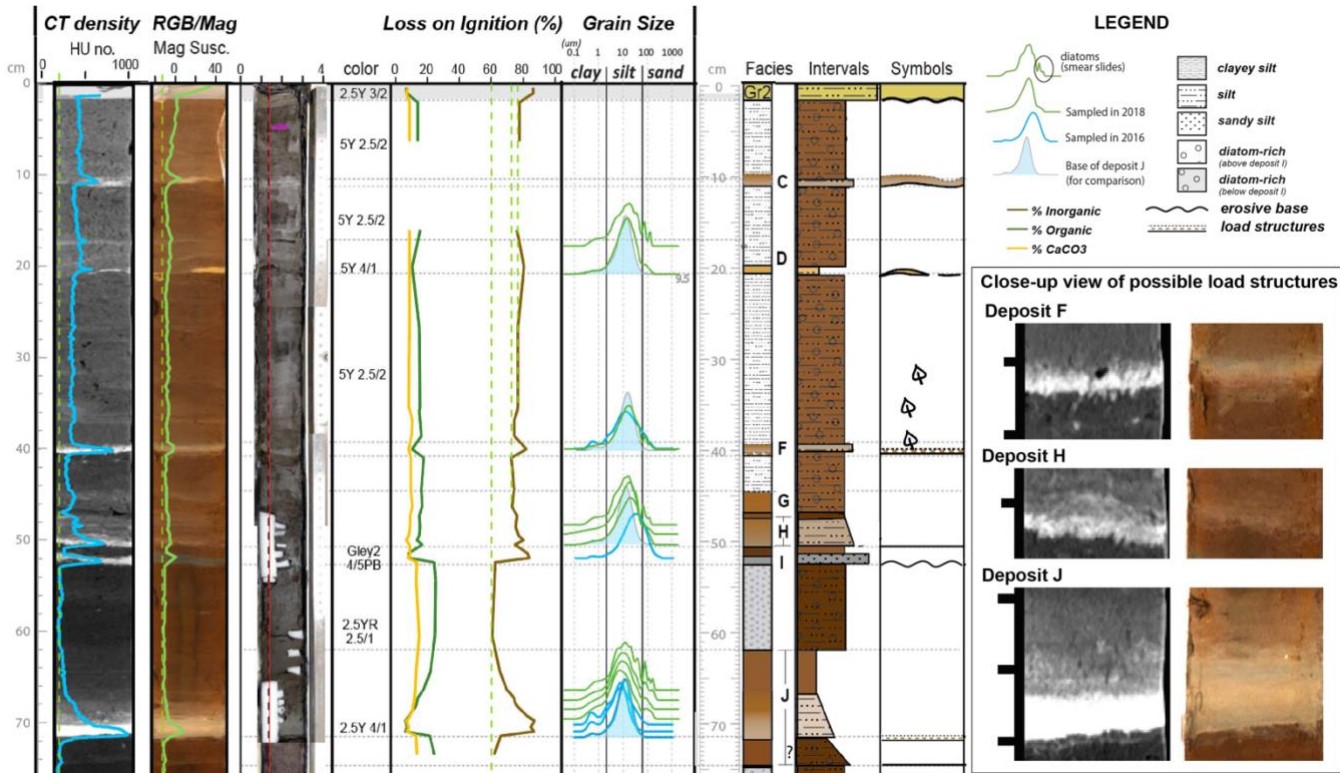

Figure 8. *Core log data for deposits C-J section SQB2A (stratigraphically below deposits A and B).* Particle-size data are shown as distributions rather than ratios or median sizes to show how they compare from the base upward and also compare to the size and narrow range shown for deposit J (shown as a pale blue filled distribution). Data shown from left to right: CT data are presented as a greyscale image with the HU data shown in light-blue; RGB data are presented as a color image with the magnetic susceptibility data shown in light green (dashed line is a reference line indicating the lowest magnetic susceptibility for that section; photo compilation showing the location of some of the samples; Munsell colour; loss on ignition (%) data representing % inorganic (brown), % organic (green) and % calcium carbonate (yellow); grain size distributions; core log representation of the core where the interval horizontal widths reflect a combination of density and grading characteristics: wider intervals are denser and coarser grained and narrower intervals are finer grained with lower density; enlarged greyscale representation of raster radiodensity data for deposits that show evidence of silt loading on the less-dense organic sediment below.

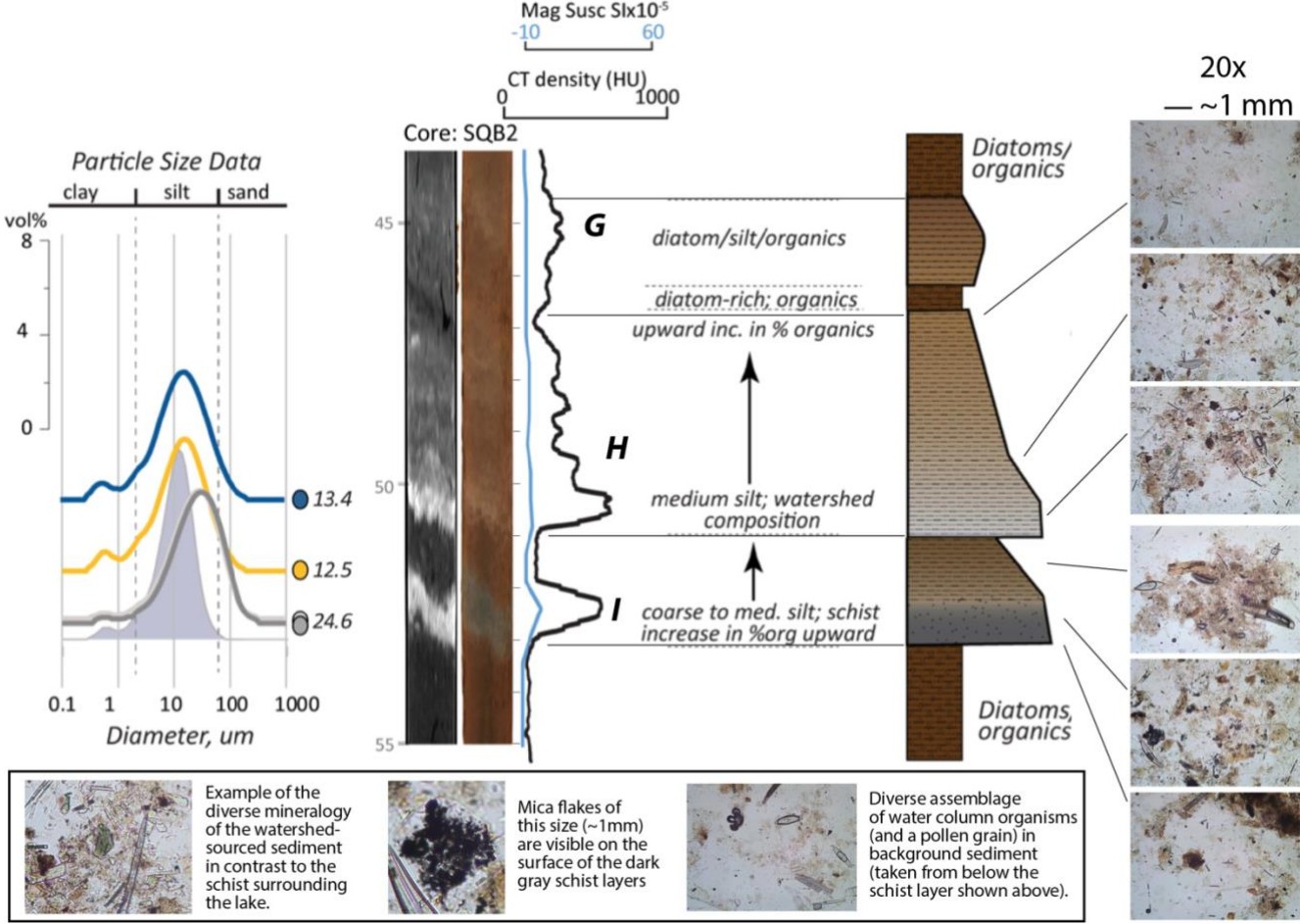

Figure 9. *Deposits G, H and I in core SQB2*. This figure shows particle-size distributions from the layers shown in core SQB2 (median particle size, in mm, is shown to the left of the coloured dots indicating sample locations in core). The blue (filled) distribution represents the particle-size distribution for disturbance J (shown for comparison). Core imagery produced from radiodensity and RGB colour data are shown in comparison to magnetic susceptibility (point sensor; light blue trace) and radiodensity data (black trace) taken from the centre of the core. Descriptions of the sediment are presented for each of the facies identified by the horizontal grey lines. To the right of the core imagery and facies descriptions is a schematic representation of the core where excursions to the right represent denser sediment. The smear slides show that the composition of the disturbance deposits varies in mineralogy and organic content. Note the different species of diatoms in the smear slide taken from the very top of the sequence (between disturbances G and H). At the bottom of the figure are higher resolution images of details from the smear slides shown at the far right of the schematic.

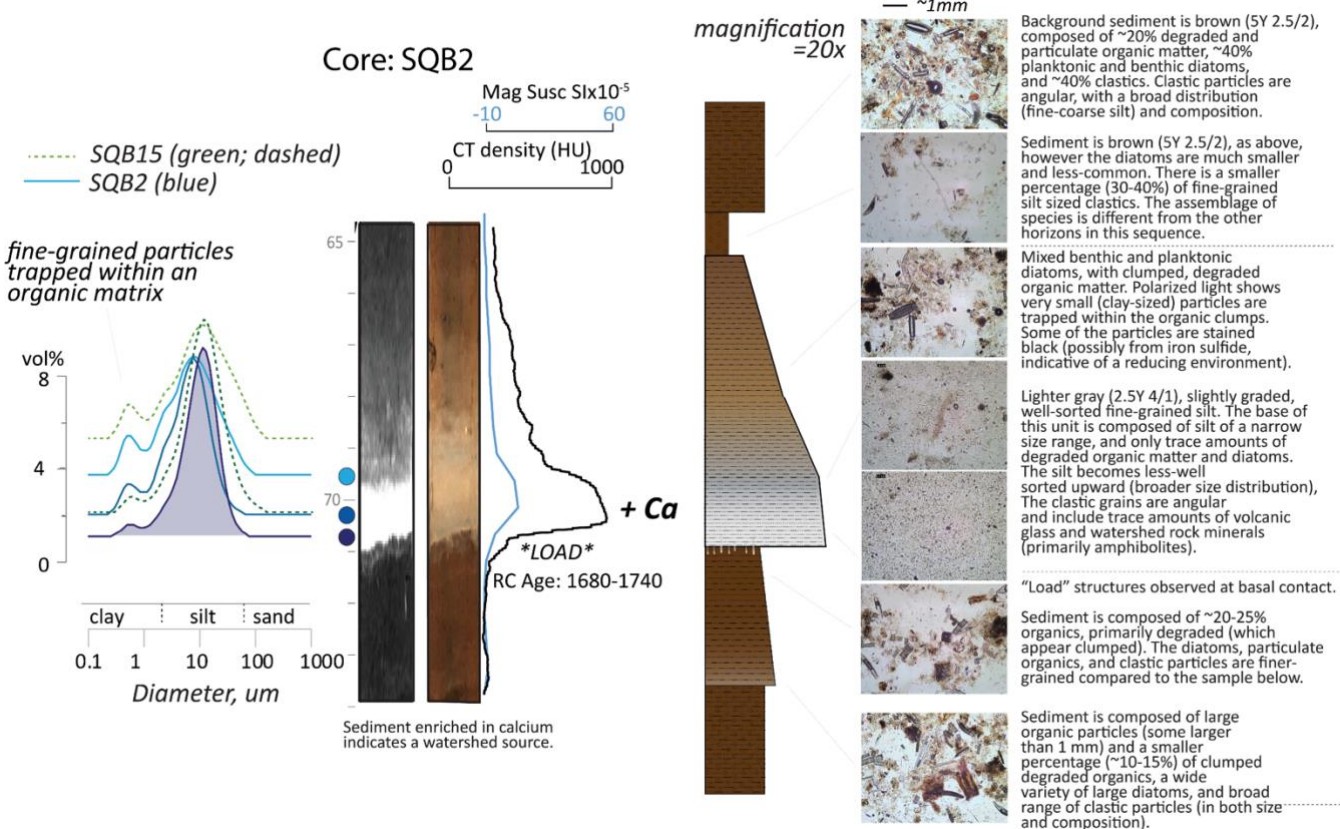

Figure 10. *Particle-size data, core imagery, physical properties, and schematic with smear slides and detailed descriptions of sediment composition for deposit J*. Particle size data for cores SQB2 and SQB15 both show a narrower distribution at the base of the silt, increasing in width upward. There is evidence of loading (or coseismic injection) of the silt unit onto the organic sediment below. Note that although magnetic susceptibility and radiodensity (light blue and black traces) show similar variability, there is greater detail present in the radiodensity allowing for the identification of a long deposit tail above not visible by eye, and the presence of another cryptic unit below the silt unit (light in colour and radiodensity > ~700 HU). These units also have distinct compositions as described in detail at right.

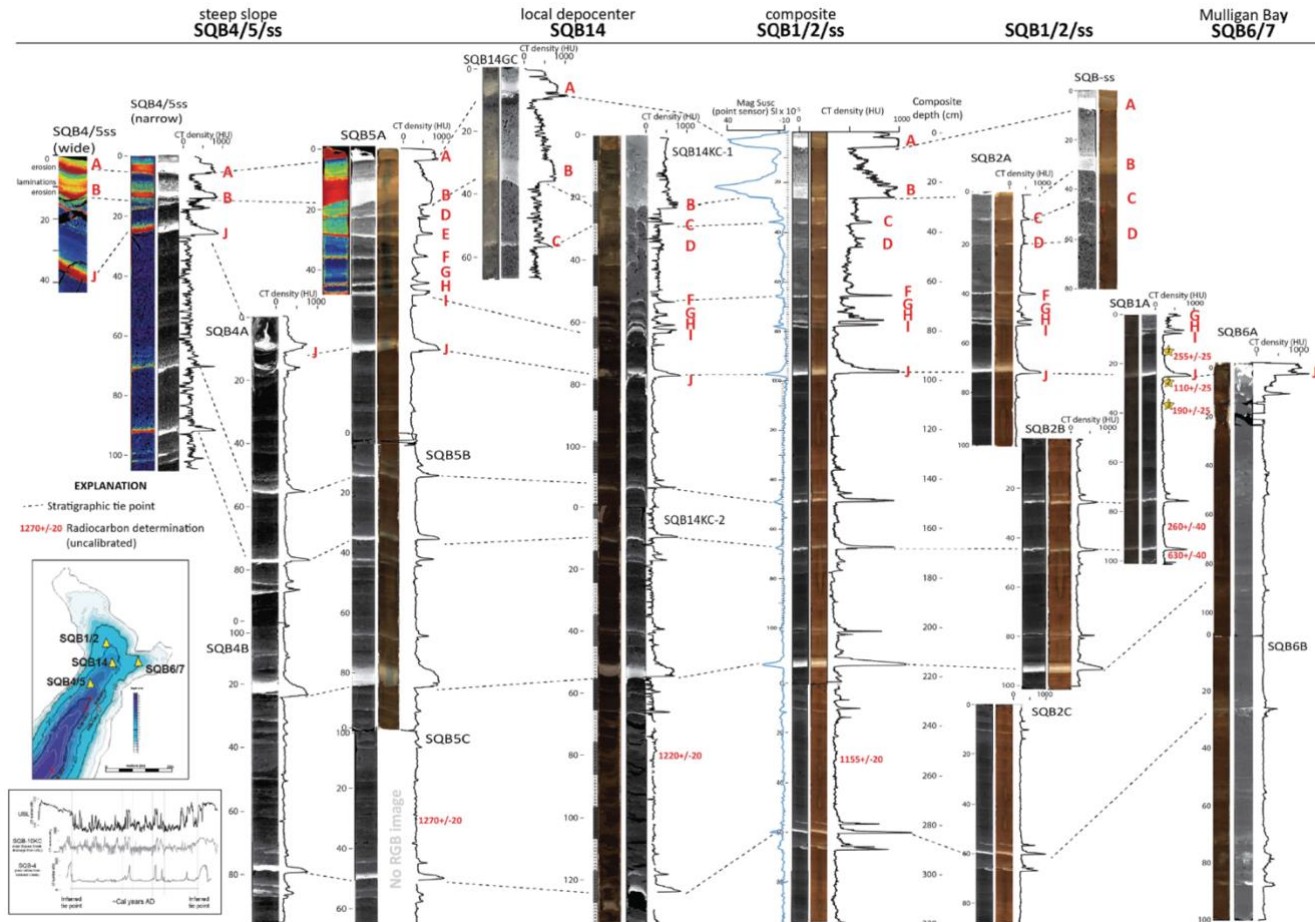

Figure 11. *Tie points between northern cores, SQB9 from the lake's depocenter, and the composite core created from SQB1, SQB2, and the surface sample*. Each downcore section of the northern cores is represented in this figure by a greyscale image of sediment radiodensity (brighter = higher radiodensity, darker = lower radiodensity), radiodensity traces (black lines; higher density to the right) through the core (in Hounsfield Units (HU)), and RGB colour imagery (other than a few exceptions). Note the strong similarity between each ~1-m section, as shown in the inset, lower left. To the right of the compilation core SQB1/2/ss are the sections and radiocarbon age data available to show how these sections of overlapping drives were spliced. Eroded sections are apparent when comparing the false colour radiodensity imagery in the cores SQB4, SQB5 and the associated surface sample.

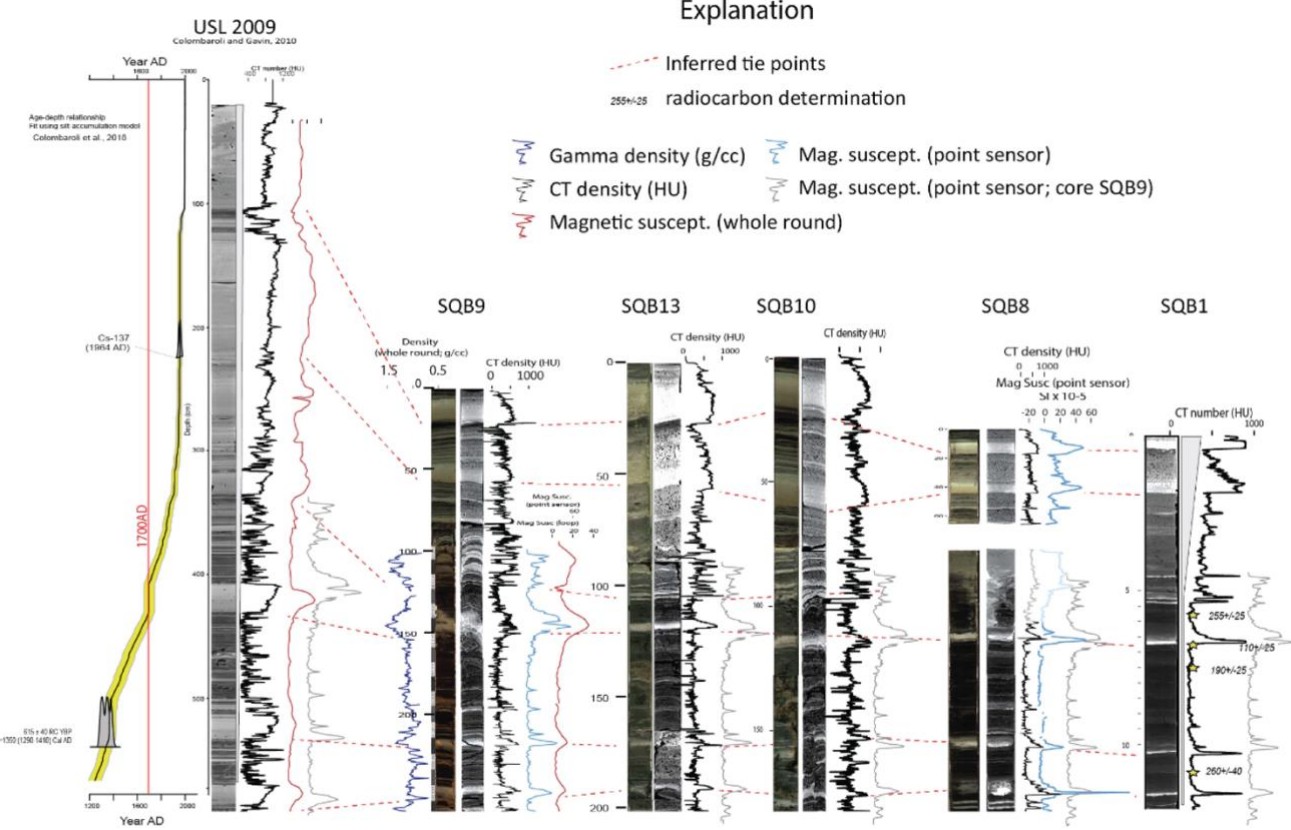

Figure 12. *Tie points between USL upper sections and the composite core from USL*. This figure shows the age-depth model, radiodensity imagery and physical property data for the Upper Acorn Woman Lake (USL 2009) core compared to key cores in Lower Acorn Woman Lake. All lines are dashed to indicate uncertainty, however the relationships between the lower lake cores are more certain than the relationship to the upper lake core because each of the cores has two distinct upper units that correlate to one another. In the USL core these units are greatly expanded, likely reflecting the higher sedimentation rate of the site. Note that each of the cores has the same pattern of disturbances as indicated by the ghost radiodensity trace (light grey) from SQB9, however deeper water cores from Lower Acorn Woman Lake such as SQB9, and the USL core, show higher frequency disturbances between the thicker, more dominant silt layers as compared to the shallower water cores such as SQB1.

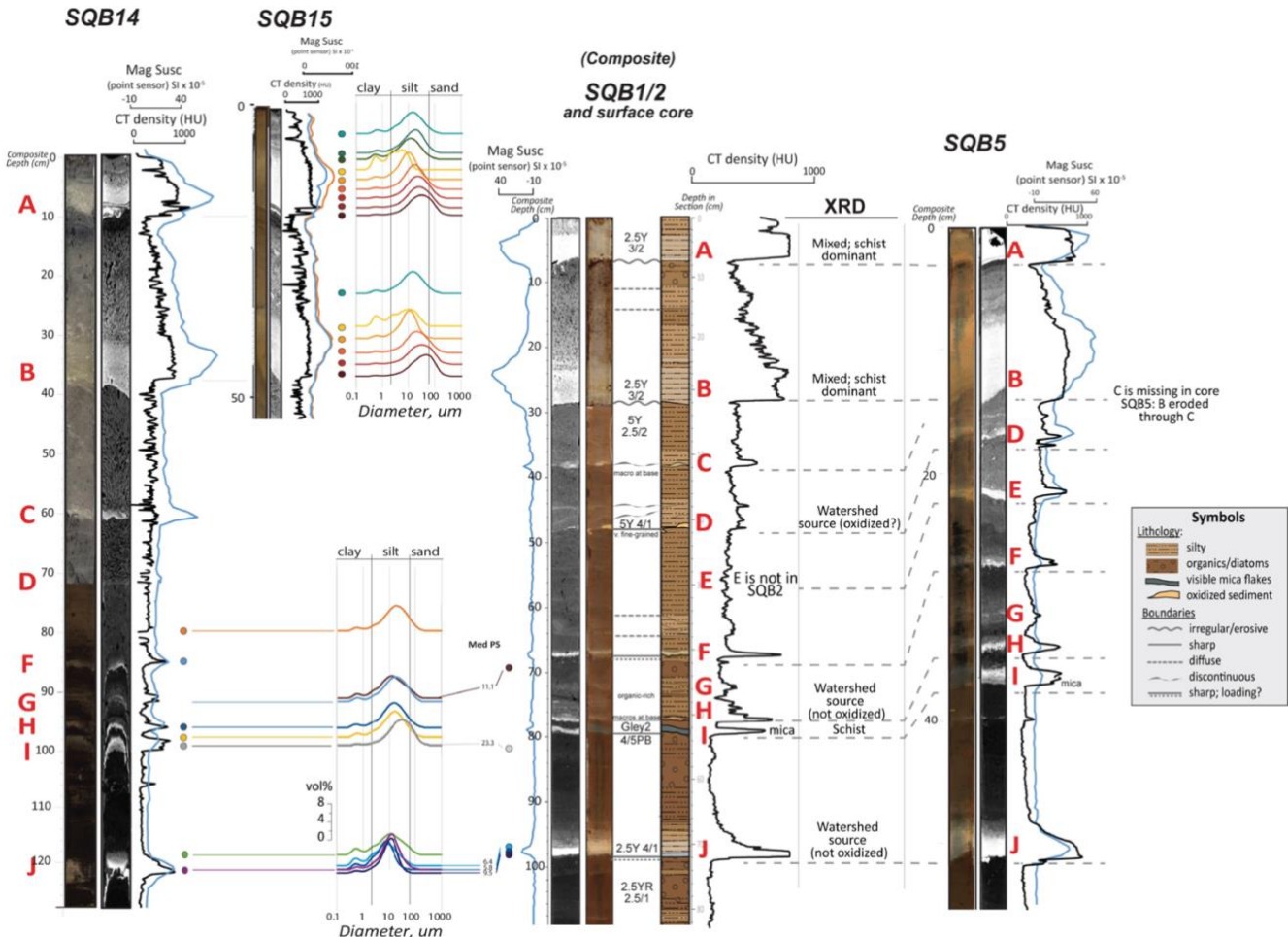

Figure 13. *Summary of the sedimentological data for the northern cores and correlations between cores.* Radiodensity was acquired from sediment cores while still as whole-round sections, and therefore contains methane pockets (black regions in radiodensity imagery). These pockets are very low density compared to the sediment and are the source of radiodensity noise in the radiodensity trace in Core SQB14 (especially between 40 and 85 cm in this figure). The bases of silt units were "flattened" to those of correlative units in the reference core SQB1/2 to emphasize the inferred relationships between cores. The anomalous disturbance event deposits (thicker or denser silt units) are identified by the letters A-J.

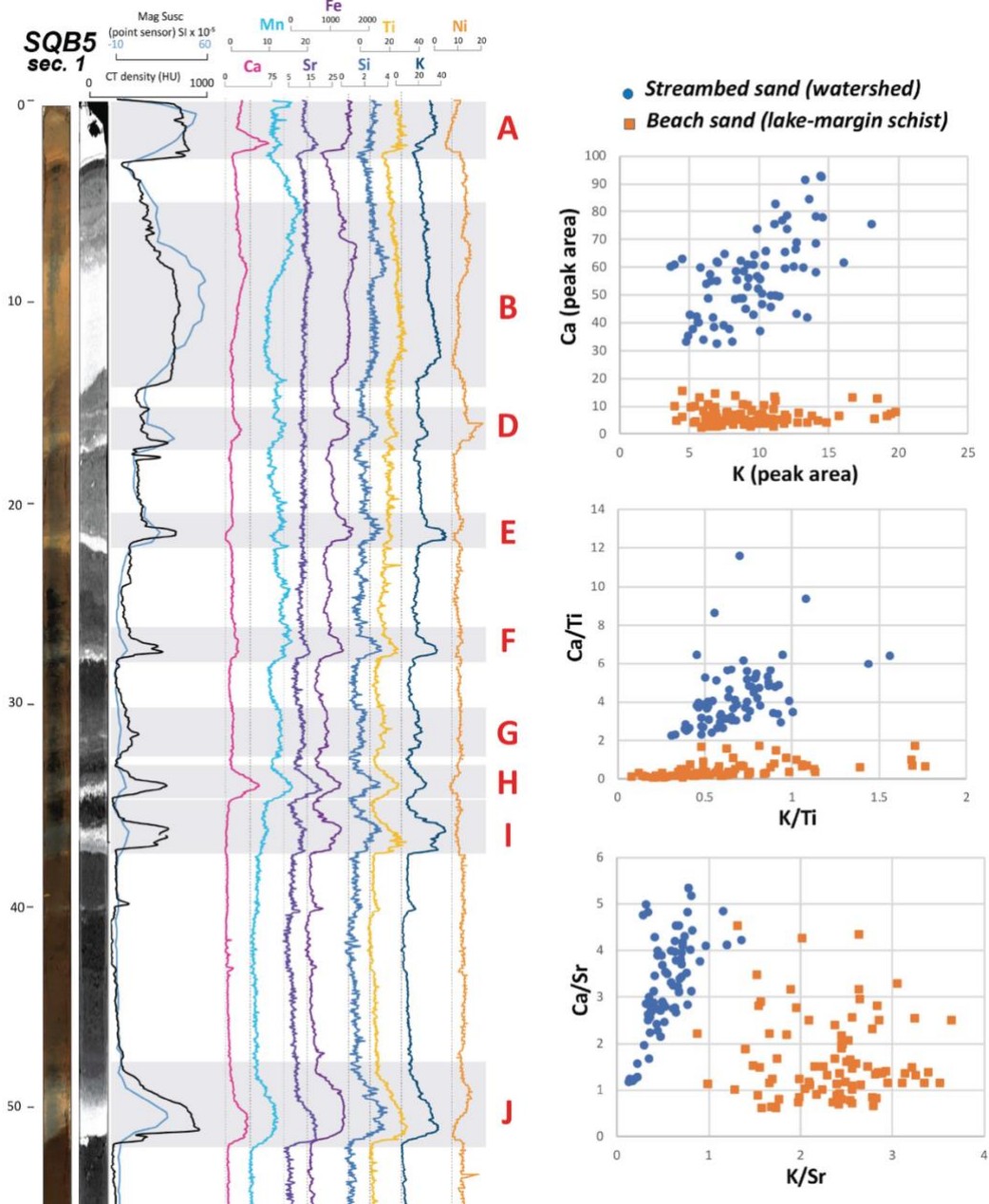

Figure 14. *X-ray fluorescence (XRF) data for core SQB5, Sec. 1*. Left: RGB imagery, radiodensity greyscale imagery, radiodensity (black trace), magnetic susceptibility (light blue line), and raw XRF data (peak area) for eight elements are shown in coloured lines. Right: Calcium (Ca) and potassium (K) from watershed and beach sand samples do not overlap, whether data are represented as raw counts (peak area) or normalized by titanium or strontium. This suggests that these elements (Ca and K) may be useful to identify sediment provenance end members.

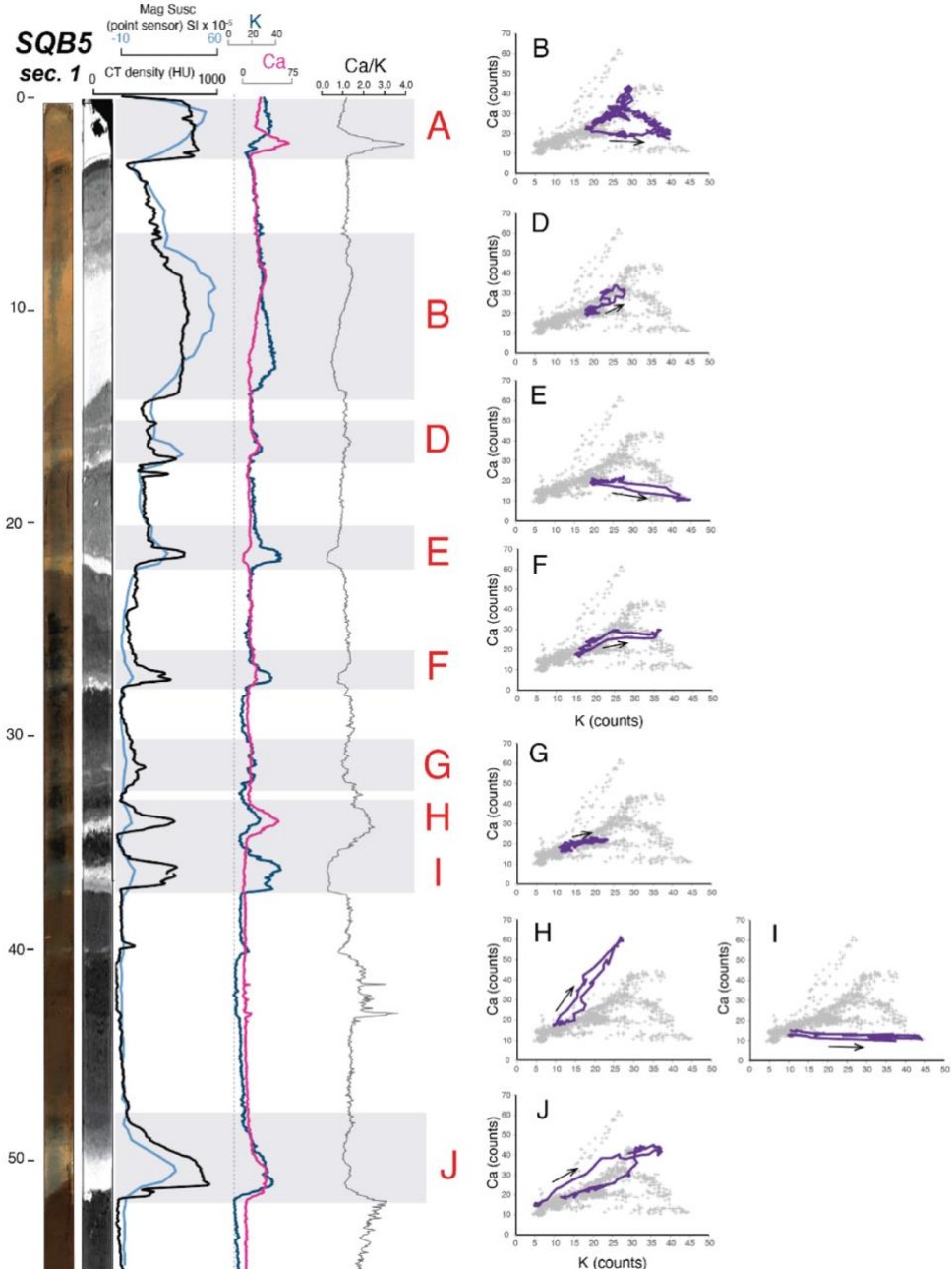

Figure 15. *Calcium and potassium downcore XRF data and scatterplots for deposits B-J in core SQB5*. Each of the grey bars represents the event deposit boundaries for units B-J based on the beginning and end of scatterplot loops (raw, unsmoothed data) at right. The direction of elemental composition from the base of the deposit upward is identified by the black arrows. Note that some of the scatterplots show clockwise evolution and others show counterclockwise evolution of calcium and potassium through the deposits. See the Discussion section for a detailed interpretation of the scatterplots.

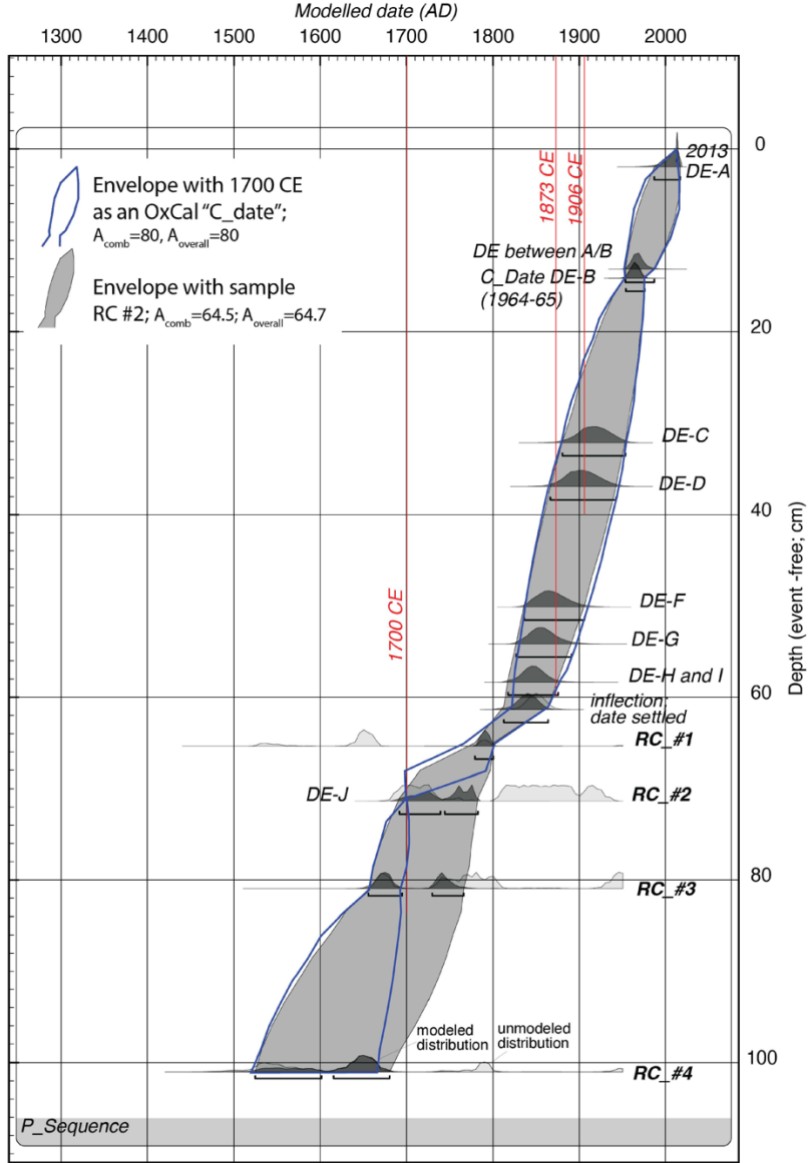

Figure 16. *Age-depth model for core SQB1/2/ss composite.* The age-depth model for composite core SQB1/2/ss was developed using event-free sediment accumulation as described in the Methods section. The depths of the event bases were used to extract age ranges and median values in calendar years for disturbances A–J (sample RC #2 was taken from just below the base of deposit J) and other unlabelled events in the sequence. The known timing of the 1700 CE, 1873 CE and 1906 CE earthquakes are identified by red vertical lines. Acomb = 64.5, and Aoverall = 64.7. A second version of the age depth model was created with a C_date of 1700 CE used in place of the radiocarbon sample #2 resulting in a higher Acomb and Aoverall (both of which are 80). Note that Aoverall refers to the product of the individual agreement indices, and Acomb refers to the test if distributions can be combined.

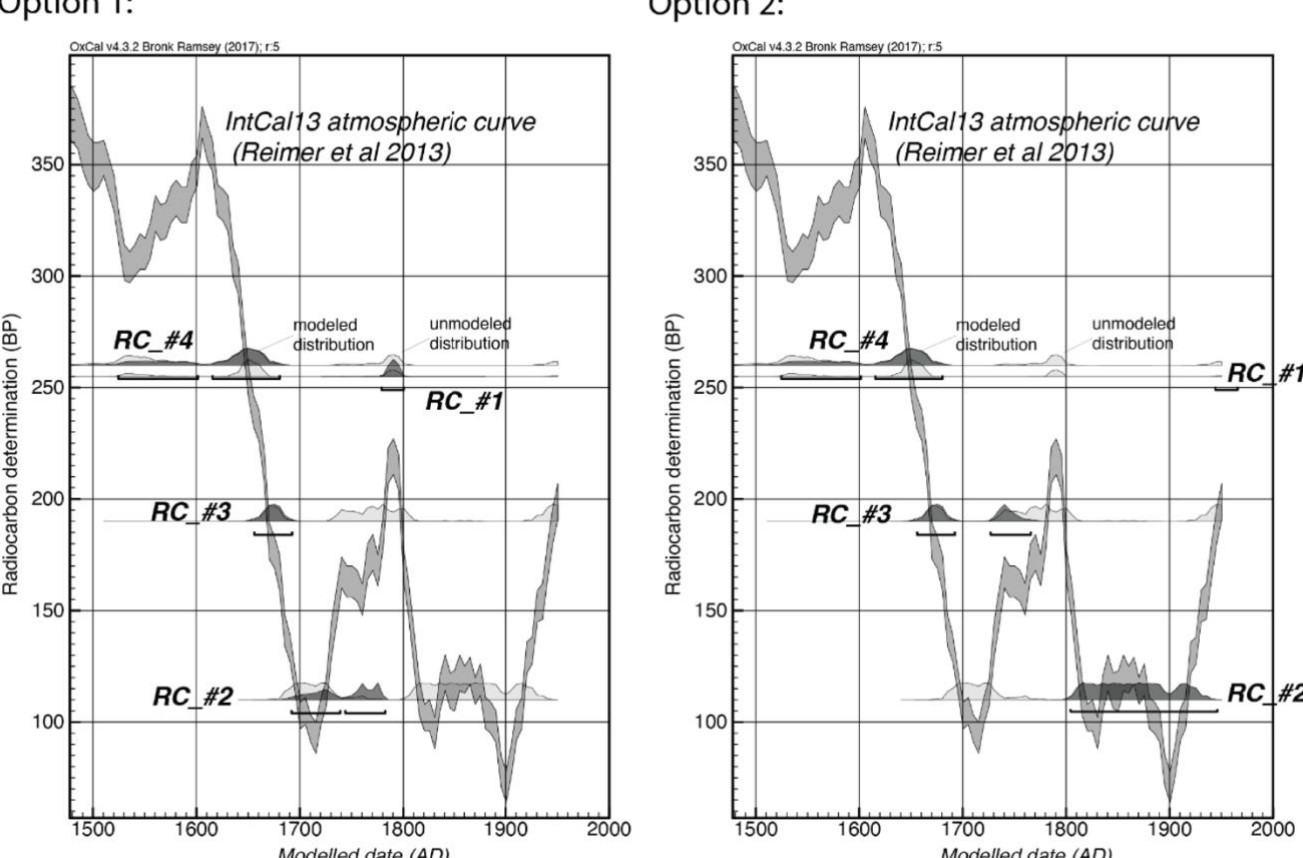

Figure 17. *Options for radiocarbon calibration assuming samples died close to the time surrounding sediment was deposited for deposit J*. Option 1 is the result of the OxCal P_sequence age-depth model presented in Figure 16 (with RC #2), and Option 2 is the alternative. Because the event-free sediment thickness is 71 cm (representing hundreds of years, Option 1 is considered most likely. See text.

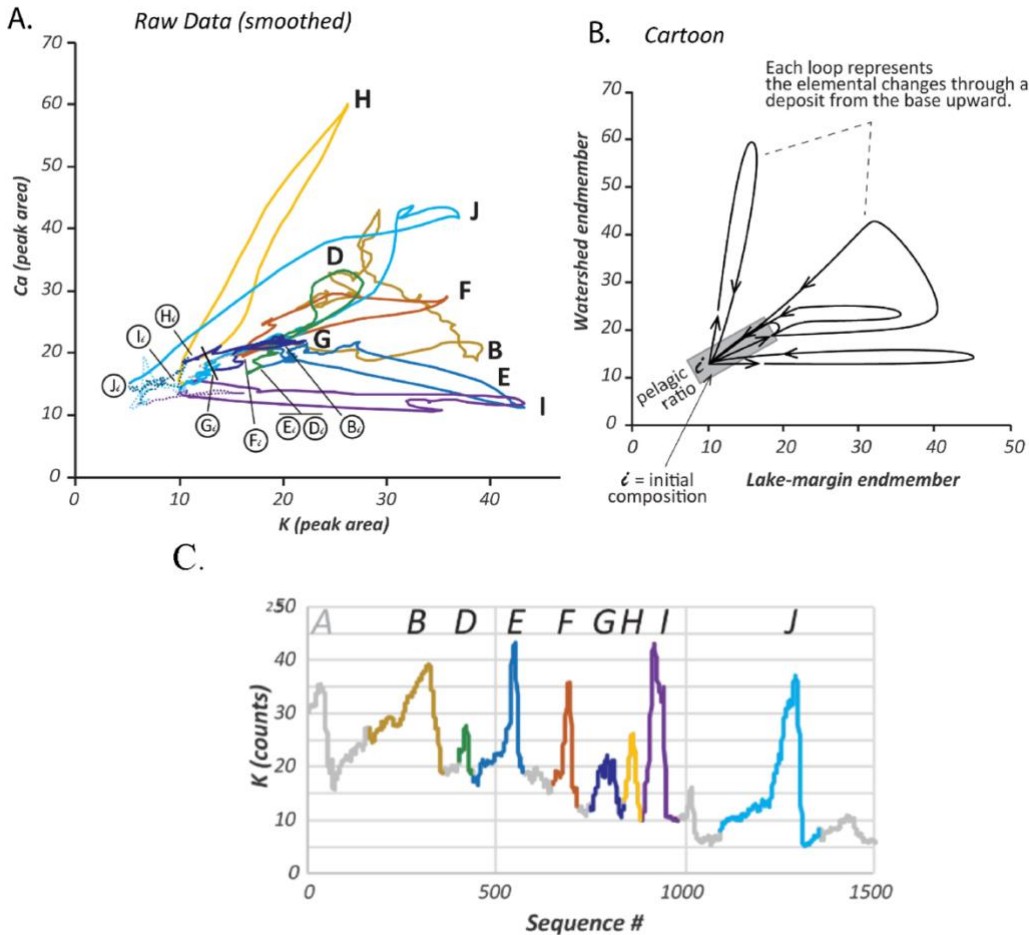

Figure 18. *XRF scatterplots; results and interpretation*. Coloured lines represent the individual disturbance deposits also shown individually in Figure 15. A) Variability through the disturbance deposits in core SQB5 can be expressed in terms of the endmembers K and Ca (shown here after smoothing). Each of the deposits displays variability that is unique and related to the initial composition of the disturbed sediment, sediment partitioning, and/or additional inputs during deposition. B) This cartoon demonstrates how patterns in the data may be interpreted as changes in composition of provenance indicators from the base of an event to the top as it evolves during deposition. Arrows show the direction with depth from the base of the deposit to the top as in Figure 15. Each deposit begins at the initial background ratio between Ca and K and increases and decreases along a distinctive path before the deposit ends as composition returns to the initial background ratio. A suggested explanation for these patterns is that they reflect the relative amounts of each variable (calcium and potassium), and a third implicit variable related to sediment density. C) Key to colours represented by the data shown in A); the vertical axis is K (raw counts), and the horizontal axis is the step number in the sequence downcore.

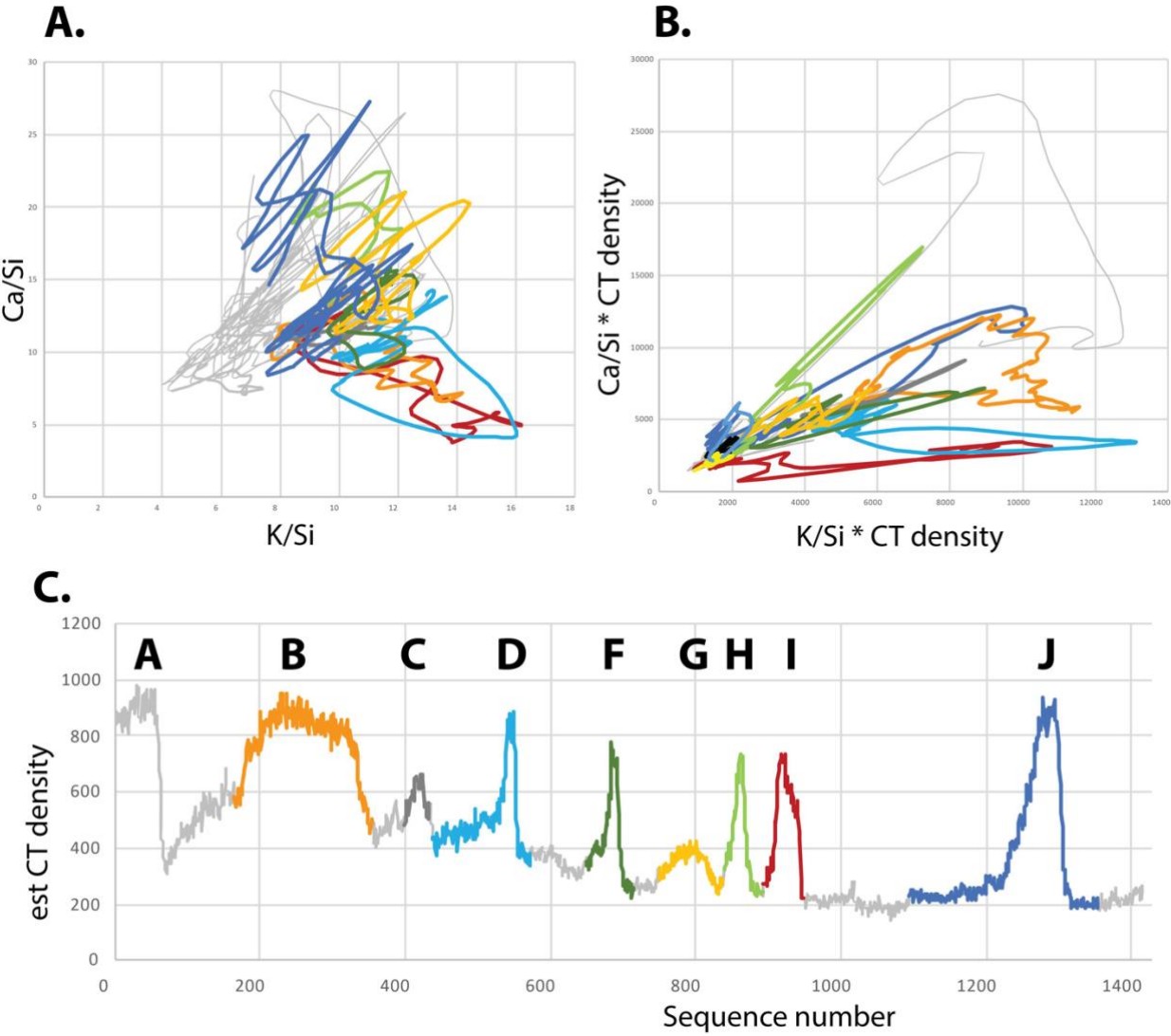

Figure 19. *XRF data normalized by silicon and scaled by radiodensity*. The smoothed, raw data were normalized by silicon (A), then scaled by radiodensity (B). The similarity between (A) in Figure 18 and (B), this figure, suggests that the relationships between variables calcium and potassium for each of the disturbance event deposits A-J (C) are different and reflect both sediment provenance and sediment density.

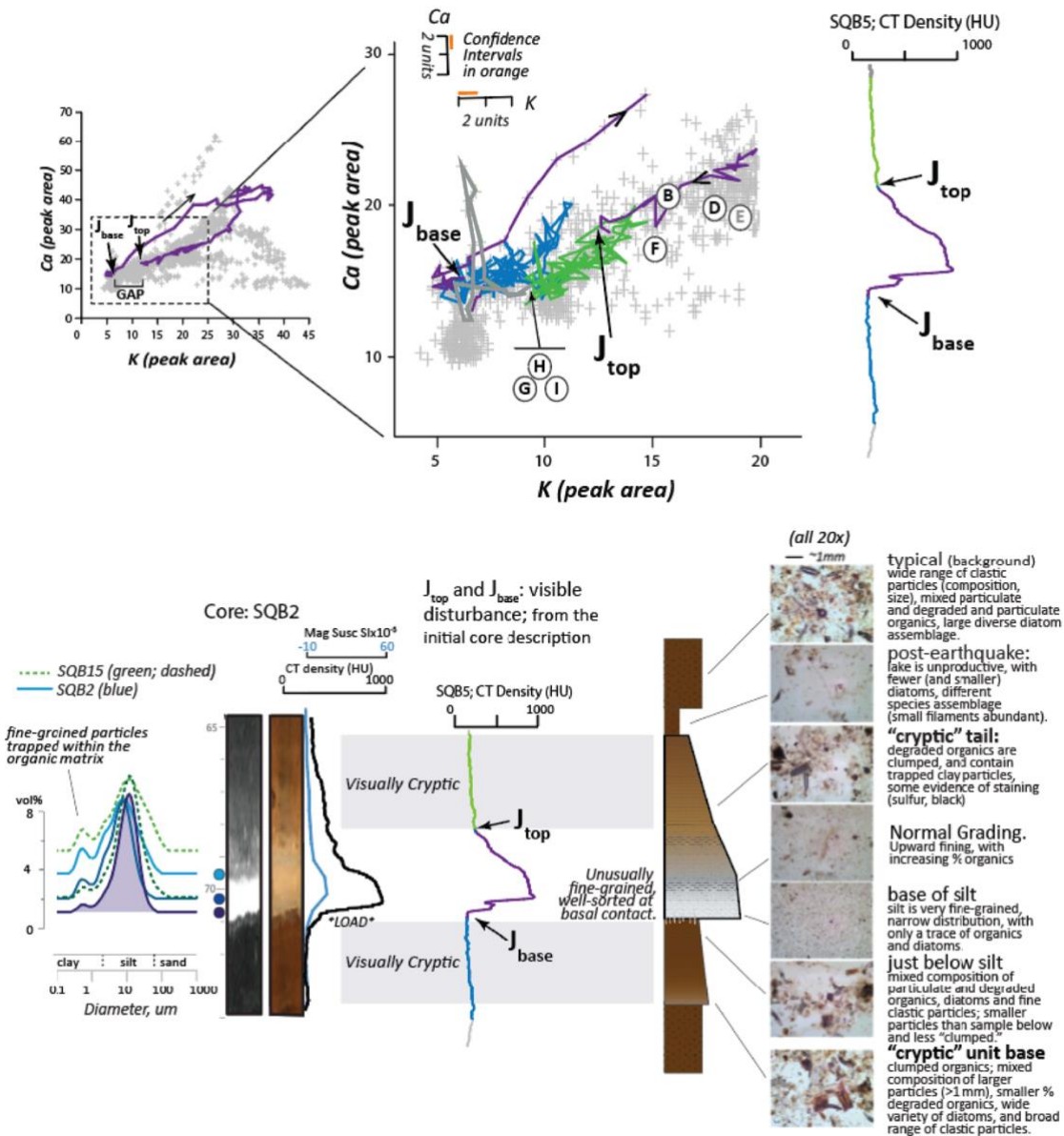

Figure 20. *Deposit J is a complex sequence with cryptic components*. Upper panel: The gap that exists between the initial and final ratios in the XRF scatterplot for deposit J can be filled in by including the sediment below (shown in blue) and by interpreting the upper portion to be a long tail (shown in green). XRF calcium and potassium confidence intervals (inset in blow-up of the scatterplot, centre top) were produced by repeatedly measuring the same section of core. The cryptic components of deposit J meet close to the initial positions of deposits G, H and I. The base of the visible silt unit is slightly enriched in calcium relative to background and is very well-sorted and fine-grained. This unit appears to "bleed down" from the clastic base into the very fine, organic-rich sediment below suggesting loading which can result when dense sediment

abruptly settles onto sediment that is less dense. Alternatively, this could be the result of coseismic injection of silt into the sediment below. The cryptic tail of the deposit is followed by a change in the size and types of diatoms and other components suggesting a post-earthquake change in water column organisms, possibly as a result of flocculation. Note that the data in the scatterplot are not smoothed to show the true variability of the data. Note also the pulse of watershed-sourced sediment (high in calcium, low in potassium) identified by the grey line above the cryptic tail of deposit J – could this be the result of a post-subduction earthquake aftershock, another subduction earthquake, a large crustal or intraplate earthquake? Lower panel: Descriptive core data and imagery are shown for deposit J and surrounding sediment with the visually cryptic components (grey bars above and below the dense silt unit) identified. Facies descriptions and interpretations are shown to the right of the smear slide images.

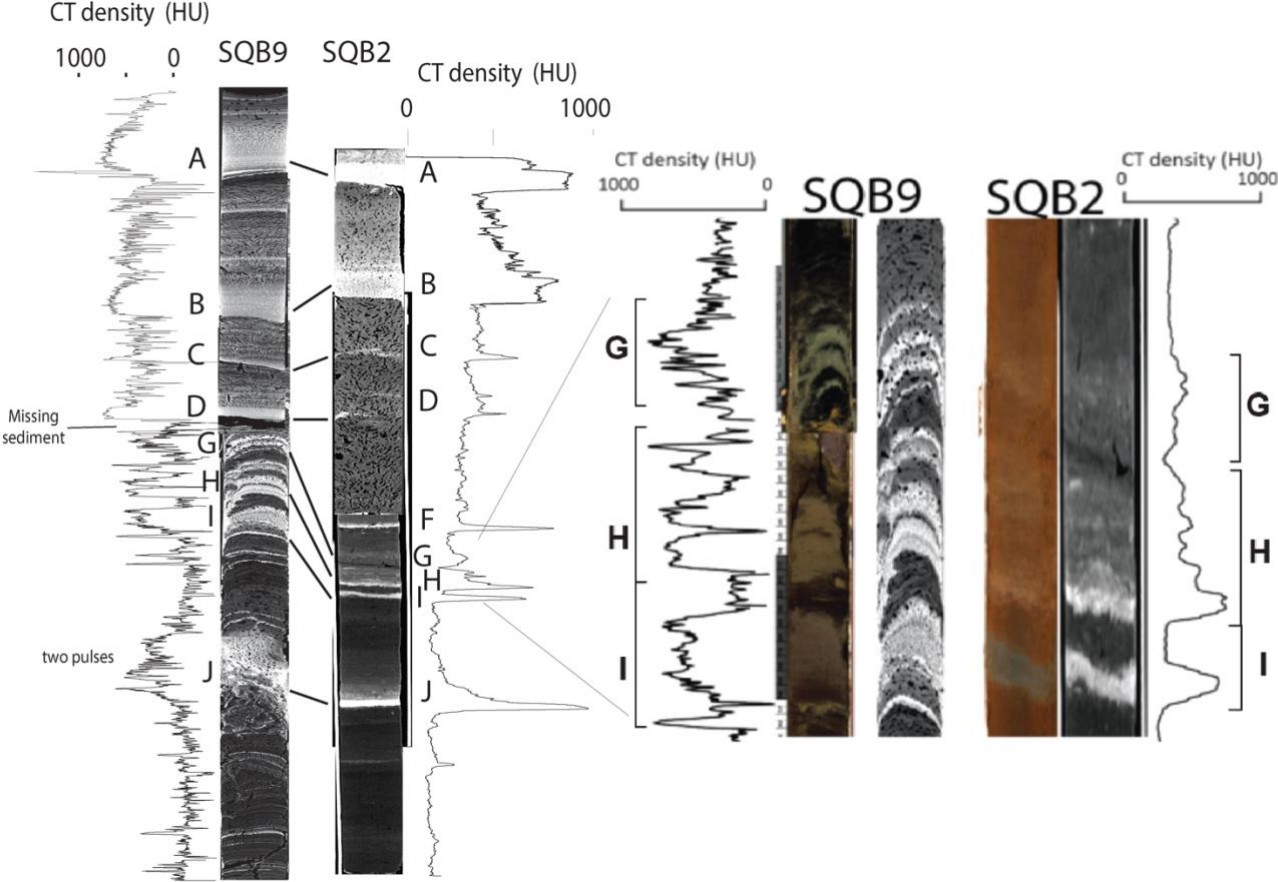

Figure 21. *Comparison of deposits G, H and I from shallow and deep cores.* A comparison of deposits G, H and I from SQB2 to G, H and I from deeper water core SQB9 (selected because it is at the lake depocenter and less likely to be missing sediment due to erosion). Deposit J has two pulses in the depocenter core SQB9, but not in the shallow water core SQB2, suggesting two channels feed the depocenter. There are more than two pulses in deposit H in the deep water core SQB9 suggesting that the multiple pulses are not the result of different travel times along multiple channel systems, but must be the result of multiple ruptures. See Discussion section for further analysis and interpretation. Note missing sediment in the image due to coring disturbance between deposits D and G in core SQB9.

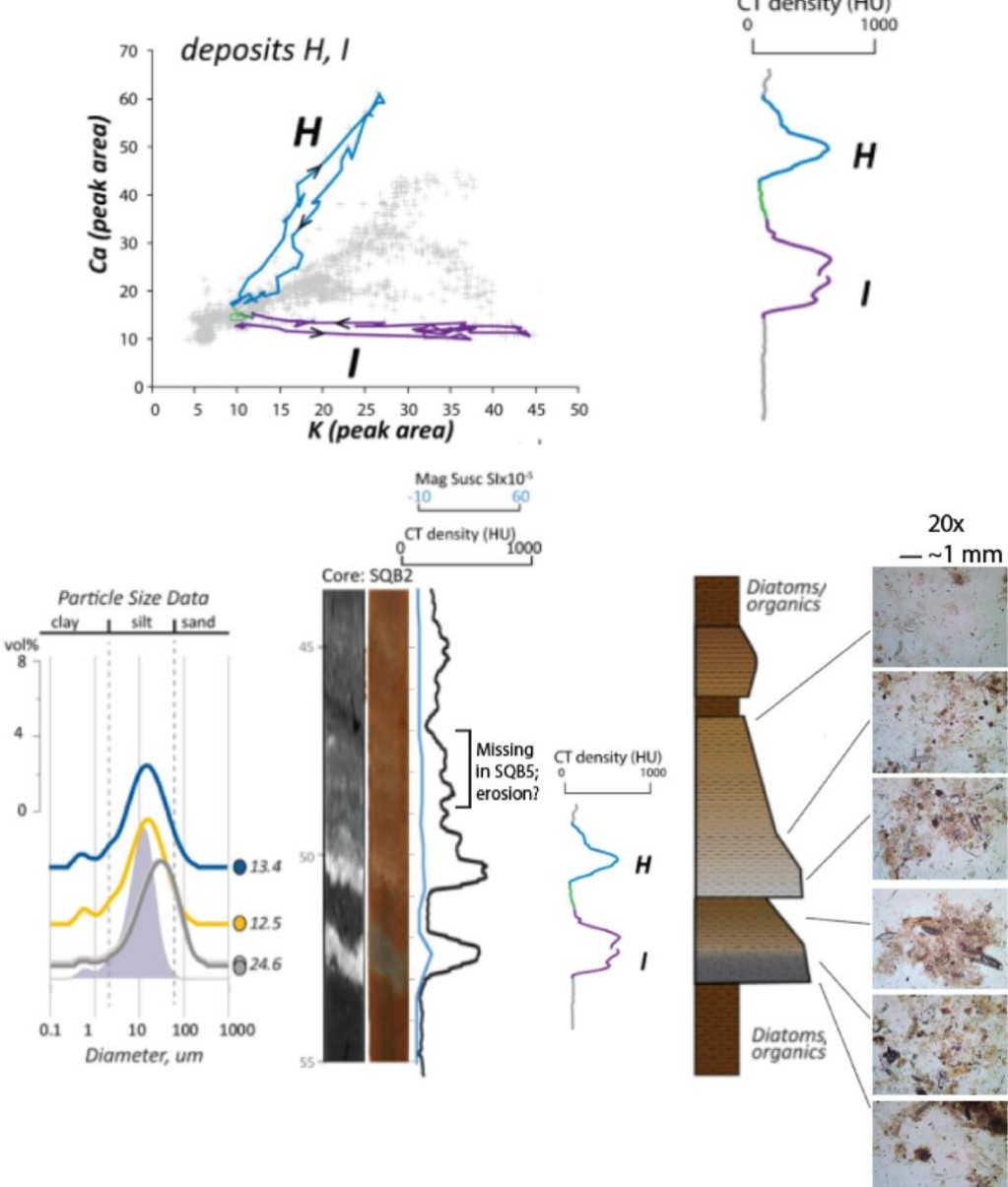

Figure 22. *Characteristics of disturbance event deposits H and I.* These deposits appear to be triggered closely spaced in time because sediment composition does not stay at background levels, but deposit I instead transitions immediately into deposit H. Top: XRF loop for H (light blue) is clockwise increasing along the calcium axis, while the loop for I (purple) is counterclockwise along the potassium axis, suggesting different depositional mechanisms. The purple shaded particle size data is from deposit J for comparison to the particle size data from deposits H and I. Note that the horizontal axis for the cartoon of the deposit sequence (bottom right) represents sediment radiodensity.

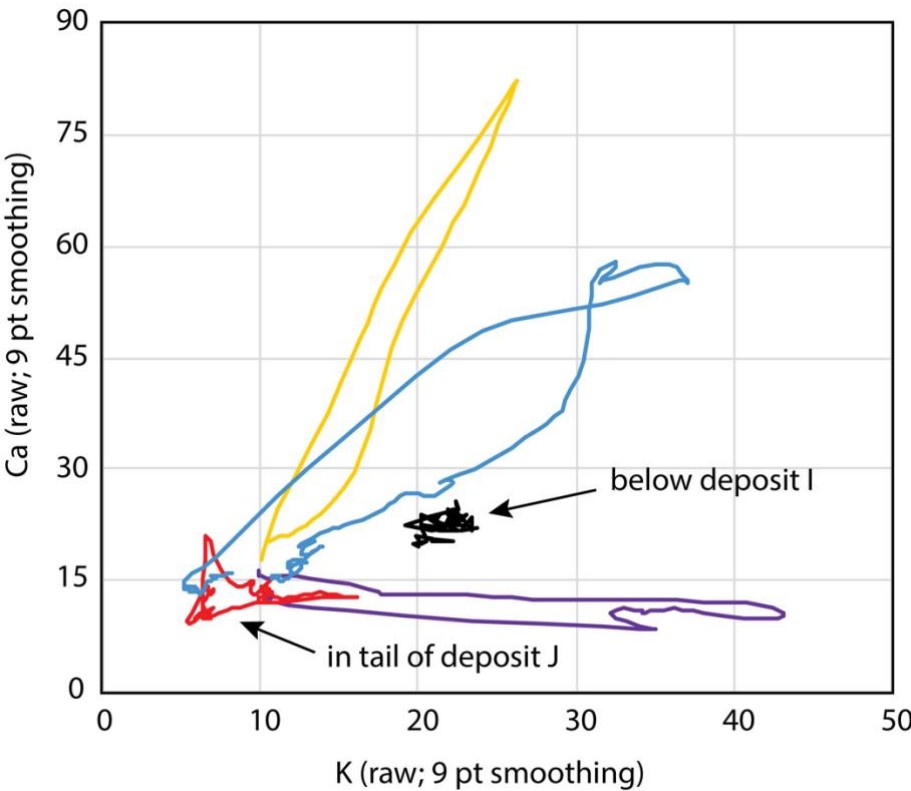

Figure 23. *A test of the methods.* The use of XRF x-y plots demonstrates how flood deposits (black trace) can be differentiated from earthquake traces (red trace). Blue trace = deposit J, yellow trace = deposit H, purple trace = deposit I.