# Peer review of "Sedimentary record of historical seismicity in a small, southern Oregon lake"

_EGUsphere, 2023_

## Referee Comment (RC3)

**Sedimentary record of historic seismicity in a small, southern Oregon lake**

**Ann E. Morey** ( ✉ ann@cascadiapaleo.org )

Cascadia Paleo Investigations    https://orcid.org/0000-0002-8702-2581

**Mark D. Shapley**

Continental Scientific Drilling Facility, University of Minnesota School of Earth and Environmental Sciences and The National Science Foundation, 116 Church St SE, Minneapolis, MN 55455

**Daniel G. Gavin**

University of Oregon, Eugene, OR

**Alan R. Nelson**

Geologic Hazards Science Center, U.S. Geological Survey, Golden, Colorado 80401 (retired)

**Chris Goldfinger**

Oregon State University, Corvallis, OR
* * *
Research Article

**Keywords:** lacustrine paleoseismology, Cascadia Subduction Zone, paleohydrology

**Posted Date:** June 9th, 2023

**DOI:** https://doi.org/10.21203/rs.3.rs-1631354/v2
* * *
**Sedimentary record of historic seismicity in a small, southern Oregon lake**

[revised manuscript text omitted]

70 the California Geological Survey) for the United States (accessed May 2019, https://earthquake.usgs.gov/hazards/qfaults/) identifies few active regional faults, however the simplified Cascadia forearc fault model of Wells et al. (2017) identifies a fault along the Klamath River, just south of Squaw Lakes (Figure 1, right).

The largest historic earthquake in Oregon was a ~M7 earthquake that occurred on November 23, 1873 (Wong, 2005). This earthquake was strong enough to topple chimneys in Jacksonville, OR, 15–20 km east of Squaw Lakes (Ellsworth, 1990) and

75 has been interpreted as an intraplate earthquake primarily because of the lack of reported aftershocks (Wong, 2005). Numerous investigations of felt reports published in regional newspapers suggest the intensity centre was located ~10 km inland from the coast, from just south of Cape Blanco, OR, to Crescent City, CA (Bakun, 2000; Toppozada et al., 1981). Brocher (2019) suggested the intensity centre was located roughly halfway between Grants Pass, OR, and the coast, ~ 75 km west of the study site.

80 Both Upper and Lower Squaw Lakes are surrounded by Condrey Mountain Schist ("lake bedrock," dark grey unit in Figure 2), a heavily foliated quartz-muscovite schist (Hotz, 1979), that has been described as failure-prone (Coleman et al., 1983). The northern portion of Lower Squaw Lake is fed primarily by Slickear Creek, which is almost entirely located in a unit mapped as metavolcanics sediment and flows (andesite) and quartz-diorite ("watershed bedrock," orange unit shown in Figure 2). The watershed rocks have a different composition and are more resistant to erosion than the schist that surrounds the lakes

85 and most of the Squaw Creek watershed.

**1.2.2 Climate**

The Klamath Mountains ecoregion experiences a Mediterranean climate characterized by hot dry summers and wet winters (Sleeter & Calzia, 2012). The wet winters are the result of equatorward shifts in midlatitude storm tracks during the winter months (Swain et al., 2018). The latitude at which this shift occurs is variable through time and can result in extreme shifts

90 between flooding and drought (Horton, et al. 2015). Atmospheric rivers are narrow pathways of tropical moisture that are regionally important because they provide a large amount of rainfall and stored water in the form of high elevation snowpack to the region (Goldenson et al., 2018). Sustained atmospheric river events can produce extreme flooding (Safeeq et al., 2015), such as occurred during the 40-day event that occurred in 1861–1862 (Engstrom, 1996).

**1.2.3 Sediment transport**

95 For the post-logging era (1930–present) the dominant influence on sediment accumulation rates identified from the sedimentary record from Upper Squaw Lake is rainfall (Colombaroli & Gavin, 2010; Colombaroli et al., 2018). Prior to that, the largest accumulation rates are related to postfire erosion and possibly earthquakes, as part of complex feedbacks (Colombaroli & Gavin, 2010; Colombaroli et al., 2018). Slope failures and slumps around Lower Squaw Lake are common on the steep hillslopes and were observed as changes in the landscape and vegetation, suggesting possible instability during

100 shaking. Rainfall, the dominant influence on slope wash from hillslopes to streams in upland regions (Lamoureux, 2002; Zolitschka, 1998), occurs primarily during the wet season from November to April (Sleeter & Calzia, 2012). Stream bank outcrops suggest occasional extreme, erosive flow. Snowmelt floods, which occur when rain-on-snow events melt snow in the upper reaches of the watershed, may also introduce pulses of sediment into the lake. Flash floods have been observed to transport and deposit sand to boulder-sized particles near Slickear Creek to the north near the lake margin where vegetation is

105 dense (Bert Harr, personal communication, September 2015; landowner).

*I can't follow this*

**1.2.4 Lake and watershed characteristics**

Lower Squaw Lake is a long, narrow (area = 22.6 ha), deep (~40 m) lake at 915 m elevation. The Slickear Creek watershed to the north is smaller (7.7 km$^2$) than the Squaw Creek watershed to the east (40.2 km$^2$). The level of Lower Squaw Lake was raised above its natural level by ~5 m in 1877 when a dam was built to increase water pressure for hydraulic gold mining

110 (Jacksonville Times, September 25, 1878). The ~0.5 km long Slickear Creek delta is composed of coarse sand, cobbles, and a few boulders near the shore of the lake where vegetation is dense. The delta has been built rapidly by floods that have occurred every ~10–20 years, occasionally depositing a thick layer of coarse sediment over the entire delta (Bert Harr, personal communication, September 2015; see Table 1). Most of the water flows into the lake from the north as subsurface flow; however overland flow occasionally occurs along the small (a few meters wide) but incised stream channel on one side of the

115 delta.

*Why not both in some unit?*

Upper Squaw Lake is a small (7.3 ha), shallow (14.2 m) lake at ~930 m elevation with a capacity of ~564,000 m$^3$. Upper Squaw Lake drains a large watershed (40 km$^2$) of steep terrain (~1,020 m relief), and the creek flows throughout the year through Squaw Creek into the southern portion of Lower Squaw Lake near the dam. Although the terrain is steep throughout much of the watershed, the proximal ~2.0 km near Lower Squaw Lake becomes gently sloping, and the creek meanders and

120 branches as it nears the lake, then enters the lake over a delta front composed of angular, well-sorted, medium to coarse sand. Groundwater likely flows through the delta, as water-tolerant trees and shrubs are present. Two sediment cores with overlapping sections were taken from near the centre (water depth of 14.1 m) of the lake and were used to create a composite depth profile of a high-resolution 10 m sediment core containing a record of watershed-sourced deposits over the past ~2,000 years (Colombaroli & Gavin, 2010; Colombaroli et al., 2018).

*(Squaw Creek)*

*Methods?*

[revised manuscript text omitted]

Lithostratigraphic correlation takes advantage of the characteristics of both the sequence pattern of disturbances as well as the characteristics of the petrophysical properties through the disturbance deposits themselves. The petrophysical properties of the

165 disturbance deposits can be considered fingerprints of the time history of deposition of the disturbance deposit (Goldfinger et al., 2008, 2013; Patton et al., 2015), and individual disturbances from independent records have been shown to correlate over long distances even though they are from different depositional settings (Goldfinger et al., 2012; Morey et al., 2013). Density is the highest resolution petrophysical property data type, has been shown to be the most sensitive property to changes in fine-grained inorganic disturbances (Inouchi et al., 1996), and does not as much of an edge effect that magnetic susceptibility data

170 has (because of the higher resolution), therefore the high-resolution CT density was heavily relied upon for correlation.

**2.4 Sediment provenance data**

We used x-ray powder diffraction spectra (XRD) to determine the mineralogy of the two endmember bedrock types. XRD allows qualitative and semiquantitative analysis of the mineralogy of sediments and rocks by measuring the diffraction properties of their mineral components. We interpreted the results using the automated pattern-matching routine in Jade

175 Software (http://ksanalytical.com/jade-9/), which compares the relative peak heights and areas from unknowns to those from samples of known mineralogy contained in the software database.

We acquired x-ray fluorescence data (XRF) with an ITrax core scanner (Oregon State University) from downcore sediment at 0.4 mm intervals and from discrete samples of lake-margin beach sand and Slickear Creek streambed sand (source locations are shown as blue triangles in Figure 3, left). The XRF downcore data were used to determine the upper and lower boundaries

180 of each deposit in addition to identifying sediment provenance.

**2.5 Development of event-free stratigraphy and age-depth model**

Event-free stratigraphy. Rapidly deposited sediment was removed from the stratigraphic sequence to avoid misinterpreting it as being deposited at the same rate as background sediment. This event-free stratigraphy was created by identifying the disturbance deposit boundaries using XRF and estimating missing sediment at erosional contacts using relationships between

185 shallow and depocenter cores. XRD data from endmember rock samples were used to initially determine the best choice of elemental variables to use as XRF provenance indicators. XRF variability through the disturbance deposits was then used to

determine where deposit boundaries exist. Sharp increases in sediment density (higher HUs; lighter values), compared to lower density background sediment (lower HUs; darker values), indicate rapid deposition or reworking as described in Morey et al. (2013).

190   Radiocarbon samples and data. We sampled Lower Squaw Lake sediment cores for radiocarbon dating after splitting cores longitudinally. We removed macroscopic samples of fragile plant material (such as fir needles and buds) from the targeted horizons of undisturbed sediment, cleaned and dried them, then had them analysed by AMS (accelerator mass spectrometer) for radiocarbon. We selected the target horizons for sampling based on a suspected temporal tie point between the Lower Squaw Lake record and the dated sequence from Upper Squaw Lake. We did not acquire $^{210}$Pb and $^{137}$Cs data to calculate

195   sedimentation rates for the most recent section of the cores because the upper portions of the sediment cores contained two thick clastic units (lake-wide and of varying thickness) with evidence of erosion, which violates the dating method assumption of continuous sedimentation. We used the strong similarity in sequences between the upper and lower lakes to infer that the younger of these clastic disturbance deposits was deposited close to 1964–1965 (as shown in the Supplementary Data).

Age-depth model. An age-depth model was developed using a Bayesian approach using OxCal (v 3.4.2; Bronk Ramsey, 2017).

*[handwritten margin note: mention the lab]*

*[handwritten note: For $^{210}$Pb perhaps, but not for $^{137}$Cs $^{137}$Cs data would've been very useful to date the 1964–1965 deposit! => delete this reasoning]*

200   **3 Results**

**3.1 The historic record of extreme events**

Historic events with the potential to influence Lower Squaw Lake sedimentation are compiled from personal accounts (from landowners and U.S. Forest Service Rangers), published hydrologic data, regional historic newspapers, and Forest Service documents (Table 1). We did not include large land-use events (logging efforts and road building) or wildfire in Table 1

205   because these events require water to transport the resulting increase in available sediment into the lake, however extreme runoff from these types of events can cause debris flows (Wall et al., 2020). Homesteading began in the region when gold was found between 1850-1852 CE (Lalande, 1995).

**3.2 Sediment core locations and recovery**

This study investigated historic records from the northern cores (Figure 4; orange circles) near the Slickear Creek delta, which

210   is saturated near the surface. Sediment core locations, lengths and water depths are shown in Table 2. Several of the upper sections of the first Kullenberg cores were distorted during coring and stuck in the casing. Small adjustments to composite sediment depths accounted for core section breaks and minor distortion.

**3.3 Sediment facies**

Background facies. Background sediment is a very dark brown to black (Munsell colour: 2.5YR 2.5/1) organic-rich sediment

215   containing planktonic diatoms (~30%), particulate organic matter, and angular, poorly sorted medium to coarse silt (50–60%). Split sections change colour quickly (over a period of hours to days) from very dark brown (or black if the core was taken in

deep water) to a lighter brown, or slightly orange color and become concreted if exposed to air. Background sediment is stiffer and slightly lighter in colour deeper in the cores compared to the sediment in the upper portion of the cores. This change in upper sediment character was assumed to be due to historic land-use changes that began between 1850 and 1900 and the

220    installation of the dam in 1877. A shift in sediment density occurs below the two thick upper deposits in all cores. This horizon, indicated in the table as "inflection," refers to the change in sediment density that is suspected to reflect changes in sedimentation from around the time settlers cleared land (mid 1800's) and first dammed the lower lake (1877 CE). Loss on ignition and physical property changes associated with this shift are shown (data are from core SQB2; Table 3).

Disturbance facies. Ten disturbance deposits from the sediment cores were identified as abrupt increases in sediment density

[revised manuscript text omitted]

**3.4.10. Deposit types**

There are ten total disturbance deposits of the following types:

Type 1 deposits (deposits A and B, possibly C) are thick graded turbidites of mixed sediment composition. The disturbance deposits are composed primarily of coarse silt and have erosive bases containing organic matter at the base, fining upward with a thin clay cap and no perceptible grading of organic content in a tail. The disturbances do not show evidence of loading into the sediment below.

Type 2 deposits (deposits E and I) are schist turbidites with erosive bases and no evidence of loading into the sediment below. These schist turbidites have an organic-rich tail.

Type 3 deposits (deposits H and J) are composed of watershed sourced sediment containing a thick well-sorted medium silt layer with evidence of loading into the organic sediment below. There is no evidence of erosion at the base and only trace amounts or organics and diatoms. The silt grades upward with respect to organic content for several centimetres. The post-tail sediment contains a different community of diatom species and other organisms and is of lower CT density compared to background sediment.

Type 4 deposits (deposit G and unnamed deposit below deposit J) are ungraded deposits with mixed composition and rounded density profiles. Although the density of the unit is higher, the composition of sediment is similar to background, suggesting an increase in clastic content of mixed sources.

Deposits D and F are difficult to characterize because they are thin layers that are challenging to sample for grain size and composition. The base of deposit D is orange in colour, composed of watershed sourced silt (basal silt layer), and has three silt layers within slightly stiffer background sediment. Deposit F is slightly lighter in colour compared to background and 
[revised manuscript text omitted]

Deposit J. Deposit J is a medium-silt deposit displaying unusual grading characteristics: it is well-sorted at the base (becoming less-so upward) and lacks diatoms and particulate and degraded organics present in the basal silt of many of the other disturbance event deposits. The base of the deposit is sharp, but there is evidence of loading into the organic-rich sediment below. *→ should be part of the results*

To gain insight into the processes influencing deposition, we look to the XRF geochemical data (Figure 20) as scatterplots of potassium to calcium. Note that the visible base and top of the deposit do not start and end at the background ratio (represented by the initial positions of deposits G, H and I). This "gap" suggests that there is more at the base and/or the top of the deposit than is visible by eye. In other words, the basal silt is preceded by, and/or followed by, sediment that is part of the disturbance deposit. This sediment (blue line, centre top) suggests the preferential reduction in potassium (mica) prior the more obvious base of the silt which has a preferential initial increase in calcium (watershed sourced amphibole). Microscopic inspection identifies a thin micaceous silt layer followed by organic-rich sediment a few centimetres below the primary silt. Whereas this silt layer is indistinct in the northern cores, it is more obvious in the deeper water cores, especially SQB9. This suggests that the precursor is a small bypass turbidite (fine-grained turbidites that are formed when the coarser sediment bypasses the location; see Bouma, 2000) that is visibly present in deeper water cores but indistinct in the northern shallower water cores. Sediment above the primary silt unit (green line, centre top) suggests a long tail that returns to the initial position with respect to the background ratio. This tail is also apparent in the CT density (Figure 20, lower panel). The tail is followed by a very low CT density layer (a few cm thick) that visually appears to be part of the background sediment; however, the diatoms and other water column organisms are of different species (see smear slides in Figure 20, bottom right). The presence of this tail is supported by the loss on ignition data that shows a 30% decrease in inorganic content along the length of the tail. The XRF data suggests that grading through the dominant silt layer is complex; as grading progresses upward, the XRF pattern changes in way that appears to reflect the partitioning of entrained sediment into components slightly enriched in each elemental endmember as the deposit grades upward, first in the direction of Ca, then returning to background slightly depleted in Ca (or enriched in potassium).

*I agree with the tail, but I am not convinced by the deposit below*

*OR the transition is very abrupt*

*→ Should be part of the results*

The XRF data from the base of the silt upward through the tail is more complex in this deposit compared to the tails from other deposits (compare the XRF loop for deposit J shown in Figure 15 to those from the other disturbance deposits). This complexity

500    is unlikely to reflect multiple events through time because values don't ever go back to background. This supports the interpretation that this is a single deposit that formed as a result of an energetic event that partitioned particles in the water column during settling, and not the result of post-earthquake watershed removal of sediment through time.   *the base of is too coarse to result from this*

**Deposits H and I**. Deposits H and I form a complex sequence that formed between 1819-1875 CE (based on the age-depth model). A thin layer of deciduous leaves between deposits H and I suggests deposit I had time to settle ( to weeks), *from days*

505    but not enough time for interevent sediment to accumulate. We describe them together in order of deposition because they appear to have formed in response to the same event or two very closely spaced events.

Deposit I is a turbidite composed of disaggregated schist with visible mica fragments. It displays reversed, then normal grading from a medium-grained silt upward to form a short organic tail followed by a thin layer of deciduous leaves (forming the boundary between the schist turbidite and the silt from deposit H above). Deposit I is very similar to deposit E, a local lake-

510    margin slope-failure deposit, in that it is a turbidite formed of dark grey schist with large mica flakes. It contrasts with deposit E in that it is found in all cores, suggesting that deposit I was formed because of a lake-wide disturbance great enough to create a synchronously triggered mass-transport deposit composed of lake-margin schist. This disturbance is suspected to be the result of the 1873 CE Brookings earthquake because of timing and that shaking was strong enough in this region to cause chimneys to topple.   → *Subaquatic or terrestrial?*

515    In contrast, deposit H is composed of watershed-sourced sediment in core SQB2 and SQB5 (more so than any other deposit based on XRF; Figure 15). The deposit in SQB2 appears to have a long tail (Figure 21), but it is hummocky with respect to CT density instead of smoothly grading upward. SQB9, from the lake's depocenter, contains a temporal correlative to deposit H, but it is composed of multiple turbidites (Figure 21). This deposit in SQB2 is like Deposit G in that the northern cores contain sediment with a higher silt content compared to background; however, SQB9 contains turbidites (see Figure 21) and

520    the correlative in SQB2 does not. Deposit H in SQB2 is also different from deposit G in having a distinctive basal silt and tail deposit composed primarily of sediment sourced from the watershed. This suggests that deposits G and H have different origins.

The multiple turbidites forming deposit H in deep-water core SQB9 have several possible explanations: they could be the result of (a) synchronously triggered "amalgamated turbidites" (using the terminology of Van Daele et al., 2017, p. 77-78)

525    from a single earthquake producing multiple individual subaquatic landslides that travelled different distances (and therefore travel times) to reach the lake's depocenter (SQB9) depositing one over the other as they are deposited, (b) reflection waves or a seiche from a single earthquake producing multiple closely-spaced deposits, c) a mainshock and aftershock sequence for a single earthquake; (d) an earthquake with a complex source function, (e) post-earthquake retrogressive failure sequence, or (f) "turbidite stacks" (again, using the terminology Van Daele et al., 2017, p. 77-78) suggesting multiple earthquakes closely

530    spaced, but not synchronous, events in time.

→ *I think core SQB9 gets too much credit here. This is the only core where these multiple pulses are observed. In all other cores it is a single pulse, apart from perhaps SQB10 & SQB13, which are in the same area of the lake.*

*I don't agree, since SQB9 is in the depocenter, this is definitely an option*

Here we discuss these options as mechanisms to produce deposit H. Given that the lake is relatively small and there is only one main channel system, mechanism (a), which invokes separate subaqueous landslides travelling different distances from source locations around the lake, is considered unlikely. Mechanism (b) is unlikely because a seiche or reflection waves would produce deposits lake-wide instead of being obvious in only the deep-water cores. Mechanism (c), explaining the deposit

535    characteristics as a result of a mainshock and aftershock sequence seems unlikely because the deposits do not get smaller upward as would be expected if a combination of main and aftershocks. This explanation is also unlikely because the deposits are less likely to have been deposited over some time (longer than days) because the XRF scatterplot through deposit H in core SQB5 is a continuous loop (which does not go back to background until the end; Figure 22). A single event with aftershocks can sometimes be immediately after the mainshock and continuous for hours (such as with the 2011 Tohoku earthquake; Toda

540    and Stein, 2018); however, the 1873 Brookings earthquake was determined to be the result of an inslab event because no aftershocks were felt. Mechanism (d), which explains deposit characteristics as the result of a complicated source function, is possible however there is no information supporting that the 1873 CE earthquake was complicated. Likewise, mechanism e) may be plausible, but there is no information supporting this interpretation. The most likely (and simplest explanation) is that deposit H is a turbidite stack (mechanism f)) from multiple earthquake ruptures closely spaced in time (but not synchronous).

*background would at least work*

*main argument?*

545    This is supported by the presence of multiple deposits closely spaced in Upper Squaw Lake as well as in the deep-water site in Lower Squaw Lake. The interpretation is that an initial local earthquake caused the landslide dam to fail creating deposit I, allowing the lake to partially drain (below the shallow lake sites, SQB1,2, and SQB5), then subsequent earthquakes triggered a sequence of turbidites preserved only in the lake's depocenter. Although this scenario seems plausible, there is no definitive evidence to support it.

550    In summary, we suggest the following sequence to explain deposits H and I. An initial earthquake caused the landslide dam to fail, resulting in the partial draining of the lake and the formation of the lake-wide slope failure deposit composed of weathered schist which settled to form deposit I. Shaking from multiple very closely-spaced, nearby earthquakes then resulted in cyclic loading and liquefaction of the lake's delta, releasing sediment and groundwater into the lake mid-depth which settled to form deposit H. Whether or not these earthquakes are a result of crustal fault ruptures, inslab earthquakes, or small (<~M8) southern

555    Cascadia earthquakes is unknown.

*A lot of additional mechanisms are invoked here to explain the evidence, for which there is no evidence*

Deposit G. Deposit G, based on the age-depth model, settled between 1827-1892 CE. It is indistinct in SQB2, appearing slightly denser than background sediment, with maximum density at the deposit centre. The composition remains mixed with changes in density (Figure 15) throughout the deposit, but the slope is relatively flat toward K (relative to the slope of background sediment), similar to (but not as extreme as) deposit E. This deposit in SQB2 shows characteristics similar to the

560    waxing and waning flood deposit reported by St-Onge et al. (2004), who analyzed a sedimentary sequence from Saguenay Fjord, Québec, that was produced from a known historic earthquake followed by a flood (which was the result of a landslide dam breach). The age-depth model suggests that Deposit G could be the result of the dam failure in 1881, the 1873 Brookings earthquake, or the flood of 1861-62. The installation of the dam is unlikely to have caused this deposit because the dam is located in the south, far from SQB2. The 1873 Brookings earthquake caused severe shaking in the region (Ellsworth, 1990)

*If the 1873 CE earthquake was described, even to the detail that no aftershocks were felt. Why are these other ruptures not described? Also, why would all these earthquakes not be recorded in all other cores?*

565    and therefore is more likely to have caused large disturbance deposits H or I (see discussion above) and not deposit G. The proximity of the sample to the inflection in sediment density suggests a timing close to when the dam was installed, and land use changes occurred. Based on timing, it is suggested that this is the result of the dam failure in 1881 CE.

This interpretation is supported by the presence of the sequence of disturbances in SQB9 located near the lake's depocenter (Figure 21). There are three possible explanations for these amalgamated deposits: either they were 1) formed as a result of

570    several wall failures with differential travel times, 2) they are reflection deposits from a small seiche in response to the dam failure, or 3) they are the result of retrogressive landslides in response to destabilization of the landslide toe when the built dam failed. Because the lake is small with only one main channel and the formation of a seiche in response to the dam failure unlikely, we propose that the sequence is the result of a retrogressive failure sequence from repeated destabilization of the landslide toe when the dam failed until the landslide stabilized. We conclude that this deposit formed in response to the flood

575    and associated dam failure in 1881 CE and adjustments of the landslide in response to the dam failure.

Deposit F. Deposit F is a simple graded deposit which the age-depth model suggests settled between 1835-1908 CE. This deposit shows evidence of loading onto the organic sediment below in SQB2. The mineralogic composition is unknown, however the XRF data suggests a mixed composition enriched in K that does not vary with changes in density at the midpoint of the deposit. Lower Squaw Lake was influenced by a sequence of closely spaced events that began with the 1861-1862 event.

580    These include the failure of the dam as the result of the 1861-1862 atmospheric river event, the 1873 Brookings earthquake, the installation of the dam in 1877, the winter rain-on-snow event in 1881 (which caused the newly installed dam to fail), a large flood in 1890 (#2 of 3 based on stream gage data), and a smaller flood in 1892. Given that the 1890 flood is the largest in this set, it seems possible that Deposit F is the result of the flood of 1890, although it could be the result of an unidentified earthquake observed in offshore Trinidad cores dated to ~1830's (Goldfinger et al., 2019).

*[handwritten margin note: → this would be inconsistent with previous correlations]*

585    Deposit E. This deposit is only found in core SQB5, which is located on a steep slope. There is no age data for this deposit because it cannot definitively be identified in the chronology core SQBss/1/2 composite, however the time of deposition is between deposits D and F. Deposit E is predominantly composed of Condrey Mountain Schist (based on XRD and the dark grey colour of the deposit with visible mica flakes). The XRF data shows that changes through the deposit goes primarily in the direction of K, implying a relative increase in mica concentration. Slope failures are common at the location of SQB5,

590    indicated by the large amount of sediment missing between deposits B and J in the short cores identified as narrow and wide diameter short cores (see false colour image of core density at the top left of Figure 11). This deposit was likely the result of a local lake-margin wall failure because it is found in only one core, and the mica composition suggests a lake-margin bedrock source for the sediment (not a mixed source as would be expected from the disturbance of surficial sediment). An aseismic local wall-failure deposit could have resulted from heavy winter rains (like the deep-seated slope failure that occurred in

595    response to heavy rains during the winter of 2016). Alternatively, it is possible that deposit E is a local wall failure that resulted from an earthquake, however it is unlikely because an earthquake is more likely to disturb sediment at more than just one location.

*[handwritten margin notes:*
*→ subaquatic or onshore?*
*→ Seems to be the case for many events on this core location*
*→ were these recorded in the lake?]*

Deposit D. The age-depth model suggests that deposit D settled between 1870-1940 CE. Deposit D is indistinct and unusual in many ways. It is a sequence of three silt units (the lower-most is thickest and visually obvious) within a stiff layer of organic-rich silt (in SQB2). This deposit has a small counterclockwise loop in XRF (Figure 15) and the lower silt unit is a simple graded unit. This basal silt is orange in colour and is fine-grained and well-sorted. The XRF and XRD data suggest that although the majority of the deposit is of mixed composition, with preferential enrichment of watershed-sourced sediment at the deposit base, there is some portioning of sediment in the direction of Ca in the middle of the deposit. This could be the result of an interflow flood deposit containing watershed-sourced sediment. It is unlikely to be the result of the 1873 Brookings earthquake because regional shaking was strong and deposit D is indistinct (except for the thin layer of orange silt at the base). The age-depth model suggests that deposit D may be the result of flood events or the 1906 San Andreas earthquake. Given that there are 58 cm (event-free) over the past 126 years (based on the location of the inflection in sediment density), and there are 14 cm between this inflection and deposit D, this makes the age of deposit D: the date of inflection (~1850 to ~1880) + 30 years = 1880 to 1910 CE. This is very close to the time of the 1906 CE San Andreas earthquake or the 1890 flood. The presence of a disturbance event deposit from the ~M7.9 1906 CE San Andreas earthquake seems plausible because felt reports from the region suggest MMI values of ~IV in this region (Dengler, 2008). The results, however, are inconclusive.

Deposit C. The age-depth model suggests deposit C settled between 1880-1950 CE. Deposit C is a normally graded unit with an unknown composition that becomes thinner with distance from Squaw Creek (see below deposits A and B in Figure 5), suggesting it is the result of a flood. The most likely events to produce this deposit (based on the size of the event) are the third largest flood of five (which occurred in 1955) or the flood in 1927 associated with a debris dam failure. Given that the average sedimentation rate is ~2.5 yr/cm and the interevent sediment thickness between Deposits B and C is 18 cm (after accounting for erosion; top left Figure 8), it is most likely that deposit C is the result of the 1927 flood with debris dam failure, however this remains uncertain because the sedimentation rate is highly variable in the lake. No sediment provenance data exists for this deposit; however, in core SQB5, this deposit is brown in colour, similar to the lower halves of Deposits A and B. Although outside the time range, there is the possibility that it could be the result of the large flood in 1955 CE or the atmospheric river event in 1861-62 CE. The 1861-62 flood is considered unlikely, however, because deposit C was deposited well after the inflection point in CT density that is assumed to be the result of land use changes in the mid-late 1800's.

Deposits A and B. Deposits A (deposited between 1980-2013 CE) and B (attributed to the 1964 flood based on comparison to Upper Squaw Lake) are 5-20 cm thick, depending on location in the lake, with lake-wide extent and similar characteristics (Figure 6). They have sharp bases with sediment likely missing below in all cores other than the deepest water cores (SQB9 and SQB10), contain basal sediment with rootlets and degraded organic matter, and are coarse-grained, normally graded deposits. These characteristics suggest they are the result of erosive turbidity currents. Although the deposits are quite similar to one another, the base of deposit A (which is incomplete in core SQB5) is composed of calcium-rich coarse silt at the base, whereas the base of deposit B is composed of potassium-rich coarse micaceous silt.

Deposits A and B were most likely deposited in response to large flood events because the most recent events are the two largest flood events that occurred in 1997 and 1964. Multiple first-hand reports describe the nature of the extreme flood of

1997 in the vicinity of Lower Squaw Lake: A landowner described the flood as having transported watershed-sourced beach sand from one end of the lake to the other (B. Harr; June 2015), and U. S. Forest Service employees (personal communication, P. Jones; December 2019; J. McKelligott, December 2020) described debris caught at the dam that caused the lake level to rise a few feet above the maximum water level. Water was seen shooting 10 feet out of the spillway and caused damage to the gate. At Applegate Reservoir, a few kilometres downstream from Squaw Lakes, water was flowing over the earthen dam and observed to undercut surficial slope sediment, causing slumping into the reservoir (P. Jones, personal communication, December 2019). The extreme nature of this flood, relative timing compared to the 1964 flood and observations of beach sand suggest that the 1997 flood produced deposit A, the uppermost deposit in the record. There were no other disturbance events around this time. Because there are no other disturbance deposits with similar characteristics downcore, flood events are either more extreme than in the past, or the supply of readily mobilized sediment has increased (which is likely given logging contributions to sediment), or both It is also possible that the built dam is more likely to trap debris and elevate the lake level in response to extreme flooding than the natural landslide dam.

→ is B related to 1964?

**4.3.2 Organic-rich tail deposits**

The most likely earthquake deposits are the sequences formed by deposits H and deposit J. These have the following similar characteristics:

1)      They have tails enriched in watershed-sourced sediment displaying organic grading.

2)      The grain size distributions from the silt upward into the tail contains a dominant medium silt and a smaller percentage of finer-grained (fine silt and clay) particles.

3)      The tail is followed by normal background sedimentation that has very low density and contains a different composition and size of water column organisms.

We hypothesize these organic-rich tails, particle size distributions, and the post-deposition change in community structure may be diagnostic of earthquake-triggered deposits. To understand their structure more fully, we used regression to describe the relationship between CT density and inorganic content (Figure 23) using data from SQB1/2 (and surface sample) and SQB14 to determine if the tail deposits display organic grading. To do this, we estimated the inorganic content of the sediment from the measured CT density data using the equation:

$$\% \ INORG = -2518*CT + 98.695 \ \text{(which explains 97\% of the variance)}.$$

The regression explains 97% of variance. The residuals demonstrate that the correlation between organic content and CT density data is very high for CT values greater than 300 HU but deviate from the relationship below this. We suspect that the correlation breaks down due to imperfect registration between LOI and CT density data, or because % calcium carbonate was not included with the inorganic content data. We used this equation to estimate the percentage of inorganic content in core SQB5 (Figure 24). This figure clearly shows a reduction in inorganic content upward from the denser silt layers for deposits H and J.

Not really needed

21 There seems to be a group of samples that have low radiodensities despite a hig in organic content. Can these samples be located? Perhaps cracks?

It is suspected that the source of the watershed-sourced silt in both deposit J and H is the result of liquefaction of the lake's large delta. Sediment loading the lake margin in the north is of mixed composition and therefore shallow slope failures are likely to be composed of the same mixed composition. Slickear Creek transports water aboveground until it reaches the upper reach of the delta, then flows as groundwater to the lake through the coarse-grained delta deposit. The deltaic sediment comprises a wide size range of particles (from gravel to sand) formed by flash floods, and as a result is stratified to some degree. It is suspected that liquefaction or settling of the delta matrix causes the release of fine particles from the delta to explain the watershed composition of deposits J and H.

*??? Why not simply a delta failure?*

**4.3.3 Inferred sequence in response to sustained, not necessarily strong, ground motion**

In summary, we suggest the following scenario to explain the sedimentological, physical, and geochemical properties observed in Deposit J (Figure 25). Initial shaking disturbed lake-margin sediments generating a small turbidite which bypassed SQB2 but is present in the depocenter core SQB9. Subsequent sustained ground motion caused liquefaction or the release of fluids from the lakes' large subaerial delta, forcing fine particles and groundwater out from its coarse-grained matrix. This resulted in the release of watershed-sourced sediment into the lake near the thermocline (similar to an interflow) where the denser silt settled out of the water column first while the platy mica grains remained in suspension longer due to a combination of surface area, density, and turbulence from a possible internal wave. As shaking slowed, the mica-rich schist sediment would begin to settle out of suspension. These events may have caused the water column to become stripped of organic matter with diatoms and fine sediment trapped in it, possibly resulting in a collapse of (or at least change in) the lake ecosystem.

*→ Why are all these new mechanisms invented when there are plenty known mechanisms that can explain the observations*

**4.4 Interpretations: attribution of deposits to historic events**

There appear to be three types of flood deposits. The first type is represented by deposits A and B. They are thick turbidites with high magnetic susceptibility and density, but with a lower magnetic susceptibility in the lower half which is brown in colour from the organic matter entrained in the base. These show evidence of erosion. The second type of flood deposit is an interflow deposit which is a simple graded silt unit exemplified by deposit C. This unit has a wavy discontinuous base in the northern cores and is thicker to the south. The third type of flood deposit is represented by deposit G which displays reverse then normal grading.

Three types of earthquake deposits are suggested by the data. The first type is represented by deposit J. Deposit J is a complex sequence with an initial bypass turbidite followed by a watershed-sourced silt and a long, organic-rich tail. The second type of earthquake deposit, represented by deposits H and I, is also complex with an initial thick turbidite sourced from the schist and followed by a watershed-sourced silt and a long, organic-rich tail. The third type of earthquake deposit suggested by the data is a simple graded turbidite deposit, represented by deposit D and possibly F. The silt units show evidence of loading on the organic sediment below.

**5 Conclusions**

695     The setting at Lower Squaw Lake, Oregon, (~180 km inland of the deformation front in Cascadia) provided a unique opportunity to determine how seismically generated disturbance deposits can be differentiated by elemental and grain-size structure from deposits from other types of disturbances, such as floods. Based on these results, disturbance types can be differentiated as follows:

Flood deposits. Flood deposits are highly variable in character; however, all have counterclockwise XRF grading patterns.
700     Extreme floods with high water produce thick turbidites like deposits A and B, with erosive bases containing organic matter and a thin silty-clay cap. Possible interflow floods produce simple graded deposits similar to deposit F. Flooding, possibly with a dam failure, results in disturbances like deposit G, with a simple reverse then normal grading in shallow water cores, but multiple disturbances (possibly retrogressive landslide failures in response to destabilization of the landslide at the toe due to the landslide dam failure) in deep water cores.

705     Earthquake deposits. Earthquake deposits identified in this study are of the following types. The deposit suspected to be the result of the 1700 CE Cascadia earthquake is a complex sequence with an initial faintly expressed turbidite followed by a watershed-sourced silt with load structures at the base. This silt is very well-sorted and pure at the base, then grades upward into a long, organic-rich tail. The deposit suspected to result from the 1873 CE Brookings earthquake is also complex but has a lake-wide wall-failure turbidite at the base followed by a watershed sourced silt, similar to that of the 1700 CE Cascadia
710 earthquake. This deposit is different in that there are multiple deposits in the deep-water cores, similar to deposit G that included the failure of the landslide dam.

    We conclude that it is possible to distinguish crustal and plate boundary earthquakes, and flood deposits, using the sedimentological characteristics and provenance data at Lower Squaw Lake. These results hold promise for the use of small lake records throughout Cascadia to be used to improve our understanding of Cascadia earthquakes, including the potential to
715 infer ground motions inland in the forearc where the greatest population centres, and potential damage, exist.

[revised manuscript text omitted]

[1]See Figure 12. [2]Brown = 2.5Y 3/2, Light Gray = 2.5Y 4/1, Dark Gray = Gley 2 4/5PB, Orange = 5Y 4/1. Variations in colors through deposits were visibly obvious but frequently difficult to differentiate from one another using Munsell color charts. [3]Magnetic susceptibility variability was compared to the variability in CT density. Note that magnetic susceptibility data is influenced by surrounding sediment (exponential decrease with distance), and therefore the magnitude can be influenced by the thickness of the unit if thin (~1 cm or less; see Figure 13). The most diagnostic features of the earthquake-triggered deposits is that they were determined to have a watershed composition (by XRF

**Figures**

[Figure]

**Figure 1**

[revised manuscript text omitted]

by the letters A-J.

[Figure]

**Figure 14**

*X-Ray Fluorescence data for core SQB5, Sec. 1.* Left: RGB imagery, CT grayscale imagery, CT density

(black trace), magnetic susceptibility (light blue line), and raw XRF data (peak area) for eight elements are

shown in

colored lines. Right: Calcium (Ca) and potassium (K) from watershed and beach sand samples do not overlap,

whether data are represented as raw counts (peak area) or normalized by titanium or strontium. This suggests that

these elements (Ca and K) may be useful to identify sediment provenance end members.

[Figure]

**Figure 15**

*Calcium and potassium downcore XRF data and scatterplots for deposits B-J in core SQB5.* Each of the

gray bars represents the event deposit boundaries for units B-J based on the beginning and end of scatterplot loops

(raw, unsmoothed data) at right. The direction of elemental composition from the base of the deposit upward is

identified by the black arrows. Note that some of the scatterplots show clockwise evolution and others show

counter-clockwise evolution of calcium and potassium through the deposits. See the Discussion section for a

detailed interpretation of the scatterplots.

[Figure]

**Figure 16**

*Age-depth model for core SQB1/2/ss composite.* The age-depth model for composite core SQB1/2/ss was

developed using event-free sediment accumulation as described in the Methods section. The depths of the event

bases were used to extract age ranges and median values in calendar years for disturbances A–J (sample RC #2 was

taken from the base of deposit J) and other unlabeled events in the sequence. The known timing of the 1700 CE,

1873 CE and 1906 CE earthquakes are identified by red vertical lines. Acomb = 64.5, and Aoverall = 64.7. A second

version of the age depth model was created with a C_date of 1700 CE used in place of the radiocarbon sample #2

resulting in a higher Acomb and Aoverall (both of which are 80). Note that Aoverall refers to the product of the

individual agreement indices, and Acomb refers to the test if distributions can be combined.

[Figure]

Figure 17

*Options for radiocarbon calibration assuming samples died close to the time surrounding sediment was*

*deposited for deposit J.* Option 1 is the result of the OxCal P_sequence age-depth model presented in Figure 16

(with RC #2), and Option 2 is the alternative. Because the event-free sediment thickness is 71 cm (representing

hundreds of years, Option 1 is considered most likely. See text.

[Figure]

**Figure 18**

*XRF scatterplots; results and interpretation.* Colored lines represent the individual disturbance deposits

also shown individually in Figure 15. A) Variability through the disturbance deposits in core SQB5 can be expressed

in terms of the endmembers K and Ca (shown here after smoothing). Each of the deposits displays variability that is

unique and related to the initial composition of the disturbed sediment, sediment partitioning, and/or additional

inputs during deposition. B) This cartoon demonstrates how patterns in the data may be interpreted as changes in

composition of provenance indicators from the base of an event to the top as it evolves during deposition. Arrows

show the direction with depth from the base of the deposit to the top as in Figure 15. Each deposit begins at the

initial background ratio between Ca and K and increases and decreases along a distinctive path before the deposit

ends as composition returns to the initial background ratio. A suggested explanation for these patterns is that they

reflect the relative amounts of each variable (calcium and potassium), and a third implicit variable related to

sediment density. C) Key to colors represented by the data shown in A); the vertical axis is K (raw counts), and the

horizontal axis is the step number in the sequence downcore.

[Figure]

Figure 19

*XRF data normalized by silicon and scaled by CT density.* The smoothed, raw data (A) were normalized

by silicon (B), then scaled by CT density (C). The similarity between (A) and (C) suggest that the relationships

between variables calcium and potassium for each of the disturbance event deposits A-J (D) are different and reflect

both sediment provenance and sediment density.

[Figure]

**Figure 20**

*Deposit J is a complex sequence with cryptic components.* Upper panel: The gap that exists between the

initial and final ratios in the XRF scatterplot for deposit J can be filled in by including the sediment below

(shown in

blue) and by interpreting the upper portion to be a long tail (shown in green). XRF calcium and potassium

confidence intervals (inset in blow-up of the scatterplot, center top) were produced by repeatedly measuring the

same section of core. The cryptic components of deposit J meet close to the initial positions of deposits G, H and I.

The base of the visible silt unit is slightly enriched in calcium relative to background and is very well-sorted and

fine-grained. This unit appears to "bleed down" from the clastic base into the very fine, organic-rich sediment below

suggesting loading which can result when dense sediment abruptly settles onto sediment that is less dense. The

cryptic tail of the deposit is followed by a change in the size and types of diatoms and other components suggesting

a post-earthquake change in water column organisms, possibly as a result of flocculation. Note that the data in the

scatterplot are not smoothed to show the true variability of the data. Note also the pulse of watershed-sourced

sediment (high in calcium, low in potassium) identified by the gray line above the cryptic tail of deposit J – could

this be the result of a post-subduction earthquake aftershock, another subduction earthquake, or large crustal or

inslab earthquake? Lower panel: Descriptive core data and imagery are shown for deposit J and surrounding

sediment with the visually cryptic components (gray bars above and below the dense silt unit) identified. Facies

descriptions and interpretations are shown to the right of the smear slide images.

[Figure]

**Figure 21**

*Comparison of deposits G, H and I from shallow and deep cores.* A comparison of deposits G, H and I

from SQB2 to G, H and I from deeper water core SQB9 (selected because it is at the lake depocenter and less likely

to be missing sediment due to erosion). H and I may be the result of a landslide dam failure and either reflection

deposits (from a seiche), post-dam-failure turbidites arriving from several locations in the lake, or retrogressive

landslides after the toe of the landslide became destabilized after the landslide dam failed. See Discussion section for

analysis and interpretation.

[Figure]

Figure 22

*Characteristics of disturbance event deposits H and I.* These deposits appear to be triggered closely

spaced in time because sediment composition does not stay at background levels, but instead transitions immediately

into deposit H. The low-density layer between H and I is an organic-rich layer containing a high percentage of

organics, with a thin layer of plant macrofossils just before the watershed-sourced medium-grained silt is deposited.

This suggests that H and I are two separate events because there was enough time for leaves to settle prior to the

deposition of deposit H. Top: XRF loop for H (light blue) is clockwise increasing along the calcium axis, while the

loop for I (purple) is counterclockwise along the potassium axis, suggesting different depositional mechanisms. The

purple shaded particle size data is from deposit J for comparison to the particle size data from deposits H and I. Note

that the horizontal axis for the cartoon of the deposit sequence (bottom right) represents sediment density.

[Figure]

[Figure]

**Figure 23**

*Identifying the relationship between % inorganics and CT density data.* A regression equation was

developed to estimate the % inorganic content from CT density data (top). This equation explains 97% of
the

variance. Some points are not well-described by this relationship, as seen in the residuals (bottom) as well as

inorganic content (ratio of inorganic to organic sediment) versus CT density. Note that the largest residuals occur

where CT density is lower.

[Figure]

Figure 24

*Estimated inorganic content, CT density and XRF ratios of calcium and potassium.* CT density (middle)

was used to estimate the inorganic content in core SQB5 (top) using the regression equation (described in Figure

23). Dark gray bars identify deposits enriched in potassium and blue bars identify deposits enriched in calcium. Pale

yellow identifies those deposits that display organic grading. Deposit D is too small to determine its grading

characteristics.

[Figure]

Figure 25

*Inferred mechanisms that result in the characteristics of deposit J.* 1) Initial ground motion (p-wave?)

triggers a small turbidite containing potassium-rich sediment which bypasses shallow water and is deposited at the

lake's depocenter. 2) This is followed by sustained shaking (s-wave?) resulting in liquefaction and the release of

groundwater and watershed-sourced calcium-rich sediment from the lake's large delta near the thermocline. 3)

Continued shaking sustains energy in the system, possibly creating an internal wave, resulting in the partitioning of

sediment in the water column. Sediment settles in density order as shaking wanes. 3) Shaking causes coagulation

and the formation of flocs which settle, stripping the water column of fine particles as they settle to form the long,

organic rich tail. 4) The water column becomes repopulated with primary producers and other organisms, returning

the lake to pre-earthquake conditions. 5)

**Supplementary Files**

This is a list of supplementary files associated with this preprint. Click to download.

- Moreysupplemendarydata.docx

---

## Author Response (AR1)

**SEE RESPONSES TO BOTH REVIEWER COMMENTS BELOW:**

**Maarten Van Daele comments to manuscript (bold), with author responses in italics.**
**AUTHOR-ADDRESSED REVISIONS IN THE VERSION SUBMITTED TO COPERNICUS ON APRIL 17, 2024 ARE IN RED.**

**It is clear that a lot of work and methods have gone into this impressive dataset and it contains a promising record. However, I do have concerns with some of the interpretations in the discussion, which can be summarized as: the authors invoke many poorly constrained mechanisms (e.g., lake level lowering, fine particles leaking out of a delta) and events (e.g. additional earthquakes in the historical part of the earthquake) too explain observations that can be more easily explained by widely recognized processes such as a delta failure. Also, some 137Cs dates would be really helpful to pinpoint the 1963/64 depth. My main comments are below and small comments are added to an annotated pdf that is attached.**

*Thank you for your helpful comments. There are no major disagreements with respect to the minor comments in the annotated pdf. The authors of this paper will correct the word "historic" to "historical." Author responses to the major comments can be found below.* ***DONE; Also changed is the name of the lake (an official name change).***

**There are (too) many figures in the manuscript. Please consider to move some to supplement.**

*Agreed. Some figures will be moved to the supplement.* ***Some figures have been removed, not moved to the supplement.***

**Methods. Ideally you present the XRF data as centered-log-ratio (CLR) transformed to reduce the closed sum effect. See Weltje et al. (2015), or application in Schwestermann et al. (2020). This is nowadays routinely done for XRF scanning data.**

*The authors understand and provide a lengthy description as to why the data are presented in raw form and show a comparison to the log-ratio method. Although we agree that to accurately represent the true geochemical composition of the data one needs to present the data as suggested (log-normalized) to reduce the closed sum effect, the objective here is to show the observed patterns in the raw data, demonstrate their relationship to other core data (density and magnetic susceptibility), and suggest an explanation as to why these patterns exist. Most important uses of the raw data were to 1) identify exactly where the deposit began and ended in the core, and 2) infer that deposit grading is a function of both elemental composition and variables associated with sediment density based on a comparison of normalized xy plots as a result of a comparison between normalized (scaled by CT radiodensity) and raw data.*

*Using methods to transform the data (such as the centered log ratio method of Weltje et al. (2015)) introduces noise and obscures the observed patterns (which we are trying to explain). This is clearly described in the manuscript. Furthermore, smoothing the high frequency noise (as in Schwestermann et al., 2020) is an additional transformation that adds to the uncertainty,*

obscuring the patterns. The goal is to be able to identify where the disturbance deposits begin and end, and how they evolve in elemental XRF space. Because the patterns in the raw XRF data correlate to other geophysical (e.g., density) and other compositional (e.g., % organic content as inferred by the CT radiodensity and measured Loss-on-ignition) data, it is considered a valid representation of changes in the sediment and a useful tool by which disturbance deposits can be differentiated from background sediment and mechanisms inferred in this study. That said, an approach for future research is to calibrate the XRF data using measured compositional data to get at the actual compositional changes through the disturbance deposits. This data could then be scaled by CT radiodensity data to more accurately reflect how the disturbance deposits evolve in composition as they settle.

*THE AUTHORS STAND BY THEIR DECISION TO USE THIS APPROACH TO EVALUATE THE DATA EVEN IF IT DOES NOT STRICTLY REPRESENT SEDIMENT GEOCHEMISTRY. THE REASONING IS DESCRIBED IN DETAIL THE TEXT; A REFERENCE HAS BEEN INCLUDED TO SUPPORT THAT THIS IS AN APPROPRIATE WAY TO REPRESENT THE DATA.*

**Discussion.**

**Deposit J. The tail related to deposit J was initially not included to the event deposit, event though from Fig. 14 it is pretty clear that there is a tail (Bouma Te division) that is indeed not included in the deposit. This tail should, however, be included already in the results, so that it can also be taken into account for the age model. Furthermore, I am far from convinced that the silt deposit below J is part of the same event. We know from comparison with well-described events (e.g., Van Daele et al., 2017; Wils et al., 2021) that a long muddy tail means a significant time lag of at least days to weeks.**

*Original author response:*

*Identifying the tail as the Bouma Te division implies that it is a turbidite, the result of a turbidity current. This is not the case as explained in the manuscript: the silt and the tail are part of the same event because the XRF data shows that sediment composition does not return to background until after the tail. Because the tail is part of the deposit that formed over a short time period (likely minutes to hours, depending on how quickly the flocs settle – which is less time compared to normal fine-grained sediment), they should not be included in the age model. The age model should only include normal background sedimentation, which is why it is so very important to know when a deposit starts and ends (see discussion about XRF data above).*

*Author corrected response:*

*The tail of the deposit was NOT included in the age-depth model, even though Maarten seems to have inferred otherwise. It would be good to know why he made this interpretation.*

*Other comments: Identifying the tail as the Bouma Te division implies that it is a turbidite, the result of a turbidity current. This is not the case as explained in the manuscript: the silt and the*

tail are part of the same event because the XRF data shows that sediment composition does not return to background until after the tail. Because the tail is part of the deposit that formed over a short time period (likely minutes to hours, depending on how quickly the flocs settle – which is less time compared to normal fine-grained sediment), they should not be included in the age model. The age model should only include normal background sedimentation, which is why it is so very important to know when a deposit starts and ends (see discussion about XRF data above).

*SINCE THE TAIL WAS NOT INCLUDED IN THE AGE-DEPTH MODEL, THERE ARE NO CHANGES THAT NEED TO BE MADE. Every turbidite identified as a possible Cascadia earthquake deposit in the downcore companion paper has the same thin silt layer just before the primary deposit, which supports the interpretation that it is part of the same deposit. It is now considered that the deposit J IS the Te or Te,Td portions of a turbidite, with some other process operating (for example an Internal wave) because of the complex grading through the deposit.*

**What about the deposit around 40 cm in SQB5 (Fig. 14), this also seems like an event deposit with a tail reaching until the base of event I.**

*This was not described because it was not originally visually identified in the record (deposits A-J) during the initial core description. This should be included, however, in the discussion as suggested. This deposit is a very thin silt deposit with a long tail that has some of the characteristics of a subduction earthquake deposit and therefore warrants a substantial discussion. THIS MAY BE A CASCADIA EARTHQUAKE DEPOSIT IN PART BECAUSE OF THE LONG TAIL AND IN PART BECAUSE OF THE XRF DATA.*

*THIS WAS INVESTIGATED AND DETERMINED NOT TO BE, USING THE XRF METHOD OF DEPOSIT EVOLUTION, A CASCADIA EARTHQUAKE, BUT RATHER THE RESULT OF A STORM, PROBABLY THE ATMOSPHERIC RIVER EVENT OF 1861/62 CE. The "tail" is not a tail at all, but reflects the land-use changes that began at around 1850 CE. This is described in a separate section near the end.*

**Deposits G, H, I. I have sincere problems with the proposed interpretations. A lot of new mechanisms are invented (e.g. line 550-555 and section 4.3.3), while there are plenty known mechanisms (rock avalanche, delta failure...) that can much easier explain the observed deposits.**

*The authors will revise lines 550-555 and section 4.3.3 appropriately to avoid "inventing mechanisms."*

*THIS HAS BEEN CORRECTED.*

**- I: do the authors interpret this as sourced from terrestrial or subaquatic slopes? If subaquatic, why did the 1700 CE earthquake not trigger any (!!!) failures on these slopes? Hence, I suggest the authors clarify in the text that this must be the terrestrial slopes.**

*We will clarify the interpretation of deposit I as sourced from subaquatic slopes; we are not sure from the comment above why Dr. Van Daele believes that deposit I must be sourced from the terrestrial slopes.*

*Regarding deposit J (inferred to represent the 1700 CE Cascadia earthquake): Recent evaluation of deposit J has slightly modified the interpretation to include a small slope failure deposit preceding the base of the silt unit and this will be included in the revised manuscript. This evidence includes a few grains of mica at the contact between the sediment below deposit J and the basal silt of deposit J. The authors interpret this to reflect a bypass flow from the shallow water (where the mica is virtually absent) to the deep water (where this mica silt unit is more obvious as a thin turbidite). The platy mica has a large surface area and would settle less quickly, staying in suspension and resulting in the water/mica mixture to be denser than water that becomes a gravity flow.*

*Note that there is also evidence not previously reported to support the interpretation that the silt units from deposits J and H are sourced from the delta and will be included in the revision. Watershed-sourced silt that is exposed to oxygenated water would have an orange color (likely iron oxide – rust). The silt that is watershed-sourced in these deposits is not orange in color and therefore has not recently been exposed to oxygenated water. This is evidence that the silt is sourced from within the delta where any minerals coating the grains would have removed by the delta's groundwater.*

***IT HAS BEEN CLARIFIED THAT DEPOSIT I IS A SUBAERIAL DEPOSIT AND THAT DEPOSIT J IS THE RESULT OF A SUBAQUEOUS DELTA SLOPE FAILURE.***

**The summary further includes a dam collapse and lake lowering for which no further evidence is provided. I have the feeling that a lot of additional mechanisms are invoked for which there is no evidence. In my opinion the authors make the story more complicated then it needs to be.**

*"Dam collapse and lake lowering": this will be reduced in importance in the discussion given there is no evidence. It will be presented as a potential mechanism that could explain the deposit, but the simplest explanation is the more likely (slope failure) for deposit I.*

***THIS HAS BEEN CORRECTED; the reference to a dam collapse has been removed.***

**- H: in my opinion the authors give too much credit to core SBQ9. This is the only core where multiple pulses are observed. Why is the option that it is in fact an amalgamated turbidite considered unlikely? This core is in the depocenter (this is indeed the location where this could be expected), and apart form this core, only in SQB13 and SQB10 there is perhaps some evidence of such amalgamation, which is indeed also in the depocenter and away from the main (deltaic) sources, where also flow partitioning could get more influence. Furthermore, also event deposit J seems to have 2 pulses in exactly these cores, (and event G!) indicating**

that the presence of multiple "pulses" seems to be related to these locations, rather then to the specific event(s). Also, how do the authors explain these additional earthquakes, while they have not been historically reported?

*The authors agree that an amalgamated turbidite is likely the simplest explanation, rather than a stacked turbidite given the historical reports of shaking do not support the hypothesis of multiple earthquakes. The presence of multiple pulses does indeed likely reflect the location of the cores. This will be fixed in the text.*

*A REVISED ARGUMENT HAS BEEN MADE THAT EXPLAINS DEPOSIT H AS THE RESULT OF AN AMALGAMATED TURBIDITE. PLEASE SEE HIGHLIGHTED SECTION IN THE TEXT.*

**- G: as reverse grading is observed, could this be a catchment response ("flood") related to events H and I? The authors link it to a documented dam failure, in that case the deposit should be coarser and thicker towards the dam (e.g. 1929 dam collapse in Eklutna Lake; Boes et al., 2018), is this the case?**

*REVISED RESPONSE: There is not the distribution of cores that would allow for the evaluation of particle size with distance from the dam (the deep-shallow signal dominates over the distance from the dam). Note that there is a time-gap (sediment accumulation) between H and G, which would not be expected if it were a post-earthquake flood removal of watershed sediment. Post-earthquake watershed removal of sediment would more likely be an immediate response post-earthquake (unless there was a period of less precipitation). It is possible that this is the result of a flood that removes post-seismic sediment.* *YES, Catchment response to earthquake – postseismic removal of sediment. DONE.*

**An alternative interpretation of this sequence would be something similar to what's discussed in Van Daele et al. (2019) (this is anyway a pretty important reference in this paper, as it also deals with the sedimentary imprint of megathrust and intraslab earthquakes and how to distinguish them). As the 1873 earthquake was an intraslab earthquake, the high-frequency content of the shaking could have cause onshore landslides (in contrast to 1700, which would've caused more voluminous deltaic failures due to the longer duration of low frequency shaking). Hence, initially onshore landslides in the schist along the lake could have traveled directly into the lake (event deposit I).**

*Onshore landslides are considered unlikely because deposit I does not look the same brown color as the undercut flood deposits A and B (which are brown because they contain detrital organics at the base). Earthquake-triggered landslides are likely localized and not lake-wide failures, therefore it is considered that deposit I is a subaqueous lake-wide failure (because deposit I is found in all the cores).*

*SUBSEQUENT EVALUATION HAS SUGGESTED THAT DEPOSIT I IS LIKELY A SUBAERIAL SLOPE DEPOSIT AS STATED BY M. VAN DAELE. DONE. VAN DAELE et al. (2019) is included now.*

**The shaking would've also cause delta failures (albeit small ones), which arrive slightly later to the core locations (event deposit H).**

*Both deposits J and H are single (not amalgamated) deposits in all the cores, even in the depocenter, suggesting they are not flow deposits but rather settled directly out of the water column.*

*WE AGREE WITH THE REVIEWER. OUR PREVIOUS COMMENT IS INACCURATE AND HAS BEEN SCRAPPED.*

**Finally, onshore landslide in the catchment would've been transported to the lake in the years following the earthquake (event deposit G). UNLESS there is actually background sediment between event deposits I and H...?**

*Event deposit G does not have a watershed-sourced composition and therefore is not considered to be a flood deposit, which is why it was interpreted to be the result of the dam failure.*

*Our interpretation has changed. Event deposit G is now considered to be a flood deposit. Flood deposits are sourced from both watersheds and would have a mixed composition. We agree that onshore landslide material would have been transported to the lake (event deposit G).*

*There are leaves between deposits I and H, but no intervent sedimentation. Still unsure as to one or two earthquakes, but now suspect two. If two events, then the inference is that the 1873 CE Earthquake is the result of an intraslab earthquake followed immediately by a Cascadia earthquake (deposit H). This is supported by the presence of a small tsunami in coastal southern Oregon (Crescent City Courier, 29 November 1873).*

*DEPOSITS H AND I ARE NOW INTERPRETED TO BE THE RESULT OF THE SAME EARTHQUAKE. THE LEAVES BETWEEN THE TWO DEPOSITS DOES NOT NECESSARILY REFLECT THE PASSAGE OF TIME. This set of deposits has been completely re-evaluated as described in the text.*

**Events C-A: Some 137Cs dates seem to be indispensable to locate the 1963/64 atomic bomb peak and thus confidently attribute the corect deposit to the 1964 floods, and probably also to the 1955 floods.**

*Future work will look for the position of the atomic bomb peak using radiocarbon (and accurately date sediment deposited since ~1955) but this will not be done for this study (given the stage of this manuscript).*

*WE AGREE, BUT ARE UNABLE TO ACQUIRE 137Cs or bomb carbon DATES FOR THIS MANUSCRIPT.*

**Fig. 23: the ratio is probably organic/inorganic, unlike what is mentioned in the caption. This data should be plotted in the same style as all other figures, and both with the same software.**

*The ratio IS organic/inorganic and is labeled as such (caption is in error and will be fixed). Figure 23 could be included in the supplement.*

*THIS FIGURE HAS BEEN REMOVED FROM THE MANUSCRIPT.*

**References:**

Boes, E., Van Daele, M., Moernaut, J., Schmidt, S., Jensen, B. J. L., Praet, N., Kaufman, D., Haeussler, P., Loso, M. G. and De Batist, M. (2018). "Varve formation during the past three centuries in three large proglacial lakes in south-central Alaska." GSA Bulletin 130(5-6): 757-774.

Schwestermann, T., Huang, J., Konzett, J., Kioka, A., Wefer, G., Ikehara, K., Moernaut, J., Eglinton, T.I., Strasser, M., 2020. Multivariate Statistical and Multiproxy Constraints on Earthquake-Triggered Sediment Remobilization Processes in the Central Japan Trench. Geochemistry, Geophysics, Geosystems 21, 1–24. https://doi.org/10.1029/2019GC008861

Van Daele, M., Moernaut, J., Doom, L., Boes, E., Fontijn, K., Heirman, K., Vandoorne, W., Hebbeln, D., Pino, M., Urrutia, R., Brümmer, R. and De Batist, M. (2015). "A comparison of the sedimentary records of the 1960 and 2010 great Chilean earthquakes in 17 lakes: Implications for quantitative lacustrine palaeoseismology." Sedimentology 62(5): 1466-1496.

Van Daele, M., Meyer, I., Moernaut, J., De Decker, S., Verschuren, D. and De Batist, M. (2017). "A revised classification and terminology for stacked and amalgamated turbidites in environments dominated by (hemi) pelagic sedimentation." Sedimentary Geology 357: 72-82.

Van Daele, M., Araya-Cornejo, C., Pille, T., Vanneste, K., Moernaut, J., Schmidt, S., Kempf, P., Meyer, I. and Cisternas, M. (2019). "Distinguishing intraplate from megathrust earthquakes using lacustrine turbidites." Geology 47: 127-130.

Weltje, G.J., Bloemsma, M.R., Tjallingii, R., Heslop, D., Röhl, U., Croudace, Ian W., 2015. Prediction of Geochemical Composition from XRF Core Scanner Data: A New Multivariate Approach Including Automatic Selection of Calibration Samples and Quantification of Uncertainties, in: Croudace, I.W., Rothwell, R.G. (Eds.), Micro-XRF Studies of Sediment Cores. Springer Dordrecht, pp. 507–534. https://doi.org/10.1007/978-94-017-9849-5_21

Wils, K., Deprez, M., Kissel, C., Vervoort, M., Van Daele, M., Daryono, M. R., Cnudde, V., Natawidjaja, D. H. and De Batist, M. (2021). "Earthquake doublet revealed by multiple pulses in lacustrine seismo-turbidites." Geology 49(11): 1301-1306.

**ALMOST ALL MINOR HAND-WRITTEN COMMENTS ON THE DRAFT WERE ADDRESSED.**

**Shmuel Marco comments to manuscript (bold), with author responses in italics.**
**AUTHOR-ADDRESSED REVISIONS IN THE VERSION SUBMITTED TO COPERNICUS ON APRIL 17, 2024 ARE IN RED.**

**Review by Shmuel Marco, 27 Sep 2023**

**Sorting out the various causes for changes in the sediment character is challenging in light of the multiple possible triggering processes, both natural and man-made.**

**The authors use numerous analytical methods for characterizing the layers suspected as seismites generated by a particular earthquake. The discussion on the depositional processes and how they were resolved is somewhat tedious and hard to follow, but I cannot suggest an improvement. The information is indeed relevant to the conclusions, so I suppose it better remain as is. The elaborate discussion is crucial because the radiocarbon dates cannot bracket the events tightly enough. Therefore, I value the submission as an example for applying multiple considerations in order to reach the best fit solution to a complex geological situation.**

**My main criticism is that the authors consider only mass transport deposits and do not ignore the option of sediment-water interaction during seismic shaking. This was shown to be significant in many previous works, in particular the ones related to the paleo Dead Sea seismites (e.g., Wetzler et al., 2010). Previous research also addressed the difference between earthquake-triggered in-situ deformation of lake bottom sediments and slope-originated mass transport deposits. However, this extensive body of works (e.g., Lu et al., 2017, 2020) is unfortunately ignored here.**

RESPONSE: *Thank you for bringing to our attention the articles on Dead Sea seismites. Although there are seismites present in the downcore record, the upper portion of the record does not have any obvious seismites. There are load structures that are the result of rapid settling of silt through the water column, but these are likely not the result of in-situ deformation of lake bottom sediments. We should make our reasoning clear in the manuscript. Also, the deposits that produce the load structures are interpreted to be the result of injection, then settling, of silt into the water column . . . and therefore are not mass transport deposits.*

*THE AUTHORS HAVE NOT OBSERVED ANY EVIDENCE OF SEDIMENT-WATER INTERACTIONS (SUCH AS IN WETZLER ET AL., 2010), OR IN-SITU DEFORMATION OF LAKE BOTTOM SEDIMENTS (LU ET AL. PAPERS) IN THE HISTORICAL PORTIONS OF THE CORES.  It is difficult to reference these papers for this manuscript, but very appropriate to include in the companion manuscript #1638.*

**Minor comments:**

**The authors conclude that their results "suggest that inland lakes can be sensitive recorders of earthquakes". This is old news since evidence that inland lakes can be sensitive recorders of earthquakes has been around for over three decades. It is not a new revelation of this research.**

*RESPONSE: This comment refers to inland lakes in Cascadia. This will be rephrased.* **FIXED**

**Line 27: Earlier, much longer earthquake records (220 ka) have been reported from the Dead Sea Basin, where seismites are directly linked to synsedimentary faults and historical accounts of earthquakes (Lu et al., 2020 and references therein).**

**ADDED**

**L. 406: Flood deposits usually sink to the bottom within hours, even in saline lakes, where the debris/brine density contrast is smaller than in fresh water.**

**THANK YOU**

**L 585: "steep" is too vague, please provide a measure of the slope. Slope failures can occur on less the 1°.**

**DONE**

*RESPONSE: Thank you for these comments!*

**The bottom line: Accept with minor revision**.

**References**

Lu, Y., Moernaut, J., Bookman, R., Waldmann, N., Wetzler, N., Agnon, A., Marco, S., Alsop, G.I., Strasser, M., and Hubert-Ferrari, A., 2021, A New Approach to Constrain the Seismic Origin for Prehistoric Turbidites as Applied to the Dead Sea Basin: Geophysical Research Letters, v. 48, doi:10.1029/2020GL090947.

Lu, Y., Waldmann, N., Ian Alsop, G., and Marco, S., 2017, Interpreting Soft Sediment Deformation and Mass Transport Deposits as Seismites in the Dead Sea Depocenter: Journal of Geophysical Research: Solid Earth, v. 122, p. 8305–8325, doi:10.1002/2017JB014342.

Lu, Y., Wetzler, N., Waldmann, N., Agnon, A., Biasi, G.P., and Marco, S., 2020, A 220,000-year-long continuous large earthquake record on a slow-slipping plate boundary: Science Advances, v. 6, p. eaba4170, doi:10.1126/sciadv.aba4170.

Wetzler, N., Marco, S., and Heifetz, E., 2010, Quantitative analysis of seismogenic shear-induced turbulence in lake sediments: Geology, v. 38, p. 303–306, doi:10.1130/G30685.1.

---

## Author Response (AR2)

**Report #1**

Submitted on 03 Jun 2024
Anonymous referee #3

**Anonymous during peer-review: Yes** No
**Anonymous in acknowledgements of published article: Yes** No

**Checklist for reviewers**

**1) Scientific significance**
Does the manuscript represent a substantial contribution to the understanding of natural hazards and their consequences (new concepts, ideas, methods, or data)?

Excellent **Good** Fair Poor

**2) Scientific quality**
Are the scientific and/or technical approaches and the applied methods valid? Are the results discussed in an appropriate and balanced way (clarity of concepts and discussion, consideration of related work, including appropriate references)?

Excellent Good **Fair** Poor

**3) Presentation quality**
Are the scientific data, results and conclusions presented in a clear, concise, and well-structured way (number and quality of figures/tables, appropriate use of technical and English language, simplicity of the language)?

Excellent Good **Fair** Poor

**For final publication, the manuscript should be**

accepted as is.

accepted subject to **technical corrections**.

**accepted subject to minor revisions.**

reconsidered after **major revisions**:

**rejected**.

**Were a revised manuscript to be sent for another round of reviews:**

I would be willing to review the revised manuscript.

**I would not be willing to review the revised manuscript.**

**Suggestions for revision or reasons for rejection**

(visible to the public if the article is accepted and published)

The authors develop an original technique of evaluating the XRF data of the event deposits in scatter plots and look for distinct patterns that help understanding the depositional processes and source regions of the event deposits. This scatter-plot technique is commonly applied for grain size data on event deposits, but using this approach on XRF data is a first time and bears a great potential in other studies as well. The authors make an extensive argumentation that they use the raw XRF data and not the log-ratio transformation of the data (a transformation suggested by the reviewer and which is common practice in sediment geochemistry. As raw counts is not geochemistry data, but rather a mixture of density, grain size and element abundance, interpretation is becoming more difficult. I suggest the authors to re-consider the usage of the CLR approach. Using the Centre-log-ratio (CLR) is important to decrease the effect of grain-size and density and get geochemical robust data from XRF scans. The authors mention that the CLR approach can produce more "noise", which is true. Therefore, it is crucial to select only elements with high signal-to-noise ratio for the calculation of the geometric mean, and to remove any 0 or negative values. So, I suggest the authors to re-consider the usage of the CLR approach on a selected suite of high-quality elements and evaluate the quality of the results. Of course, the final decision is up to the authors on what they consider the most meaningful.

AUTHOR RESPONSE: Yes, the XRF data is a mixture of density, grain size and elemental abundance. The use of the raw (not transformed) XRF data is important because grain size, porosity, and density are implicit variables in addition to the elemental abundance and elemental patterns with grading only show up when viewing the raw data. In otherwords, we are not strictly evaluating geochemistry, but rather the approximate geochemistry with variables that change along with grading (porosity, grain size, density, etc.). This is what allows for the geochemistry to be evaluated as loops (which reflect deposit grading). This cannot be seen in the CLR data because the influence of the other variables has been removed. This is an essential component of this study. Because of the use of the variables in this way we were able to infer processes not as apparent from the log-ratioed XRF. Although both K and Ca have high signal-to-noise ratios, we are limited to these variables as they represent the two endmember sources. If we wanted to get true geochemistry data from XRF, we would use the CLR method, but we are interested in getting the combination of approximate geochemistry and grainsize/density through these graded deposits.

NOTE: The reviewers state the novel approach of this method has value when they state that "but using this approach on XRF data is a first time and bears a great potential in other studies as well"

In any case (raw counts or CLR transformed data), the data presented in Fig. 19 B (K/Si x Density) and interpretation is not acceptable to me. I do not understand the scientific meaning of this procedure. If radiodensity is inherently part of the raw XRF counts, I suppose you just increase this effect by multiplying XRF data with CT density? The resulting quantity is in HU, what would be the physical meaning of this value? Please elaborate it properly and make this more convincing in the article, or alternatively, delete this part.

AUTHOR RESPONSE: This was an attempt to show that the XRF loops are the result of the combination of geochemistry and the other implicit variables (such as radiodensity). Yes, the ratioed data (K/Si) times density was displayed simply to demonstrated that radiodensity is inherently part of the raw XRF counts. There is no meaning to the units, they are simply a way to scale to show how Log-ratio data for the two endmember variables (K, Ca) are influenced by other variables (in part, sediment density) during grading.

For the loops in the XRF scatter plots: the authors rely on the assumption that the background sediment before and after the event is exactly the same, and based on this try to identify the base and top of the event deposit (see event J: Fig. 20). This assumes that earthquakes do not produce any long-term catchment response and no other paleo-environmental changes (human influences in the catchment, climate changes, etc) occurred in the specific timeframes as this would lead to background sediments with different values. This assumption should be evaluated and clearly stated in the manuscript.

AUTHOR RESPONSE: We do not "rely" on the assumption that background sediment before and after the event is exactly the same, but suggest this as a potential explanation, which is supported by changes in sediment density. Since an earthquake does not last very long, it is reasonable to assume that there would be little change in the relationship between raw K and Ca relationships just after earthquakes, especially since any watershed influence would occur some time (weeks to months?) after the earthquake. This interpretation that the start and end close the grading loop is supported by the fact that the identification of these starting and ending points coincide with other variables that also signify the beginning and end of the deposits, including changes in radiodensity, and biogenic composition.

I agree with one of the previous reviewers that the authors invoke several new mechanisms (which have not been documented elsewhere) to explain observations that can be more easily explained by widely recognized processes (e.g. delta failure). I am not going to summarize these (this was done by the previous reviewer). This extra process complexity is not necessary in my opinion, but accept it to be the choice of the authors to put forward these ideas and hypotheses. What matters most for good science is to present robust data in the best way possible, whereas interpretations may change over time. The authors discuss their interpretations in an extensive way, which is required to for the readership to follow the logic argumentation.

AUTHOR RESPONSE: The delta failure mechanism cannot explain the large addition of Ca that produces the subduction earthquake deposits and therefore a new mechanism (liquefaction) was required. Likewise, the change in water column biogenic components can be explained by flocculation which removes primary producers and other components from the water column. Note that we describe the interpretations extensively.

Several figures need to be improved (see list below).

Overall, I recommend moderate revisions.

Detailed comments:

Abstract:
I suggest to write "Cascadia megathrust earthquakes", not just "Cascadia earthquakes".
DONE

Methods: you write that mineralogy was spot-checked using the CSD desktop scanning electron microscope. Where do you show data for this? Also, in the results you mention: "Inspection by Energy-dispersive x-ray spectroscopy (Bruker Quantax 50 EDS; CSD Facility) showed that some mica flakes from the Condrey Mountain Schist surrounding the lake contain a large amount of carbon (as much as 77%) and scanning-electron-microscopic analysis shows the presence of pyrite." You do not show that data. Please add this data to the supplement.
AUTHOR RESPONSE: We do not show data because there isn't any. We just periodically took samples and used the SEM to check what we were seeing. We have some SEM images, but do not have all the minerals in them identified. We were looking for micas, specifically, and a few other minerals to confirm what we were seeing. Our sampling was very inconsistent. Maybe we should remove this statement as a result. The Bruker Quantax data is similar in that we were simply looking at samples and noticed in one sample that 77% was carbon.

Discussion:
L404: please update "Only one study (Van Daele et al., 2019) has successfully discriminated between intraplate and megathrust earthquakes using lake sediments." See e.g. Praet et al., 2022 https://onlinelibrary.wiley.com/doi/abs/10.1111/sed.12986
DONE
L405-410: this explanation is partly correct. The intraslab events in that study also produce thin coseismic turbidites in that lake, AND are followed by catchment response.
DONE
L465: I suggest to also refer to the review paper of Bertrand et al., 2024 which is discussing all these issues. https://www.sciencedirect.com/science/article/pii/S0012825223003288
DONE
L472: "When scaled by radiodensity (Figure 19b)," I do not understand the scientific meaning of this calculation. If radiodensity is inherently part of the raw XRF counts, I suppose you just increase this effect by multiplying XRF data with CT density. What would be the purpose of this? Please elaborate and make this more convincing in the article, or alternatively, delete this part.
DONE. What we have attempted to do is remove the influence of artifacts (including density) by normalizing, then show that scaling by radiodensity reproduces the patterns in the xy plots of the raw data. The purpose was to demonstrate that sediment density is likely an implicit variable in the raw data.
L619: Is SQB5 really located on a ~45° slope? I would not expect any sediment accumulation on slopes steeper than 25°, so that seems strange. Please verify the slope angle calculation you did.
REMOVED THIS ANGLE. THE DATA ARE NOT DETAILED ENOUGH TO

CALCULATE ACCURATELY.

To me, the "summary of interpretations" and "conclusions" contain to much repetition. I suggest to keep the conclusions short and focus on the implications of your study rather than explaining the nature of each event deposit again.

AUTHOR COMMENTS: The summary of interpretations contains repetition because the discussion of the deposits was very long and complex. It seems important to follow with a simplified summary of the interpretations. We agree that this, then, does not need to be repeated in the conclusions.

Clearness and structure of the figures (also in the supplement):

In general, I think that the figure captions are often too large. They should help the reader to understand the figure, but do should not go into the interpretation of the data. This is done in the main text of the article. I suggest to shorten the figure captions to avoid repetition. E.g. a discussion sentence "could this be the result of a post-subduction earthquake aftershock, another subduction earthquake, a large crustal or intraplate earthquake?" should not be in a figure caption (Fig. 20). FIXED FIG. 20 CAPTION. SHORTENED MANY CAPTIONS.

Labeling and units: many figures are missing key elements such as labels and units for the axes. Also, several cores and core plots (such as CT density) are missing a depth scale! (e.g. fig. 21). FIXED.

Fig. 1: what are the blue lines? Are these the faults from the simplified Cascadia forearc fault model? YES. NOW IN CAPTION.

Fig. 3: How is this "spike" in conductivity obtained and why is it not part of the grey curve? In which month is the data measured (important for the development of a thermocline)? THIS VALUE WAS UNSTABLE AND THEREFORE WAS NOT CONNECTED TO THE GREY CURVE. THIS WAS THE MAXIMUM IT FLUCTUATED AT THAT DEPTH. Now indicated in caption.

Fig. 4: many cores were taken on slopes, which can make the record more complicated. As seen in e.g. Molenaar et al.,2021 (https://onlinelibrary.wiley.com/doi/full/10.1111/sed.12856) a slope angle of a few degrees can be enough to have stratigraphic gaps caused by surficial remobilization (due to earthquakes) and to develop in-situ soft sediment deformation structures. It would be good for the reader to get information about the coring strategy.

DONE. This was a problem during coring. The lake is very narrow and prevailing winds made it very difficult to stay on location to core the lake's depocenter (which was the intent).

Fig. 5: The legend shows many lithology symbols, but none of them are used in the figure. FIXED

Fig.5 and 6: It would be useful to indicate the top and bottom of each event deposit. In this way, it is much easier for the reader to see how the event deposits and the background sediments change throughout the lake basin. CAN'T DO THAT BECAUSE WE USED XRF LOOPS TO DETERMINE THE BASE AND TOP OF THE DEPOSITS, AND THAT ISN'T DONE UNTIL FURTHER INTO THE DOCUMENT.

Fig. 7: typo in legend "vivianiate" FIXED

Fig. 10: there is way too much text next to the smear slides FIXED

Fig. 11: At the bottom left there are some plots which are illegible. REMOVED
Fig. 12: The age model is illegible FIXED
Fig. 18: A) Please provide an arrow so we see the direction of the "loops" DONE
Also, why do you use the sequence number as X-axis in Fig. C? It would be easier for the audience if you just plot sediment depth. Same for Fig. 19 DONE
Fig. 19: The colors related to some event deposits have changed compared to Fig. 18. This makes comparison a tedious task. E.g. deposit I is purple (Fig. 18) or red (Fig. 19)… FIXED
Fig. 20: the legend states "confidence intervals in orange". However, I do not see any orange in the plots. IT IS THERE, NEAR THE TOP OF THE XY PLOT.
Fig. 22 (and others): "Note that the horizontal axis for the cartoon of the deposit sequence (bottom right) represents sediment radiodensity." Good that you mention this, but it should be mentioned for all figures that show such a cartoon (e.g. Fig. 10). Otherwise, any sedimentologist in this world would think that the horizontal axis in these cartoons represent grain size and not radiodensity. DONE
Fig. 23: This caption needs more explanation: which deposit are you using as representative of "flood" and of "earthquake"? Also indicate the arrow on the XRF data loops. DONE

Supplement:

2a: XRD figures: screenshots: data and scales are illegible. Can't fix because screenshots were all that were available.
2B: XRD data: the plot axes have no units. Also, the core has no depth scale. FIXED
3. I do not see any "red fault lines "on the map. I guess you refer to the black lines? As these are based on tracing lineaments on topographic data, the map would be more convincing if it would include topography information (i.e. DEM of the region). Yes, black lines. FIXED.
4. The table headers need units. DONE